# Costs of transitioning the livestock sector to net-zero emissions under future climates

Franco Bilotto [1,2,3], Karen Michelle Christie-Whitehead [4], Bill Malcolm[5], Nicoli Barnes[4,6], Brendan Cullen[5], Margaret Ayre[5] & Matthew Tom Harrison [1] ✉

Land managers are challenged with the need to balance priorities in production, greenhouse gas (GHG) abatement, biodiversity and social license to operate. Here, we develop a transdisciplinary approach for prioritising land use, illustrated by co-designing pathways for transitioning farming systems to net-zero emissions. We show that few interventions enhanced productivity and profitability while reducing GHG emissions. Antimethanogenic feed supplements and planting trees afforded the greatest mitigation, while revenue diversification with wind turbines and adoption of livestock genotypes with enhanced feed-conversion efficiency (FCE) were most conducive to improving profit. Serendipitously, the intervention with the lowest social licence—continuing the *status quo* and purchasing carbon credits to offset emissions—was also the most costly pathway to transition to net-zero. In contrast, stacking several interventions to mitigate enteric methane, improve FCE and sequester carbon entirely negated enterprise emissions in a profitable way. We conclude that costs of transitioning to net-zero are lower when interventions are bundled and/or evoke productivity co-benefits.

Anthropogenic greenhouse gas (GHG) emissions are intensifying the global water cycle and frequency of extreme weather events[1,2]. Since 1980, natural disaster occurrence has quadrupled, causing aggregate losses of 280 billion US dollars (US$280B) in crop and livestock production[3]. Despite this, the vast majority of research hitherto has focused on gradual climate change, perhaps because the prediction of and effective adaptation to extreme weather events is much more challenging[4,5].

The concept of carbon neutrality (or net-zero carbon emissions) refers to the balancing of GHG emissions with removals, often measured in carbon dioxide equivalents ($CO_2$-equivalents). In contrast, 'climate neutrality' refers to net-zero atmospheric temperature change[6]. Pathways to either carbon or climate neutrality typically comprise a combination of strategies[7] aimed at limiting GHG emissions from entering the atmosphere (e.g. provision of livestock feed additives to reduce enteric methane ($CH_4$)) and/or reducing atmospheric GHG through additional removals (e.g. enhanced soil organic carbon

[SOC] accrual)[8–10]. 'Mitigation' can thus be either reduction and avoidance (reducing and/or eliminating the quantum of GHG entering the atmosphere) and/or removal (withdrawal of GHG from entering the atmosphere as a result of deliberate human activities)[7–12]. Mitigation can occur via 'insetting', where new removals or reductions are used to counterbalance baseline GHG within an enterprise (e.g. a practice change resulting in additional SOC sequestration[10]) or value chain (e.g. reducing GHG emissions associated with transport of farm products[13]).

Greenhouse gas emissions from an entity may also be 'offset', defined here as the reduction, avoidance or removal of one GHG unit by an entity, purchased by another entity to counterbalance a unit of GHG by that other entity[12]. Offsets are subject to strict environmental integrity criteria to ensure that carbon credits realise their stated mitigation, including avoidance of double counting and leakage, use of appropriate baselines, additionality, transparency, conservativism,

[1]Tasmanian Institute of Agriculture, University of Tasmania, Newnham, Launceston, TAS, Australia. [2]AgResearch, Grasslands Research Centre, Tennent Drive, Palmerston North, New Zealand. [3]Department of Global Development, College of Agriculture and Life Sciences, Cornell University, Ithaca, NY, USA. [4]Tasmanian Institute of Agriculture, University of Tasmania, Burnie, TAS, Australia. [5]School of Agriculture, Food and Ecosystem Sciences, The University of Melbourne, Parkville, VIC, Australia. [6]IEAC Federation University Australia, VIC Berwick, Australia. ✉e-mail: matthew.harrison@utas.edu.au

permanence (or measures to address impermanence), measurability and independent verification[12]. Offsets can be quantified in 'carbon credits', a market instrument in the form of a tradable certificate representing one tonne of $CO_2$-equivalent emission reduction, avoidance, or removal as a result of a project, intervention, or activity[14,15]. Carbon taxes are monetised levies on net positive GHG emissions of an entity, and are often greater than the price per unit carbon credit to account for the true damage society causes by each incremental tonne of $CO_2$[16,17].

To realise and sustain net-zero emissions, the agrifood sector must reduce carbon dioxide ($CO_2$), $CH_4$ and nitrous oxide ($N_2O$) emissions, which according to the Intergovernmental Panel on Climate Change (IPCC), contribute 10.8-19.1 gigatonnes (Gt) $CO_2$e year$^{-1}$ or 21–37% of global anthropogenic GHG emissions[14,18]. Of those agrifood emissions—comprising GHGs from agriculture, land use, storage, transport, packaging, processing, retail, and consumption[14]—farm-level crop and livestock activities account for 9–14% of global anthropogenic GHG emissions ($6.2 \pm 1.4$ Gt $CO_2$e year$^{-1}$ or $11.1 \pm 2.9$ Gt $CO_2$e year$^{-1}$ including land use)[14]. Proposed mitigation innovations must however go beyond consideration of GHG reduction or removal by also accounting for co-benefits and trade-offs on food security, conservation or restoration of natural resources, ecosystem services and climate change impacts[19,20]. Achieving deep, sustained reduction of enteric $CH_4$ emissions, which in 2019 accounted for 23% of anthropogenic agrifood GHG emissions (2.9 Gt $CO_2$e year$^{-1}$)[21–23] or 40% of global agricultural (farm level) GHG emissions, requires innovations in livestock feed, genetics, reproduction and health management[24,25]. While there has been substantial research of individual mitigation interventions in isolation (e.g. the influence of ewe fecundity or pasture type[26,27]), few studies have explored holistic implications of bundling practices aimed at simultaneous $CO_2$ removal, GHG emissions reduction and climate change adaptation.

Carefully conceived adaptations may enable food systems transformation, but only if due consideration is given to a wide range of influential socioeconomic, institutional and cultural factors[28,29]. As a corollary, few bona fide examples of agrifood systems transformations exist, perhaps because research has traditionally progressed in a reductionist fashion, with primarily unidisciplinary and siloed foci. This has emanated 'carbon myopia' phenomena[20], wherein only GHG emissions reduction, avoidance or carbon removals are assessed for innovations purported for GHG mitigation. Effects of and interactions caused by such interventions with extraneous factors, such as prosperity, productivity, regulatory barriers, environmental stewardship and social license to operate, are often downplayed or ignored completely, even though such factors collectively determine whether or not an intervention will be sustainably perpetuated[20,30]. The present study addresses this gap through development and operationalisation of a transdisciplinary holistic systems approach designed to navigate trade-offs between production, profit and GHG emissions under an increasingly variable climate. Compared with unidisciplinary approaches, transdisciplinary work (cross discipline and cross institutional, respectively) tends to be more difficult to lead, and more costly in time and money, hence the majority of GHG emissions mitigation research continues to be promulgated in siloed pockets[20,30,31].

Most previous climate change adaptation and mitigation work for the livestock sector has been conducted through a biophysical lens. Such studies have examined, for example, (1) GHG emissions of cropping and livestock systems[32], (2) GHG emissions from model ensembles[33], and (3) the influence of genotype by management by environment combinations on GHG emissions and productivity[34–37]. Much less emphasis has been placed on understanding how interventions aimed at adaptation and/or mitigation influence productivity, profitability and GHG emissions of the beef and sheep industries[38,39]. While land managers have multiple opportunities to mitigate GHG emissions (e.g. through carbon removals, and GHG emissions avoidance and reduction), scientific literature that develops and contrasts economic pathways to carbon-neutral farming systems is scarce.

Here, our aim is to (1) co-develop a range of management, genetic, environmental, livestock and landscape interventions for adapting livestock systems to the changing climate while reducing GHG emissions, (2) quantify the costs of plausible pathways to net-zero emissions and (3) co-design interventions with a 'regional reference group' (RRG) of industry practitioners to ensure relevance, credibility and legitimacy of our proposed adaptation/mitigation interventions[31]. We calibrate our models using two real farms in southern Australia, iteratively refining methods based on feedback from the RRG, then explore the impact of singular and stacked (bundled) interventions on productivity, profitability, GHG emissions and adoptability[40]. Stacked interventions are categorised into groups based on similarity of intent, including 'Low Hanging Fruit' (simple, reversible, immediate changes that could be made to the farm system), 'Towards Carbon Neutral' (interventions designed to reduce GHG emissions year on year), 'Income Diversification' (enabling revenue generation from enterprises other than livestock to reduce dependence on rainfall as a primary source of income) and 'Transformational' (innovations realised over a long term with high downside and upside risk). While we exemplify our methods using two case studies, our approach can be generically adapted to any location, production system or transdisciplinary problem. The scope of each case study was the farm enterprise, noting that such enterprises may have multiple land parcels that are geographically distinct.

## Results

### The nexus between productivity, profitability and net greenhouse gas emissions

In comparing baseline (*status quo*) beef and sheep production systems under historical climates (1986–2005) with baseline management of the same farm in 2030 and 2050 (i.e. cf. Figures 1a, c, 2a, c with Figs. 1b, d, 2b, d), we show that (1) few individual interventions elicited significant simultaneous benefit on all indicators (productivity, profitability, GHG emissions mitigation, adoptability) and (2) interventions caused greater effects on these indicators compared with impacts of climate change alone (Figs. 1 and 2). The combination of higher monthly temperatures (4–14%) and lower rainfall in 2030 (3–7%) and 2050 (5–11%) with elevated atmospheric $CO_2$ concentrations evoked modest increases in pasture production of the beef farm (2–3%) and sheep farm (7-8%) compared with historical climates. This translated into incremental gains in meat and wool production and reduced supplementary feed requirements, resulting in greater profit for the beef farm (3%) and sheep farm (36%) in 2050; the larger gain for the latter underpinned by a larger reduction in supplementary feed requirement in 2050 cf. historical climates.

Individual interventions targeting livestock enteric $CH_4$ ($CH_4$ produced by fermentation in the gut) were most promising for reducing GHG emissions, such as the seaweed feed additive *Asparagopsis taxiformis*. Assuming 80% enteric $CH_4$ mitigation based on peer-reviewed evidence[41–44], *Asparagopsis* feed additive reduced farm enterprise $CO_2$e by 46–72% under future climates (Fig. 1a, b, 2a, b, supplementary tables 1–4). However, this was also one of the most expensive singular interventions, reducing profits by \$23–25 megagram (Mg) $CO_2$e$^{-1}$ mitigated (Figs. 1c, d, 2c, d). In contrast, interventions that were considered most adoptable by the group of expert practitioners (the RRG) often had the lowest mitigation potential (Figs. 1 and 2).

Climate diversification—purchasing a farm in a distinctively different climatic zone and altering lambing/calving times accordingly—evoked the greatest improvement in productivity (16–18%), while enterprise diversification (capital investment to enable income generation from irrigated grapevines or wind turbines), pasture renovation with deep-rooted legumes and improvements in animal genetic

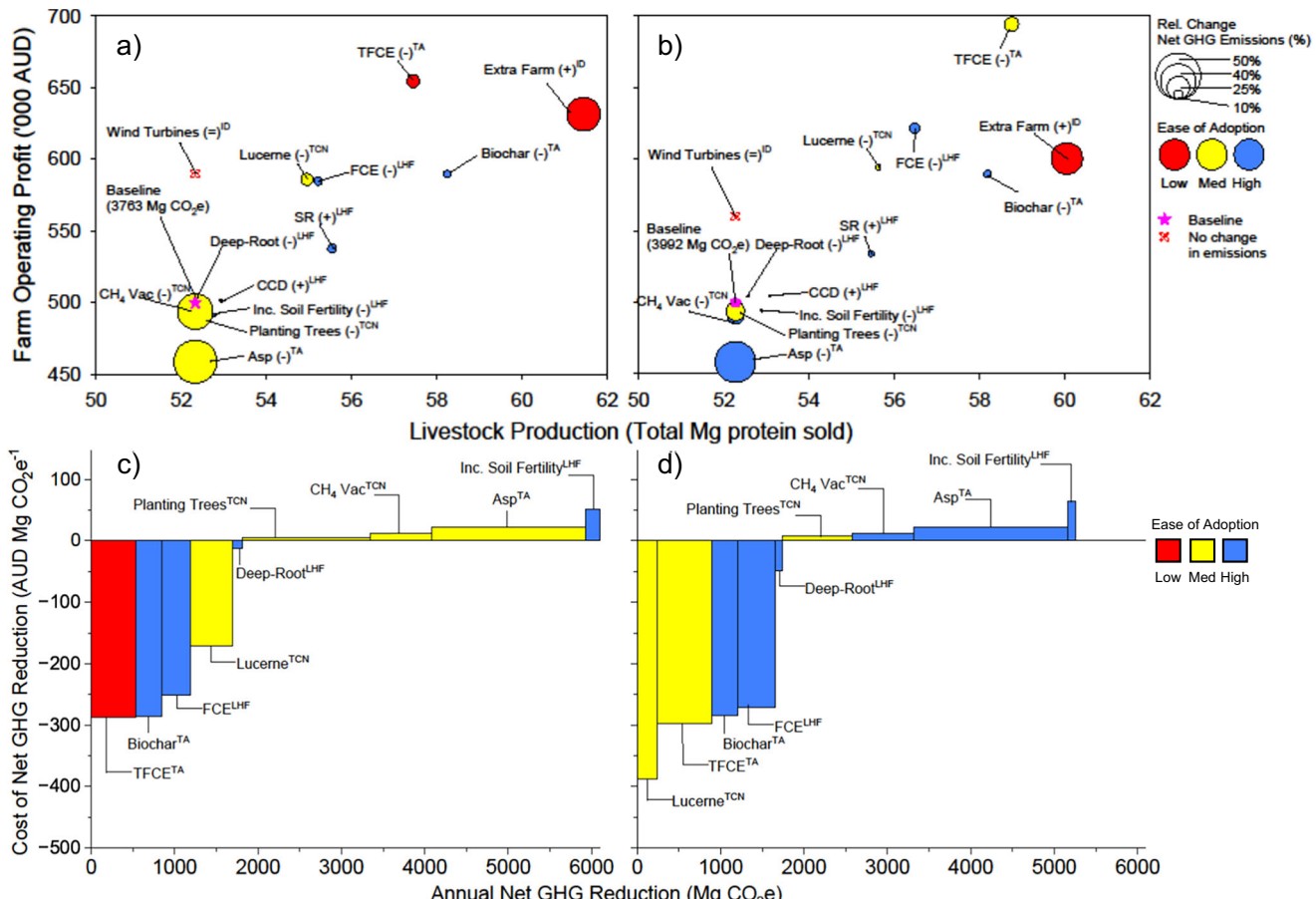

**Fig. 1 | Trade-offs between profit, production, mitigation and adoptability for multiple adaptation/mitigation interventions to beef farms under 2030 and 2050 climates. a, b** show the relationship between production and profit with change in GHG emissions and adoptability. **c, d** show marginal abatement cost curves, with trade-offs between mitigation quanta and cost of intervention. Adaptation/mitigation interventions were co-designed with a Regional Reference Group of expert practitioners for 2030 (**a, c**) and 2050 (**b, d**) climates, LHF Low Hanging Fruit, TCN Towards Carbon Neutral, ID Income Diversification, TA Transformational Adaptation. Purple stars depict the baseline scenario. Total emissions for the baseline scenario shown in parentheses in (**a**) and (**b**). Asp *Asparagopsis taxiformis* as a feed supplement; CH4 vac injecting animals with an enteric $CH_4$ inhibitor vaccine, CCD changing calving date, Deep-Root increasing pasture root depth, FCE increasing livestock feed conversion efficiency, SR increasing stocking rate, TFCE transformational increases in livestock feed conversion efficiency. The operators represent a relative increase (+), reduction (−), or no changes (=) in GHG emissions compared to the baseline.

feed-conversion efficiency (FCE) were most conducive to improved profit (17–39%). Interventions that achieved the greatest gains in productivity and profit often had little influence on GHG emissions, underscoring challenges in decoupling the recalcitrant linkage between productivity and GHG emissions[20].

Improving FCE was operationalised by increasing pasture utilisation and liveweight gain. This increased profit ($70–250 Mg $CO_2e^{-1}$ mitigated; Figs. 1c, d, 2c, d, supplementary tables 1–4), but only had modest impacts on productivity (0–6% increase) and GHG abatement (−9 to 15% reduction). Transformational improvement in animal genetic feed conversion efficiency (TFCE) increased livestock production and profit by 8–39% while reducing net GHG emissions by 11–17% (Figs. 3a, d, 4a, d). While our TFCE target (20–30% gain in animal FCE) is aspirational, our assumptions are grounded on advice from eminent livestock geneticists (Dr Rob Banks pers. comm.) and peer-reviewed literature[45], ensuring that results are robust (see methods).

We revealed that climate change had significant ramifications for carbon removal quanta, with warmer climates increasing evapotranspiration, reducing the length of the pasture growing season, increasing soil respiration and impacting on the duration of soil carbon sequestration. By 2050, GHG mitigation potential associated with soil carbon accrual was reduced by 6–13% for interventions that expanded farm area covered by deep-rooted perennial legumes

(lucerne or *Medicago sativa*), and by 20–40% for carbon sequestered by planting native vegetation (Figs. 1 and 2; supplementary fig. 1, supplementary tables 1–4). Planting trees on farm decreased profits for each unit of $CO_2$ mitigated compared with incorporating lucerne into pastures (Figs. 1c, d, 2c, d). This occurred because lucerne enabled pasture growth and improved livestock production, whereas planting trees was assumed to represent a new investment (beef farm) or occur within remnant vegetation (sheep farm), with no ensuing effect on livestock production. This assumption was made for conservatism, acknowledging that some tree species could provide productivity co-benefits via provision of forage, shelter or pasture production co-benefits[46,47].

The RRG considered biochar feed supplementation as highly adoptable based on ease of implementation and comparison with other interventions. Grounding liveweight gains and enteric $CH_4$ mitigation on peer-reviewed evidence[48–51] (supplementary fig. 2), we showed that biochar feed supplementation reduced net GHG emissions by 8% and increased profit of the cattle enterprise by 18% (Fig. 1c, d), but reduced profit of the sheep enterprise by 10% (Fig. 2c, d). The aforementioned studies do not account for upstream (pre-farm) GHG emissions associated with biochar production[52], which may reduce perceived climate benefits at the farm scale. Even so, biochar has benefits additional to mitigation, including recycling of

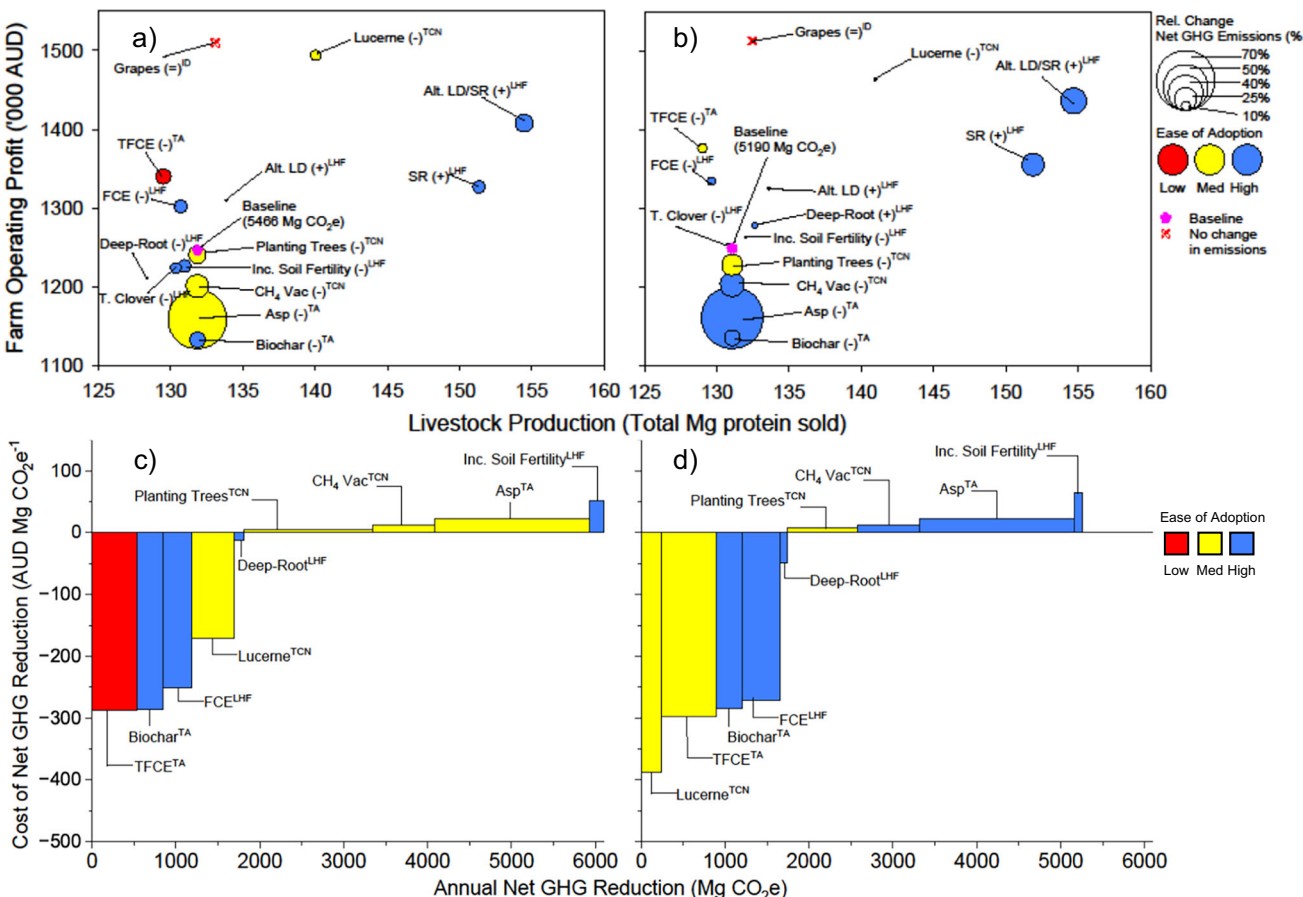

**Fig. 2 | Trade-offs between profit, production, mitigation and adoptability for multiple adaptation/mitigation interventions to sheep farms under 2030 and 2050 climates. a, b** show the relationship between production and profit with change in GHG emissions and adoptability. **c, d** show marginal abatement cost curves, with trade-offs between mitigation quanta and cost of intervention. Adaptation/mitigation interventions were co-designed with a Regional Reference Group (RRG) of expert practitioners for 2030 (**a, c**) and 2050 (**b, d**) climates, LHF Low Hanging Fruit, TCN Towards Carbon Neutral, ID: Income Diversification, TA Transformational Adaptation. Purple stars depict baseline scenario. Total emissions for the baseline shown in parentheses in (**a**) and (**b**). Alt. LD altered lambing date, Alt. LD/SR altered lambing date and increased stocking rate, Asp Asparagopsis taxiformis as a feed supplement, $CH_4$ vac injecting animals with an enteric $CH_4$ inhibitor vaccine, Deep-Root increasing pasture sward root depth, FCE increasing livestock feed conversion efficiency, SR increasing stocking rate, T. Clover introduction of Talish clover (Trifolium tumens), TFCE transformational increase in livestock feed conversion efficiency. The operators represent a relative increase (+), reduction (−), or no changes (=) in GHG emissions compared to the baseline.

agricultural or forestry waste in line with the circular economy[53], and potential to displace energy derived from fossil-fuels through electricity generation and/or eucalyptus oil production[54]. These observations underline a need for more systematic, holistic assessments that evaluate environmental, agronomic and economic implications associated with using biochar as a livestock feed supplement.

To guard against downside risk of drought under future climates, avenues enabling income independence of rainfall were co-designed. These interventions included planting a small area of irrigated grapevines on the sheep farm, leasing part of the beef farm to an energy company to construct wind turbines, and climatic diversification by purchasing a block of land for cattle farming in a distinctively different climatic zone to that of the existing beef farm. While wind turbines, developing irrigated grapevines and purchasing additional land for beef cattle production improved enterprise profits by 12–18%, 20% and 15% respectively (Figs. 1 and 2), purchasing additional land in a diverse agro-climatic region also increased GHG emissions (net and emissions intensity, supplementary tables 1 and 2) because grazing area and total animal numbers increased (supplementary table 5). While the new land received a similar annual rainfall quantum as that of the existing farm in north-western Tasmania, the sandy loam soils (Kurosols) of the

north-east were less productive than those of the clay-rich Ferrosols in the north-west[55], reducing annual pasture production by 40% (supplementary tables 1, 2 and 5). As such, the north-east region has lower sustainable stocking rate and SOC accrual (supplementary table 5). Taken together, these factors were conducive to higher net GHG per unit livestock production, resulting in greater net GHG emissions per unit enterprise protein (supplementary tables 1 and 2). Our analysis of purchasing of additional land highlights the tight coupling between livestock numbers and GHG emissions, as well as the influence of SOC sequestration on protein emissions intensity. While purchasing land in a distinct agroecological zone arguably reduces enterprise risk of exposure to extreme weather events, purchasing additional land does not necessarily improve enterprise carbon footprint (supplementary tables 1 and 2).

Our participatory workshops with the RRG resulted in co-designed and co-refined interventions, including those pertaining to income diversification. For example, the RRG indicated that purchasing additional farmland to diversify enterprise climate exposure (north-eastern Tasmania, 400 km from the existing beef cattle farm in north-western Tasmania) would require additional labour, costs of transporting cattle between regions, infrastructure on the new land,

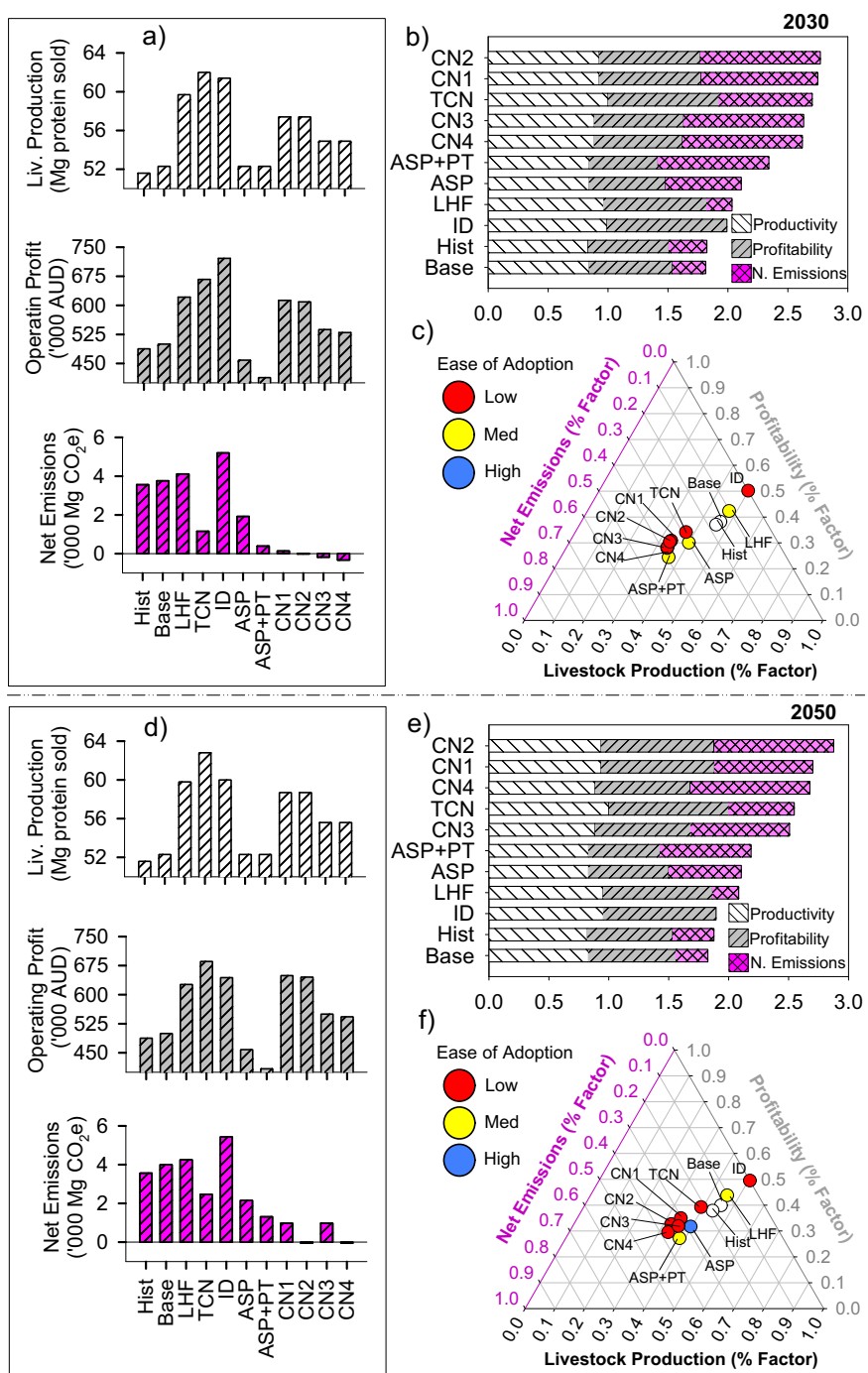

**Fig. 3 | Trade-offs between production, profit (pre-carbon tax), adoptability and net farm emissions for multiple thematic adaptations to beef farming systems in 2030 (a–c) and 2050 (d–f).** Bar charts on the left with dimensions shown on vertical axes (**a**, **d**) for livestock production (top), profit (centre) and net GHG emissions (bottom). These values were normalised by the greatest corresponding value for each metric (see 'Methods': Normalised Multidimensional Impact Assessments) in stacked horizontal bar charts (**b**, **e**) for multidimensional impact assessment. Normalised values for each metric (**b**, **c**, **e**, **f**) range from zero to one. Ternary plots (**c**, **f**) show normalised net emissions, profit and livestock production as well as ease of adoption attributed by the regional reference group. Hist:

historical climates; Base: existing farming system under future climates; LHF: low-hanging fruit package, TCN towards carbon neutral package, ID income diversification, Asp *Asparagopsis taxiformis* as a feed supplement, Asp + PT (Asp + planting 50 ha trees), TFCE adopting livestock genotypes with transformational feed conversion efficiency, CN1 carbon neutral package 1 (Asp + TFCE + planting 50 ha trees), CN2 carbon neutral package 2 (Asp + TFCE + 55 ha trees 2030 and 110 ha trees 2050), CN3 carbon neutral package 3 (Asp + renovating pastures with lucerne + planting 50 ha trees), CN4 carbon neutral package 4 (Asp + renovating pastures with lucerne + 55 ha trees 2030 and 110 ha trees 2050).

and higher management coordination across regions. As such, the RRG opined that this intervention would be difficult to sustainably pursue over the long-term (Fig. 1a, b). The RRG contended that establishment of an irrigated grapevine enterprise would require specialist input, given disparate skillsets and knowledge requirements

for managing vineyards compared with livestock production (Fig. 2a, b), including the need for micrometeorological and soil data surveys to identify suitable locations for the vineyard on farm. The RRG suggested that installing wind turbines would require proximity with three-phase powerlines (to feed electricity generated into the main

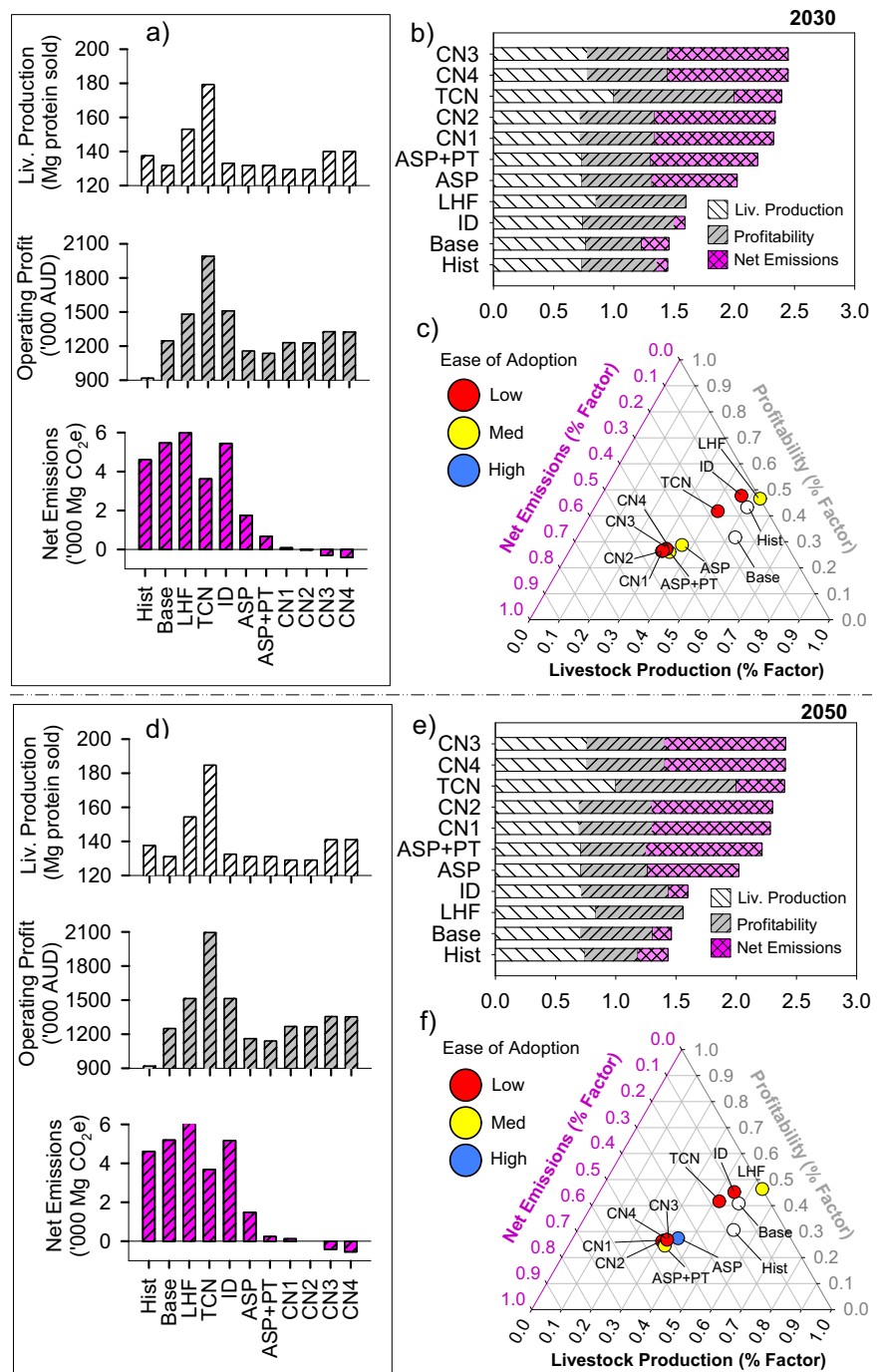

**Fig. 4 | Trade-offs between production, profit (pre-carbon tax), adoptability and net farm emissions for multiple thematic adaptations to sheep farming systems in 2030 (a–c) and 2050 (d–f).** Bar charts on the left with dimensions shown on vertical axes (**a**, **d**) for livestock production (top), profit (centre) and net GHG emissions (bottom). These values were normalised by the greatest corresponding value for each metric (see Methods: Normalised Multidimensional Impact Assessments) in stacked horizontal bar charts (**b**, **e**) for multidimensional impact assessment. Normalised values for each metric (**b**, **c**, **e**, **f**) range from zero to one. Ternary plots (**c**, **f**) show normalised net emissions, profit and livestock production as well as ease of adoption attributed by the regional reference group. Hist:

historical climates, Base existing farming system under future climates, LHF low-hanging fruit package, TCN towards carbon neutral package, ID income diversification, Asp *Asparagopsis taxiformis* as a feed supplement, TFCE adopting livestock genotypes with transformational feed conversion efficiency, Asp + PT (Asp + planting 200 ha trees), CN1 carbon neutral package 1 (Asp + TFCE + planting 200 ha trees), CN2 carbon neutral package 2 (Asp + TFCE + 220 ha trees), CN3: carbon neutral package 3 (Asp + renovating pastures with lucerne + planting 200 ha trees), CN4 carbon neutral package 4 (Asp+ renovating pastures with lucerne + 220 ha trees).

grid) as well as consistent and high prevailing winds (e.g. coastal regions). We suggest that interventions enabling climatic or enterprise diversification could be generically adapted to any production system or agroecological zone and are very much worthy of further interdisciplinary investigation.

## Contextualised adaptation-mitigation bundles: stacking interventions

We next co-designed and stacked together contextualised intervention bundles, each group based on synergies of intent (Figs. 3 and 4). Simple, immediately actionable, and relatively reversible changes to

the farm systems were stacked into a 'Low Hanging Fruit' (LHF) theme. Once operationalised, the LHF bundle improved annual productivity (15–16%) and increased profit (19–25%), but also increased GHG emissions by 6–18% compared with the baseline scenarios under future climates.

A 'Towards Carbon Neutral' (TCN) package was co-designed with the intent of improving productivity and reducing year-on-year GHG emissions. This bundle of interventions combined the LHF package with mitigation interventions ($CH_4$ inhibition vaccine, planting trees and renovating pastures with deep-rooted legumes). The TCN package respectively increased livestock productivity by 18–20% (beef farm) or 36–40% (sheep farm) under future climates (supplementary tables 6–9). Despite costs associated with buying land and planting trees and theoretical $CH_4$ vaccine inoculation (supplementary table 10), biophysical changes realised from pasture renovation in the TCN package increased profits by 33–37% and 60–68% for the beef and sheep farms, respectively. Overall, the TCN package reduced net GHG emissions by 37–69% for the beef farm (Fig. 3) and 29–34% for the sheep farm (Fig. 4), diluting emission intensities by 30–50% (supplementary tables 6–9). While TCN was one of the most prospective intervention bundle in terms of profit, production and GHG emissions (Figs. 3c, f, 4c, f), practical barriers to implementation (planting trees) and lack of commercial readiness ($CH_4$ inhibitor vaccine) reduced overall adoptability of this intervention.

Multiple combinations of stacked interventions facilitated profitable transitioning of farm systems to net-zero emissions (Figs. 3a, d, 4a, d). The four carbon neutral packages (CN1-CN4) were co-designed with consideration of various tree planting area, adoption (or not) of livestock genotypes with transformational gains in FCE (TFCE) and/or renovation of pastures with the deep-rooted perennial legume, lucerne. For the beef farm, feed supplementation with *Asparagopsis*, planting trees and adoption of genotypes with TFCE were most prospective (CN1 and CN2), facilitating not only carbon neutrality but also increasing productivity by 13% and profit by 30% in 2050 (Fig. 3). For the sheep farm, improvements in production and profit associated with carbon neutrality were most likely with stacking of *Asparagopsis* feed supplementation, planting trees and renovating pastures with lucerne, with the CN3 and CN4 increasing production and profit by 8% relative to their respective baselines (Fig. 4).

### Costs of transitioning to net-zero emissions under future climates

We next assessed implications of a hypothetical regulatory scenario where annual gains in net GHG emissions above net-zero were taxed, and additional net GHG below net-zero were credited. For consistency, we adopted a carbon tax of $80 Mg $CO_2$e$^{-1}$ and a carbon credit spot price of $28 Mg $CO_2$e$^{-1}$ for all analyses (see Methods). In the absence of intervention (continuing business as usual), carbon taxes were shown to reduce farm profits by 64% and 33% for the beef and sheep farms, respectively (Fig. 5). While use of *Asparagopsis* as a feed supplement decreased profit by 7-8%, post-carbon tax profit was significantly greater compared with the baseline (Fig. 5c, d, 5g, h). When feed supplementation with *Asparagopsis* was stacked with purchasing an additional farmland that was planted with trees (ASP + PT), a further 38-87% net GHG emissions were counterbalanced in the beef and sheep farm (Fig. 5a, b, 5e, f). Relative to the baseline farm in which all residual GHG emissions were taxed, ASP + PT improved profits by 53–68% (Fig. 5c, d) and 25–34% (Fig. 5g, h) for the beef and sheep farms in 2030 and 2050 climates, respectively.

While trade-offs in implementing CN packages were raised by the RRG (supplementary table 11), our results demonstrate that adoption of TCN practices may increase pre-carbon tax profit and significantly reduce GHG emissions. Relative to the baseline farm system in 2050, if all net GHG emissions were subject to carbon taxes, profit was at least three times higher for the beef farm and

1.5 times higher for the sheep farm. When the CN packages stacked TFCE (CN1 and CN2) or lucerne in the pasture swards (CN3 and CN4) with ASP + PT to mitigate GHG emissions, the cost of carbon tax was further reduced or, for cases where net-negative GHG emissions were realised (Fig. 5), contributed revenue through sale of carbon credits. Under 2030 climates, both farms showed residual net emissions after implementing TFCE (CN1) of -100 Mg $CO_2$e while CN2-CN4 scenarios were carbon negative (Fig. 5a, e). For the beef farm, there was little difference in net GHG emissions after implementing TFCE (CN1) and lucerne in the pasture sward (CN3), both with residual GHG emissions of -1000 Mg $CO_2$e in 2050 climates (Fig. 5b). Profits after carbon tax on residual GHG emissions were greater for the CN1 package compared with the CN3 package (Fig. 5c, d), and were more than $500 K higher than the baseline farm. Additional land for tree plantings was required for CN1 on the beef farm (2030 and 2050) and CN3 packages (2050) for the CN2 and CN4 packages to be net-zero GHG, respectively (Fig. 5a, b). For the sheep farm, the lucerne CN3 package achieved net-zero, with net sequestration of 300-400 Mg $CO_2$e (Fig. 5e, f) and pre-carbon tax profit of $1327–1366 K in 2030 and 2050 (Fig. 5g, h), increasing slightly if surplus carbon credits were sold (Fig. 5g, h).

## Discussion

We invoked participatory research encompassing nascent science with a transdisciplinary lens, allowing benefits in one dimension (e.g. GHG emissions mitigation) to be holistically quantified against trade-offs in other dimensions, such as food security, prosperity and environmental stewardship. While such assessments are more difficult to operationalise cf. reductionist studies due concerted coordination and co-learning required between parties, participatory approaches are arguably more amendable to impact, because they elucidate enablers and inhibitors of behavioural change[20]. Use of people-centric design in this way meant engaging end-users to develop fit-for-purpose interventions and thematic innovation bundles to adapt to future climates and/or mitigate GHG emissions[20,56,57]. This modus operandi allowed us to build end-user trust, credibility and legitimacy in methods invoked, affording researchers with the opportunity to validate model outputs while focusing on contemporary demand-driven problems[58]. While landscape-level assessments offer macro insights on emissions mitigation across farms, our purpose here was to co-design farm-level pathways for reducing GHG emissions and adapting to the changing climate in consultation with regional practitioners. This approach enabled refinement and validation of farming systems assumptions and, through iteration, allowed practitioners to gain confidence in methods we invoked. Indeed, the namesake of our study (costs of transitioning to net-zero emissions) was put to us by the RRG, evidencing the demand-driven nature of this work[27,59–61].

We thus emphasise that transitioning agricultural enterprises to net-zero emissions may result in different outcomes compared with transitioning farming landscapes to net-zero emissions. Adoption of the LHF intervention at the landscape scale may result in greater food security, assuming benefits at the farm level were realised *ceteris paribus* at the landscape level. Increased livestock supply on the market may reduce local prices due to trade-offs between supply and demand[62]. Implications of the TCN intervention at the farm scale may have similar implications for food security and market prices if adopted at the landscape level, although the latter may also influence carbon prices. For example, assuming widespread concurrent enrolment in carbon markets with adoption of the TCN intervention, available carbon credits on the market would be expected to fall, which would increase carbon prices. This may prohibit market entry for new practitioners and favour those with greater purchasing power and/or access to financial capital.

Adoption of renewable energy *en masse*—such as wind turbines and/or agrivoltaics—is unlikely due to geographical prerequisites for

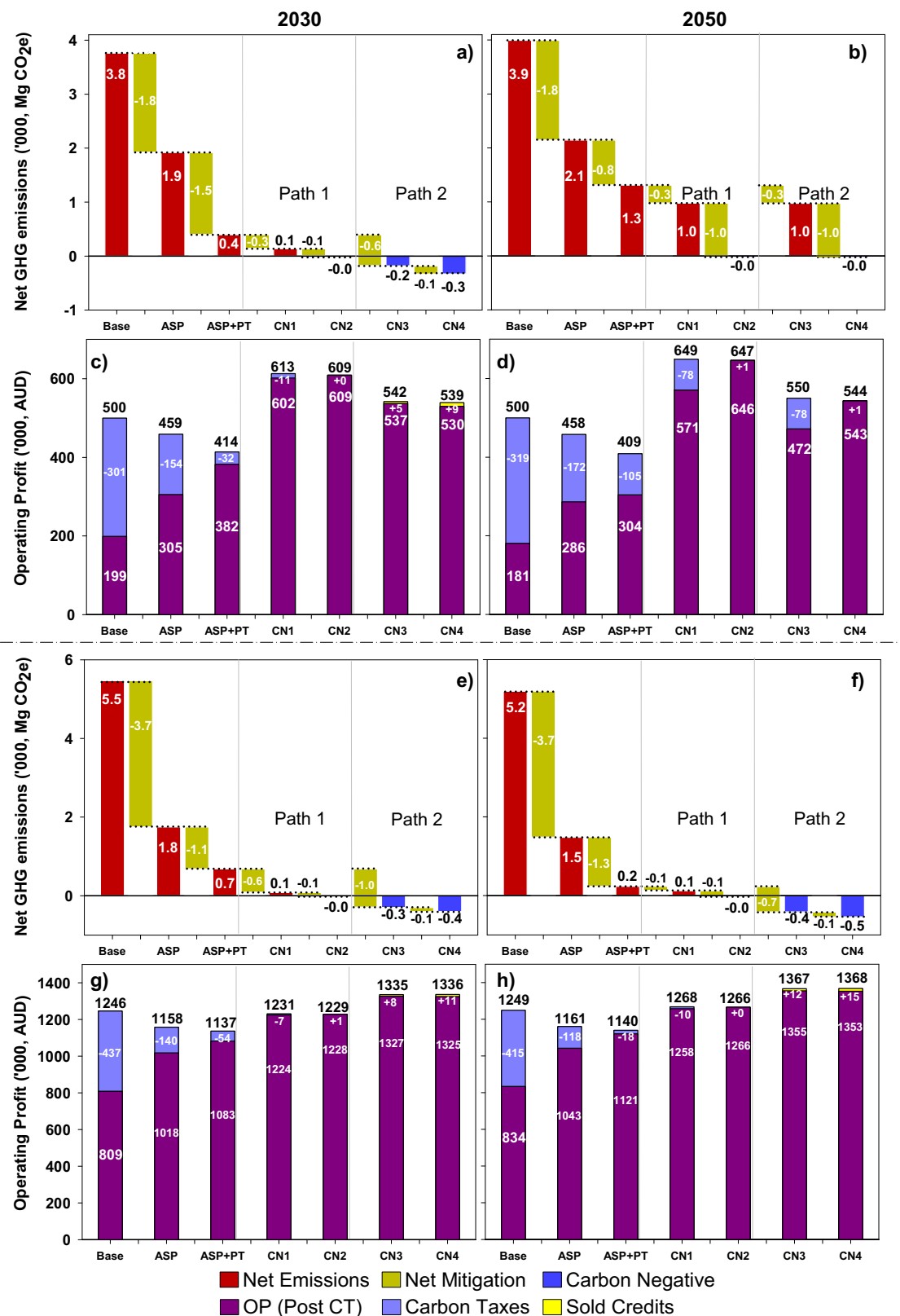

establishment, including consistent prevailing winds (for wind turbines) or high annual sunshine hours (for agrivoltaics), as well as proximity to three-phrase powerlines. In the same vein, the transformative interventions we examined are unlikely to be adopted at the landscape scale over a short period. Evrett Rogers famous 1962 theorem of the diffusion of innovation suggests that spatiotemporal adoption tends to be normally distributed, first with a few pioneering innovators, followed by the early adopters, then the early majority, the late majority and finally the laggards[63]. Stacking or bundling of interventions in the TCN, Income Diversification and Transformative interventions would be arguably more difficult to realise – evidenced by our results in Figs. 1 and 2—implying lower rates of adoption given additional knowledge, labour and practical requirements to successfully implement, refine and benefit from such interventions. Landscape

**Fig. 5 | Costs of two alternative pathways to net-zero GHG emissions for beef and sheep farming systems in 2030 and 2050.** Bars show pathways to net-zero emissions (red, dark yellow and blue) with associated carbon taxes (light blue), post-carbon tax profit (purple), and income from selling carbon credits (yellow) across climate horizons and thematic adaptations for the beef farm (**a**–**d**) and sheep farm (**e**–**h**). Stacked bars (**c**, **d**, **g**, **h**) represent operating profit before deductions from carbon taxes ($80 Mg CO$_2$e$^{-1}$) or income from carbon credits ($28 Mg CO$_2$e$^{-1}$; see Methods). Pathways 1 and 2 reflect net-zero farming systems attained by improving animal genetics (CN1 and CN2) or renovating pasture swards with lucerne (CN3 and CN4). Base: status quo performance in 2030 or 2050; ASP A. taxiformis as livestock feed supplement, Asp+PT A. taxiformis + planting trees (50 ha), CN1 carbon neutral package 1 [A. taxiformis + planting trees, 50 ha (beef farm) or 200 ha (sheep farm) + transformational feed conversion efficiency], CN2 carbon neutral package 2 [A. taxiformis + planting trees, 55 ha in 2030 and 110 ha in 2050 (beef farm) or 220 ha (sheep farm) + transformational feed conversion efficiency], CN3 carbon neutral package 3 [A. taxiformis + planting trees, 50 ha (beef farm) or 200 ha (sheep farm) + Lucerne], CN4 carbon neutral package 4 [A. taxiformis + planting trees, 55 ha in 2030 and 110 ha in 2050 (beef farm) or 220 ha (sheep farm) + Lucerne], OP operating profit, CT carbon tax.

level implications pertaining to adoption of interventions examined here would also require assessment of other socio-economic factors that could be ascertained using approaches such as agent-based modelling to account for social interactions, together with the influence of economic, regulatory and environmental drivers of land use[64].

Our study suggests that interventions for reducing GHG whilst maintaining or increasing profit under future climates are available, but depend on production system and agroecological context. Serendipitously, the option with lowest social licence—continuing business as usual and taxing net farm emissions—was also the most costly (Fig. 5g, h). Some scholars perceive use of carbon credits to offset GHG emissions (as opposed to insetting through carbon removal or reduction of GHG) as greenwashing, claiming that offsetting justifies lack of action to reduce GHG emissions and is conducive to double-counting. Even so, some GHG emissions are difficult to avoid and will require carbon offsets if the business aspires to net-zero emissions. Provided offset credits are (1) equivalent to the type and duration of the GHG source they are designed to counterbalance, (2) transparent, (3) legitimate, (4) certified, (5) measurable and (6) additional, income from offset purchasers can arguably finance carbon removal actions, such as planting trees or instigating practices for improving SOC[65].

Carbon taxation is a crucial tool for positive climate action designed to increase public acceptance by compensating low-income households and providing funding for climate projects[13]. Carbon taxes thus reflect the social cost of carbon (SCC), which monetise damages caused by each incremental tonne of CO$_2$e emitted to long-term environmental effects, such as sea-level rise, extreme weather events and agricultural yield losses[16]. In contrast to SCC, voluntary carbon markets reward proponents for GHG mitigation via carbon credits, promoting private investment and innovation. The lack of voluntary market mandates has resulted in wide variation in the integrity and legitimacy underpinning credit quality and climate impact[65]. Here, we adopted a hybrid approach including both carbon taxes and carbon credits from voluntary markets. While hybrid approaches discourage greenwashing[65], disparities between the SCC and voluntary carbon market spot prices risk undervaluing emissions reductions and removals[16]. Addressing these challenges through policy alignment and more robust regulation will be key to ensuring equity in attributing carbon cost and maximising climate benefits.

We found that stacking together interventions improved pasture growth and soil carbon sequestration, and adopting superior animal genotypes with greater liveweight gain for the same/less feed intake (FCE, TFCE), along with planting small proportions of farms with trees went considerable way towards negating farm emissions. When interventions to reduce GHG emissions instigated a productivity co-benefit—such as improved metabolisable energy per unit area, or shade and shelter via planting of trees, both carbon neutrality as well as improved profit were possible under future climates.

Taxes for net farm GHG were reduced as additional interventions were combined, especially when such interventions catalysed improved animal performance (CN packages, Fig. 5). This implies a need for producers to (1) adapt to changing climatic and economic circumstances, (2) reduce GHG and improve carbon sequestration. Depending on cost, purchasing additional land with the explicit objective of planting trees to offset livestock emissions and feeding a CH$_4$ inhibitor such as *Asparagopsis* in combination with TFCE (CN1 and CN2) or renovating pastures with lucerne (CN3 and CN4) showed promise for improving profit while also achieving carbon neutrality. As the need for non-agricultural industries to also offset their GHG emissions increases, the price of arable land is likely to increase in line with public pressure to maintain or improve institutional carbon removals[66]. As a corollary, carbon insetting (practices to reduce GHG within the enterprise) may become more profitable for some land managers, rather than seeking to purchase new land to conduct carbon sequestration activities[67].

As system flexibility determines management agility, tactical and strategic farm management planning in response to market volatility or climate change should encapsulate new and available technologies, together with consumer demand[62]. Carbon projects are typically underpinned by fixed 'permanence' obligations of 25–100 years[68,69], where carbon sequestered must be prevented from re-entering the atmosphere for at least the permanence period. While permanence refers to the longevity of the carbon sequestration, 'additionality' ensures that GHG emissions reduction or removal would not have occurred without incentive provided by carbon credits[70,71]. As such, obligations under carbon offsetting programs may differ from those of insetting. For example, there are several biomass carbon removal and storage (BiCRS) methods that result in permanent or geologic sequestration[72], offering greater durability than temporary carbon storage achieved through tree planting, although perhaps less biodiversity benefits. BiCRS combines the natural ability of plants to convert CO$_2$ into biomass with human engineering to store the biomass derivatives thereof, such as biochar[73–76], in a manner that prevents carbon from re-entering the atmosphere. For the global livestock sector to achieve net-zero emissions, permanent removals will likely be necessary, suggesting that BiCRS would be a prospective intervention worthy of further investigation.

The time with which GHGs are removed or prevented from entering the atmosphere depends on the modus operandi of the intervention and GHG in question. Interventions that mitigate enteric CH$_4$ emissions—such as feed supplementation with *A. taxiformis*—permanently prevent CH$_4$ that would have otherwise entered the atmosphere, assuming other aspects are *ceteris paribus*. In the same vein, adopting animals with genetics that afford greater feed-conversion efficiency can permanently avoid enteric CH$_4$, assuming animals with greater FCE are sold at the same liveweight and earlier than the *ceteris paribus* system. Interventions that sequester additional carbon in soils and/or vegetation, such as renovation of pastures with lucerne or planting trees are however characterised by diminishing longitudinal carbon sequestration[77] (supplementary fig. 1). Assuming other aspects remain unchanged, annual carbon emissions would be expected to vary around some constant value, while carbon removals would diminish as trees approach maturity, making prospects of attaining net-zero increasingly difficult with the passage of time. This could be countered in many ways, for example via agroforestry, where portions of farm area are sequentially sown then harvested for timber as trees approach maturity. Provided carbon in harvested timber was not permitted to re-enter the atmosphere (e.g. use of timber in

construction materials), longitudinal carbon removals of the farm business would no longer plateau, as some plantations would always be approaching, or in, periods of peak growth. We suggest that farming systems approaches that afford such asynchronous temporal carbon sequestration, together with practices that reduce $CH_4$ emissions (such as *A. taxiformis* feed supplements) in concert with practices that avoid $CH_4$ emissions (improving animal growth rates and earlier sales) or $CO_2$ (renewable energy on farm), will be increasingly called for in future, particularly if farms are mandated to reduce GHG emissions.

A consideration relating to permanence of carbon sequestration via planting trees is risk of wildfire. Warmer, drier conditions borne by climate change have increased fire propensity and seasonal duration, significantly increasing areas burnt over the last decade[78]. To ensure that carbon sequestered in vegetation used for insetting is prevented from re-entering the atmosphere by fire, a buffer pool may be necessary. This may include planting trees across multiple locations and setting aside some carbon credits as insurance against potential losses due to wildfires. Buffer pools act as safeguards, ensuring that each carbon credit delivers the intended $CO_2$ removal or avoidance, in the event that some carbon stocks are lost[79]. However, this strategy may increase costs associated with insetting, potentially undermining their economic viability, particularly if farm area is small or if access to multiple locations is constrained[80].

The *Asparagopsis taxiformis* feed supplement intervention was examined through the lens of a transformative intervention enabling deep cuts in enteric $CH_4$ emissions, with the 80% enteric $CH_4$ mitigation value calibrated based on empirical evidence from several peer-reviewed studies[41–44,81,82]. Use of *A. taxiformis* in this way showed significant promise at the farm scale, decreasing net farm GHG emissions by 46-72%. Although some in vitro studies report greater $CH_4$ inhibition than we assumed (up to 99% enteric $CH_4$ reduction[41,42,82]), enteric $CH_4$ mitigation in grazing systems may be more modest due to animal access to feed supplements, foraging behaviours, seasonal variation in forage quality, diet composition and other enterprise constraints in situ[20,81]. As such, to determine the sensitivity of net farm GHG emissions and enterprise profit associated with enteric $CH_4$ mitigation, we conducted sensitivity analyses with enteric $CH_4$ mitigation ranging from 10% to 99%[44,82] (Supplementary Figs. 3 and 4). For the beef farm, net emissions ranged from 1484–1709 Mg $CO_2$e year$^{-1}$ with a 99% $CH_4$ reduction, to 3533–3761 Mg $CO_2$e year$^{-1}$ for 10% $CH_4$ reduction. After inclusion of carbon credits ($28 Mg $CO_2$e$^{-1}$), such $CH_4$ reduction yielded profits of $321–340 K and $157–176 K, respectively (supplementary fig. 3 and 5). For the sheep farm, net GHG varied from 596-871 Mg $CO_2$e year$^{-1}$ (99% $CH_4$ reduction) to 4726–4997 Mg $CO_2$e year$^{-1}$ (10% $CH_4$ reduction), with post-carbon tax profit ranging between $1088–1113 K and $758–783 K (supplementary fig. 4 and 6). This analysis demonstrates that adaptation/mitigation bundles are sensitive to the quantum of $CH_4$ reduction, and while we acknowledge that enteric $CH_4$ emissions in any production system will vary between animals, seasons and farms, these results highlight farm-level mitigation quanta should some transformative intervention for reducing enteric $CH_4$ be realised. If enteric $CH_4$ mitigation were lower than assumed here, further measures would be required to negate residual GHG emissions. One way for this may be through additional tree plantations. If C sequestration were improved by 20%, enteric $CH_4$ mitigation required to realise net-zero status would fall from 80% to 60% for the beef farm (CN2 and CN4) and from 80% to 70% for the sheep farm (CN3 and CN4). If tree C sequestration were 20% lower, either enteric $CH_4$ reduction would need to be greater than 80% or further measures would be necessary to achieve carbon neutrality. Taken together, these results demonstrate that sustained maintenance of net-zero emissions for livestock businesses will be challenging. Any aspiration for GHG abatement would need to be conducted by combining a range of technologies, practices and infrastructure for carbon reduction and removals, given that sequestration in soils and vegetation tends to diminish over time.

Similar to any GHG mitigation intervention, societal impact of feed supplementation with *Asparagopsis* will be dictated by manifold economic, social, environmental, institutional and psychological factors, such as ease of implementation, market supply and price, social licence, government regulation and animal/human health and welfare implications. Forecasts suggest a future Australian 1.5B seaweed production industry, creating up to 9,000 jobs and 10% national GHG emissions reduction year$^{-1}$ by 2040, which would comprise a substantial contribution towards the UN Sustainable Development Goals[83]. Nonetheless, many challenges must be resolved if such forecasts are to eventuate. For example, certain seaweed species are invasive; such species may detrimentally impact native species in marine environments should they escape their captive environments. Other authors contend that bromoform (the compound in *Asparagopsis* which inhibits enteric $CH_4$) may have adverse connotations for animal health or ozone depletion[81,82]. Such concerns highlight the need for further investment, research and development to carefully elucidate the benefits and risks associated with large-scale adoption of seaweed as a livestock feed supplement[81,84].

The expert group of practitioners involved in this research stressed the importance of integrating legumes within existing grass pastures (CN3 and CN4 packages). Sturludóttir, et al.[85] further demonstrated that mixed grass-legume swards tend to have higher herbage yield, dry matter digestibility (DMD) and crude protein (CP). The N yield advantage from grass-legume mixtures supported by symbiotic $N_2$ fixation[86], given the close linkage between C and N cycling in grazing systems[87], may also improve SOC accrual[88–90]. However, superfluous legume sward composition can have animal welfare implications, such as bloat and even death[91]. Effects of pasture renovation with lucerne modelled here account for impacts of root depth, soil moisture, SOC stocks, sward crude protein and digestibility, feed intake, and nitrogenous fertiliser, among other factors[92] (supplementary table 12). Impacts of stocking rates (urine and faecal loading per area), pasture growth (mineral N use) and nitrogenous fertiliser on $N_2O$ emissions were accounted for using equations prescribed under the Australian National GHG Inventory[92]. While we note that the Australian National GHG Inventory and IPCC exclude $N_2O$ associated with Biological Nitrogen Fixation (BNF)[92,93], we acknowledge that nascent empirical experimentation demonstrates that lucerne agroecosystems can propagate significant $N_2O$[94] due to BNF, among other factors[95]. While inclusion of such empirical data is not within the purview of the present study, we contend that impacts of BNF on $N_2O$ at scale is worthy of deeper analysis in future, provided robust empirical datasets across a range of agroecosystems and management options are available.

We made several discoveries relating to prospective pathways to net-zero GHG emissions across farm enterprises. We revealed that singular interventions realised limited concurrent improvements in productivity, profitability and GHG mitigation. Under future climates, strategies such as adoption of low-emission livestock feed supplements (*Asparagopsis*) and planting of appropriate tree genotypes were most effective in reducing emissions, albeit came with higher economic costs. Diversifying farming enterprises spatially by purchasing land in diversified climatic zones, along with adoption of animal genotypes with transformative feed-conversion efficiency (TFCE) promised substantial benefits in productivity and profitability. Establishing wind farms provided significant financial benefits, enabling farms with certain conditions (suitable wind speeds, etc) to generate renewable energy while reallocating resources to climate-smart practices (Figs. 1, 2). While continuing business-as-usual (BAU) and paying carbon taxes was the most adoptable intervention we assessed, it was also the most expensive strategy, and as such should be discouraged. We underscore the triple bottom-line potential associated with appropriately contextualised bundling of interventions, particularly when they target productivity, mitigation of enteric $CH_4$

and carbon removals. Interventions such as planting small areas of trees on farm, renovating grass-based pastures with deep-rooted legumes, and adopting high FCE animal genotypes were shown to not only realise net-zero GHG emissions, but also improve productivity. In all cases, interventions that realised productivity gains were most conducive to propitious outcomes. We contend that purported innovations are more likely to be transformational if aspiring developers consider multiple sustainability indicators, including environmental stewardship, food security, market access, social licence to operate and the changing climate.

## Methods

### Study overview

Our research complies with the Australian Code for Responsible Conduct of Research, including all relevant ethical regulations. The University of Tasmania Human Research Ethics Committee (Tasmania) approved this research (Ethics Reference Number H0017705). Farming systems were co-designed using an integrated cross-disciplinary framework[31]. Through a series of workshops, a Regional Reference Group (RRG) of expert industry practitioners co-designed biophysical, environmental, and economic interventions (Fig. 6). Co-designed interventions (singular and combined technologies and practices) were examined using a social science lens, including assessment of adoption barriers, social license to operate, and new skills required for adoption. This co-design framework was used to quantify and stack individual adaptations on top of the baseline farm system, before each intervention was iteratively refined with the RRG over several cycles[31].

To showcase this approach, farm systems across two regions of Tasmania, Australia, were selected to be modelled: a sheep production system (hereafter 'sheep farm') in the low rainfall zone in central Tasmania and a beef production system (hereafter 'beef farm') in the relatively high rainfall zone of northwestern Tasmania. Individual interventions aimed at income diversification and/or transformational were suggested by the RRG. Transformational adaptations were regarded as longer-term, higher-risk interventions involving some degree of irreversibility. These adaptations were stacked together in a mutually synergistic way based on commonality of intended outcomes. Incremental adaptations were defined as those that do not significantly alter the *status quo*. Income diversification interventions were designed such that new income streams would be derived that were independent of rainfall in the location of the current farm system, as rainfall was perceived to be a climatic index that would change under future climates, and these livestock systems relied primarily on pasture produced from rainfall. Income diversification was thus classified as those interventions affording either climatic diversification or enterprise diversification. Manifold approaches were used to simulate farm systems (Fig. 6).

To prevent dangerous climate change, the international scientific community has indicated that GHG emissions must be net-zero by 2050[96,97]. We thus adopted this temporal scope to contextualise farm interventions with national and international climate mitigation policies. Future climate projections[58] accounted for increased frequency and severity of extreme weather events. The whole-farm model GrassGro® (version 3.3.10[98]) used to simulate daily pasture and livestock production and was driven by historical and future climate data. Soil organic carbon (SOC) sequestration was simulated using RothC model (version 26.3 in Microsoft Excel format[99]) with GrassGro outputs, while FullCAM (version 4.1.6[100]) was invoked to estimate tree carbon sequestration. Net farm GHG emissions were calculated using SB-GAF version 1.4[92] using outputs from GrassGro®, RothC and FullCAM. The @Risk model[101] was used to account for market volatility using a partial budgeting approach (i.e. earnings before interest and taxes, herein referred to as profit) to compare the costs and income of incremental, income diversification and

transformational adaptation/mitigation interventions by each farm business.

### Historical and future climates

The beef farm was located at Stanley in the cool temperate zone of north-western Tasmania (40° 43' 41"S 145° 15' 43"E), while the sheep farm was located in the Midlands, west of Campbell Town (41°56'30"S 147°25'02"E). Stanley and Campbell Town have long-term mean and standard deviation annual rainfall of 807 ± 139 mm and 499 ± 103 mm, respectively, with average daily temperatures of 16.5 °C and 16.7 °C in January (summer) and 9.1 °C and 6.5 °C in July (winter), respectively (supplementary fig. 7). Daily historical climate data for the baseline period of 1 January 1980 to 31 December 2018 was sourced from SILO meteorological archives (http://www.longpaddock.qld.au/silo). Data from SILO was used to generate future climate realisations following Harrison et al.[58] using a stochastic approach to account for changes in climatic extremes, including heatwaves, droughts and extreme rainfall events[58]. Future climate projections were downscaled from global circulation models (GCMs) to regional and farm-scale[102]. To generate future climate data, (1) we estimated mean changes in future climates projected for a region based on ensembles of global climate models (GCMs), (2) accounted for historical climate characteristics (obviated by raw GCM data) and (3) generated climatic projections with increased variability. Future climate projections for 2030 and 2050 were developed using monthly regional climate scaling factors (Supplementary Table 13) from GCMs provided by Harris et al.[102] based on Representative Concentration Pathway (RCP) 8.5. Atmospheric $CO_2$ concentrations were set at 350 ppm, 450 ppm and 530 ppm for the historical, 2030 and 2050 climate scenarios, respectively[103]. Projected climate changes are derived from the Coupled Model Intercomparison Project Phase 5 (CMIP5), which use the IPCC reference period of 1986 to 2005 (referred to as the 'Historical climate')[103]. We selected this period as our baseline, as future climate projections are made relative to this window. Monthly changes are provided for variables such as mean, maximum, and minimum temperature, rainfall, humidity, solar radiation, wind speed, and evapotranspiration. These projections focus on two 20-year periods, centred on 2030 (2022 to 2041) and 2050 climates (2042 to 2061).

### People-centred design and the regional reference group (RRG)

We ground-truthed assumptions and model inputs using an iterative process with a regional reference group (RRG) of livestock industry specialists comprising farmers and consultants. Model outputs refined in consultation with the RRG included pasture growth rates, stocking rates, liveweight and wool production, supplementary feeding, costs, income, depreciation, net cash flows and wealth. When RRG consensus was reached for each historical period, outputs from biophysical and economic models were run for 26-year periods (we discarded the first six years of data to allow for model initialisation), each period centred on 2030 or 2050. Over several workshops, we gleaned RRG thinking and feedback on incremental, systems and transformational adaptation and mitigation opportunities in light of qualified holistic impacts of climate change. Based on RRG recommendations, we explored individual adaptations to understand potential effects on productivity, profitability and offsetting of GHG emissions and acceptability within the industry. We also combined individual interventions into four adaptation/mitigation themes: 'Low Hanging Fruit', 'Towards Carbon Neutral', 'Income Diversification' and 'Carbon Neutral'; modelled results of these themes were compared with the baseline scenario (detailed Fig. 6 and Table 1). We refined model inputs based on RRG feedback on practicality and magnitude of variables simulated. This process (1) ensured that model results were realistic, (2) provided the research team with nascent knowledge relating to opportunities for adaptation and mitigation of climate change from expert practitioners, (3) engendered end-user confidence in the analytical process and

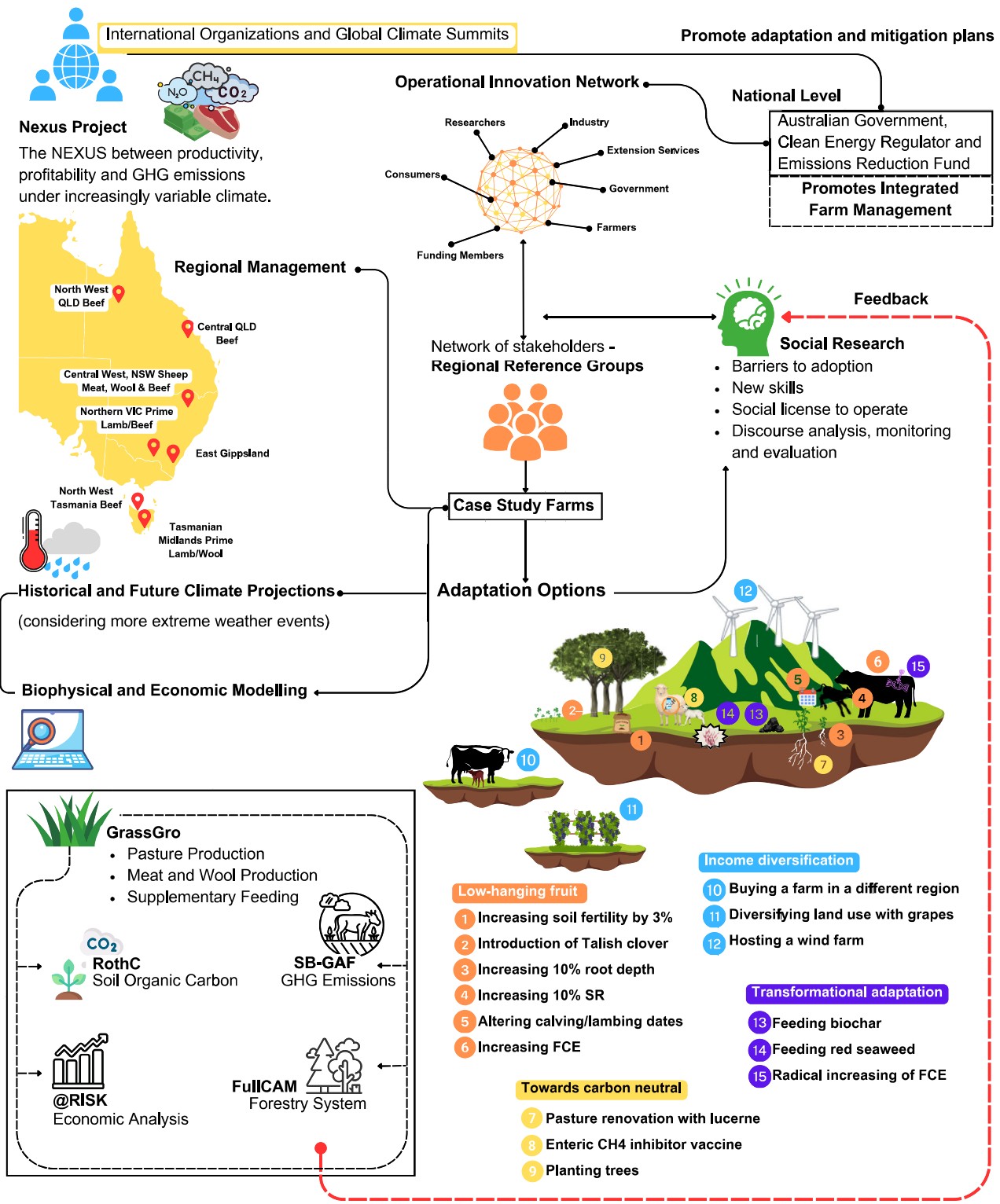

**Fig. 6 | Co-design framework for elucidating economic, environmental and social factors enabling or inhibiting adoption of adaptation/mitigation interventions under past and future climates.** Modelling and social research were iteratively refined with stakeholders to improve research rigour but also build trust through demand-driven, bottom-up research. Policy, economic, climatic, social and cultural factors were considered in the co-design of interventions for either reducing/removing GHG emissions, adapting to future climatic conditions, or both. Orange, light brown, blue and purple circles represent Low-Hanging Fruit (LHF), Towards Carbon Neutral (TCN), Income Diversification (ID) and Transformational adaptation-mitigation themes, respectively.

**Table 1 | Thematic adaptations co-designed with a Regional Reference Group (RRG)**

| Theme | Incremental, systemic and transformational adaptations stacked into holistic adaptation themes |
|---|---|
| LHF | - Altered lambing/calving dates to better match seasonal pasture supply |
|  | - Altered selling dates/SR/LW to better match seasonal pasture supply |
|  | - Adopting pasture species with 10% improvements in maximum root depth[123] |
|  | - Increasing soil fertility with SSP and N by 3% (Harrison et al.[60]; all paddocks except the native pastures for the sheep farm) |
|  | - Increasing FCE by 10% in 2030 and 15% in 2050, relative to baseline[45,61]. |
|  | - Introduction of Talish clover (*Trifolium tumens*) to a proportion of the sheep farm[124,125] |
|  | - Removing cattle from sheep farm and increasing rainfed pasture area to the two sheep flocks |
| TCN | - Strategic manipulation of livestock selling dates/SR/LW to better match seasonal pasture supply |
|  | - Pasture renovation with (and increased farm area of) lucerne pastures; following RRG guidance, renovation was conducted by seed broadcasting. |
|  | - Injecting animals with a vaccine to inhibit enteric $CH_4$ by 30%[24] |
|  | - Purchase 50 ha of land for the beef farm to establish a tree plantation of Tasmanian Blue Gums to offset livestock GHG emissions |
|  | - Thickening of 200 ha of existing nature pasture (non-grazed) land for sheep farm with environmental plantings (trees, shrubs and understory species endemic to the region) |
| ID | - Buying additional land parcels in a different agroclimatic region (by translocating cow calf systems to Gladstone, NE Tasmania to dedicate the current farm for backgrounding and finishing of weaners) |
|  | - Diversifying land use with grapes by repurposing 30 ha from the sheep farm to grow Pinot Noir and Chardonnay grapes (processed offsite and outside scope of the current project) |
|  | - Hosting a wind farm (by leasing land for 12 wind turbines to generate an extra income, no insetting of $CO_2$ from turbines to reduce on-farm GHG emissions, in line with the business model of the wind turbine company) |
| CN | - Red seaweed (*Asparagopsis taxiformis*) as a feed supplement to offset $CH_4$ by 80%[41–44,81,82] considered as aspirational target (see Sensitivity analysis in methods and supplementary table 12). |
|  | - Biochar as a feed supplement to increase in liveweight production by 5% for the beef farm and reduced enteric $CH_4$ fermentation in SB-GAF by 10% for both case studies. Effects of biochar on SOC by C enrichment in manure are described in supplementary fig. 9. |
|  | - Pasture renovation with (and increased farm area of) lucerne; following RRG guidance, renovation was conducted by seed broadcasting. |
|  | - Purchase 55 to 110 ha of land for the beef farm to establish a tree plantation of Tasmanian Blue Gums to offset livestock GHG emissions |
|  | - Thickening of 200 to 220 ha of existing nature pasture (non-grazed) land for sheep farm with environmental plantings (trees, shrubs and understory species endemic to the region) |
|  | - Transformational increase in FCE, to 20% in 2030 and 30% in 2050, relative to baseline[45,61] |

Each thematic adaptation comprised multiple stacked incremental adaptations suggested by the RRG; the extent to which each factor was varied from the baseline was derived from peer-reviewed literature. Further details are provided in Supplementary Tables 5, 12, 14 and 15.

*LHF* Low-Hanging Fruit, *TCN* Towards Carbon Neutral, including all incremental adaptations for LHF, *ID* Income Diversification, *CN* Carbon Neutral Package, *SR* Stocking Rate, *LW* Liveweight head-1, *FCE* Feed Conversion Efficiency, *RD* Rooting Depth, *SSP* Single Superphosphate fertiliser, *N* Nitrogen fertiliser.

results and (4), provided end-users with credible, legitimate and fit-for-purpose bundles of adaptation/mitigation interventions. Detailed information pertaining to baselines and each intervention is articulated below and in the supplementary information (supplementary tables 5, 12, 14, 15, supplementary fig. 8).

## Pasture and livestock production

The model GrassGro® enables simulation of ruminant grazing enterprises of southern Australia by combining biophysical (climate, soils, pastures and livestock), farm management (soil fertility, paddock size and layout, pasture grazing rotations, stocking rate and animal management) and economic data (gross margin). GrassGro® has been used to explore the effects of climate, soil, pasture, herd/flock management and adaptation for predicted climate change impacts on livestock productivity and profitability[104] in pasture-based industries across Australia[38,105], North America and Northern China[106,107]. GrassGro® computes soil moisture, pasture production, pasture quality (CP% and DMD%) on a daily basis for each pasture species, paddock and farm. Other variables calculated by the model include sward characteristics, pasture cover, pasture persistence, pasture availability, pasture intake, feed supplement requirements, liveweight change, and feed carry-over effects from year to year. We initialised and parameterised GrassGro® using baseline information collated from each case study farmer.

## High rainfall beef production system

The beef farm ran a self-replacing cow and calf operations on a land area of 569 ha. This enterprise comprised 367 mature cows calving in late

winter (1 Aug with 95% weaning rate, first calving at two years of age), assuming a typical replacement rate of around 20% each year (74 heifers). Home-bred non-replacement heifers and steers were sold at 25 months (1 Sep) at approx. 550 and 600 kg, respectively. An additional 115 of weaners were purchased at 6 months of age (1 Feb) at approx. 200 kg liveweight (LW) and were sold at 25 months (1 Sep) at approx. 600 kg LW. A group of 155 steers was also purchased at 16 months of age (1 Feb) at approx. 375 kg LW each year and sold at 28 months (31 Jan) at approx. 545 kg LW. Before being cast for age on 10 Feb, mature cows were retained for five lactations. Pasture species mainly comprised perennial ryegrass (*Lolium perenne* L.) and white clover (*Trifolium repens* L.), but also cocksfoot (*Dactylis glomerata* L.), subterranean clover (*Trifolium subterraneum* L.) and lucerne (*Medicago sativa*). According to the Northcote classification[108], the soil type defined in GrassGro was Uc2.3. Five percent of farm area (20 ha lucerne/ryegrass and 8 ha perennial ryegrass/cocksfoot/white clover pastures) was irrigated between 21 Nov and 31 Mar each year (20 mm each irrigation event on a 14-day interval) to replicate long-term average irrigation water applied. To either maintain LW (cows) or achieve target LWs (all other stock), production feeding rules were implemented in GrassGro using hay (DMD of 77% and CP of 20%). While all stock grazed rainfed pastures, home-bred steers were also given access to irrigated pastures throughout the year. Further information can be found in Supplementary Table 14.

## Low-rainfall sheep production system

The sheep farm ran a self-replacing Merino superfine wool, prime lamb and secondary, a beef cattle enterprise grazing 3170 ha and consisted

of 49% native grasslands, 48% rainfed developed pastures and 3% centre pivot irrigation (introduced grasses and legumes). A total of 4600 ha of native woodlands were also present on the farm that were not subjected to grazing. According to the Northcote classification[108], the soil type defined in GrassGro was Dy5.61. The modelled rainfed pastures were composed of pure stands of Phalaris (*Phalaris aquatica* L.), Phalaris-subterranean clover mixtures or native grasses ((Wallaby grass (*Austrodanthonia spp.*)) and Weeping grass (*Microlaena stipoides*) implemented in GrassGro. One paddock of lucerne was used for grazing and hay production and another paddock of dual-purpose wheat (*Triticum aestivum* L., modelled using GrassGro's annual ryegrass (*Lolium multiflorum* L.) to best reflect the growth pattern of wheat) was grazed for four months prior to grain production. The lucerne and wheat paddocks were irrigated from 1 Sep to 31 Mar with 18 mm of water per irrigation event to fill the soil profile to 95% of field capacity when soil water deficit reached 50%.

The sheep farm ran 24,750 animals, grouped in two flocks: a self-replacing Merino flock (SMF) and a prime lamb flock (PLF). The SMF comprised three groups: 5300 mature superfine Merino ewes, 7500 wethers and 5500 replacement ewes and wethers. The SMF ewes were first lambed at 2 years of age and retained for three lambings before entering the PLF for two more annual births before being cast at 7 years old (16 Dec). Before wethers were cast for age (14 Oct), the animals were retained for five years. Non-replacement wether lambs and ewes were sold 1 Feb. A total of 3,450 Merino ewes were mated with White Suffolk rams in the PLF; the 2950-lamb progeny were sold in mid-December at 27 kg LW. The sheep (except prime lambs) were all shorn on 20 Jul, clean fleece weight (CFW) was 3.3-4.1 kg with fibre diameters of 17.4–18.1 μm (variation in CFW and micron depended on stock class and age). Further details are provided on maintenance and production feeding rules, as well as grazing rotations in supplementary material (Supplementary Table 15). The beef cattle herd consisted of 340 mature cows and 60 replacement heifers per age group. Two-year-old mature cows calved (30 Aug) and were retained for eight years before being cast for age. After weaning date (1 Apr), steers (150 head) were sold at 18 months of age (28 Feb at ~ 460 kg LW) while non-replacement heifers (90 head) were sold at 200 kg LW. Further information can be found in Supplementary Table 15.

## Net farm greenhouse gas emissions

The Sheep Beef Greenhouse Accounting Framework (SB-GAF version 1.4[92]), which incorporates Intergovernmental Panel on Climate Change methodology and equations prescribed under the Australian National Greenhouse Gas Inventory, was used to calculate net farm GHG emissions. Use of outputs from biophysical models[58,104] as SB-GAF inputs has been previously shown to be robust for beef[109] and sheep enterprises[110]. Twenty-six-year seasonal mean data from GrassGro were used as input data for SB-GAF. To convert $CH_4$ and $N_2O$ into carbon dioxide equivalents ($CO_2e$), SB-GAF assumes 100-year global warming potentials ($GWP_{100}$) of 28 and 265, respectively. Greenhouse gas outputs were calculated as net farm emissions (Mg $CO_2e$ annum$^{-1}$) and emissions intensity (Mg $CO_2e$ Mg product$^{-1}$). Greenhouse gas emissions included $CH_4$ from livestock enteric fermentation and manure; $N_2O$ from nitrogenous (N) fertiliser, waste management, urinary and dung deposition and indirect N emissions via nitrate leaching and ammonia volatilisation; $CO_2$ from synthetic urea applications, electricity and diesel consumption, as well as $CO_2e$ pre-farm embedded emissions from fertiliser and supplementary feed. We exclude $N_2O$ emissions associated with leguminous symbiotic nitrogen fertilisation following approaches prescribed by the Australian National GHG Inventory and Intergovernmental Panel on Climate Change[92,93]. Annual electricity and diesel consumption are computed as a function of location, enterprise type, cultivation and machinery use, as well as livestock numbers and use of farm infrastructure. Allocation of GHG emissions between meat and wool was based on protein mass ratio

following Wiedemann et al.[111]. Net farm GHG emissions were based on GWP100 computed on an annual basis using SB-GAF as the sum of carbon sequestration in soils and vegetation with GHG emissions (sequestration being negative; $CO_2$, $CH_4$ and $N_2O$ emissions being positive). Annual net GHG emissions represent 20-year averages for each of the historical, 2030 and 2050 climate horizons (Supplementary Tables 1–4 and 6–9). Net-zero GHG emissions were defined as the point at which annual net GHG emissions averaged over each 20-year simulation equated to zero.

## Soil organic carbon in grazed pastures

The Rothamsted Carbon model (RothC; version 26.3 in Microsoft Excel format[99]) was used to simulate dynamic soil organic carbon (SOC). RothC has been used globally to model the impacts of climate and management on SOC stocks[112]. RothC simulations are driven by historical and projected monthly means of temperature, rainfall and pan evaporation (see Historical and future climate data). Monthly average GrassGro outputs were input into RothC including dung and litter. Root residue C inputs were derived from GrassGro outputs considering litter, allocation of net primary production between plant components, active root length density and proportion of root by layer (0–30 cm and 30–100 cm depth) and dung excreted by animals. Further details on the process invoked to translate GrassGro outputs into RothC can be found in the in the supplementary material. Soil types primarily consisted of clay loam Red Ferrosols on the beef farm[113], and Dermosols on the slopes adjacent to native vegetation and Vertosols on the river flats on the sheep farm[114]. Soil clay contents in the 0–30 cm and 30–100 cm layers were derived from the TERN-ANU Landscape Data Visualiser (https://maps.tern.org.au/#/) and historical SOC was sourced from regional sources[113]. RothC simulates C transfers between several soil organic matter pools, including decomposable plant material (DPM), resistant plant material (RPM), fast and slow microbial biomass (BIOF and BIOS), humified organic matter (HUM) and inert organic matter (IOM)[99]. RPM, HUM and IOM fractions were comparable to historical data for the three soil types across the two farms[113]. Allocations across SOC pools given by Hoyle et al.[115] for initial fractions of DPM, BIOF and BIOS were adopted here (1%, 2% and 0.2% of initial SOC stocks, respectively) and IOM fraction was similar to that reported by Falloon et al.[116]. Soil carbon decomposition rates at 30 cm were derived following Jenkinson and Coleman[117], except for the decomposition rate for RPM, which was set to 0.17 following Richards and Evans[100], similar to the 0.15 reported by Cotching[113], such that decomposition rates constants for DPM, RPM, BIO and HUM were 10, 0.17, 0.66 and 0.02, respectively. At 30–100 cm, decomposition rates were calculated following Jenkinson and Coleman[117]; all values were lower than those for 0–30 cm, reflecting lower decomposition rates at depth. Decomposition rates constants for DPM, RPM, BIO and HUM were 0.33, 0.01, 0.02 and 0.00, respectively. Further details are available in the supplementary material.

## Tree growth, carbon in vegetation and soil beneath tree canopies

We invoked FullCAM (version 4.1.6[100]) to simulate dynamic temporal tree growth, along with carbon sequestration in biomass and in soils beneath trees. FullCAM is used in the Australian National Carbon Accounting System and is driven using mean monthly temperature, rainfall and pan evaporation. Soil organic matter and carbon in FullCAM is simulated by RothC; all soil parameters were matched with those we used for RothC described above. FullCAM simulates C cycling between forest and soil components, including litter, surface and subsurface debris. We modelled planting of Tasmanian blue gum (*Eucalyptus globulus* L.) and 'environmental' plantings (combination of trees, understory and shrubs native to the region) for the beef and sheep farms, respectively. FullCAM simulations were run continuously from 2022 to 2062 by combining the climate data for the two future

time frames, as opposed to two individual simulations commencing 2022 and 2042. We modelled planting of shelter belts for the beef farm and woody thickening of pre-existing woody vegetation for the sheep farm. The simulated data on tree carbon sequestration was comparable to collected data from regions in southern Australia with precipitation levels exceeding 660 mm per year for the beef farm and ranging from 400 to 660 mm per year for the sheep farm as a function of tree age for species described by RRG (Supplementary Tables 16 and 17). Livestock grazing beneath trees (silvopasture) was not permissible following advice from the RRG that such farming systems would require additional knowledge and practical expertise to successfully operationalise (elsewhere however, silvopasture has been promulgated as a sustainable management practice, having benefits for livestock productivity via provision of shade and shelter, carbon sequestration, and biodiversity conservation[118,119]). We assumed that carbon sequestration exceeding net farm greenhouse gas (GHG) emissions would be sold, so the enterprise would obtain revenue from surplus carbon credits; at no point were trees harvested (further details are provided in the supplementary information).

## Economic analyses

In concert with GrassGro outputs, we used the @Risk Software[101] to simulate stochastic annual feed supply, livestock carrying capacity, supplementary feed requirements, commodity prices and farm incomes, following approaches outlined in previous studies[120]. Long-term wool, meat and livestock prices adjusted for inflation were adopted from Thomas Elder Markets, Data and Consultancy (http://thomaseldermarkets.com.au). The probability distribution of each price variable was derived from analysis of the price data series using BestFit software (Accura Surveys Ltd) (Supplementary Tables 10, 18-20). Prices of livestock products were correlated. Economic assessments of the baseline and interventions were assessed using the @Risk model. To account for economic risk and uncertainty, we performed Monte Carlo simulations using 10,000 iterations of runs of 10-year annual operating profit (earnings before interest and tax), as well as measures of return on capital.

We applied a hybrid mechanism to GHG post-intervention, where carbon taxes were imposed on residual GHG emissions and carbon credits were issued where net farm GHG emissions were negative. Carbon taxes were defined as the cost of reducing net-positive farm GHG emissions to net-zero, with the quantum of mitigation required being the equivalent of the annual net-positive GHG averaged over the simulation duration. Farm businesses with net-negative average annual GHG emissions received one carbon credit valued at $28 tonne $CO_2e^{-1}$ following Australian Carbon Credit Unit (ACCU) spot prices[121]. Residual GHG emissions above net-zero were taxed at $80 tonne $CO_2e^{-1}$, reflecting actual environmental and social impacts of such emissions (often referred to as the social carbon cost[16,17]). The price differential between carbon credits and taxes ensures that greater emphasis is placed in realisation of net-zero, rather than further mitigation when GHG status is already net negative. The $80 tonne $CO_2e^{-1}$ value attributed to carbon taxes reflects actual social cost of carbon[17], while the $28 tonne $CO_2e^{-1}$ attributed to carbon credits reflects (ACCU) market supply and demand[121]. As in voluntary and compliance markets, when an entity achieves net-negative emissions (exceeding GHG neutrality), surplus reductions or removals beyond net-zero emissions can be sold.

## Normalised multidimensional impact assessments

Normalised multidimensional impact assessments were used to rank all interventions and climate horizons through integration of the relative benefit of each adaptation across economic, biophysical and environmental disciplines into a singular unified metric. Following Gephart et al.[122], liveweight production, net operating profit (pre-carbon taxes) and net farm GHG emissions were selected for normalisation by the maximum value for each corresponding metric, such that normalised values ranged from 0 to 1. Normalised net farm GHG emissions were computed as the additive inverse of 1 [i.e. 1 less the normalised net farm GHG emission factor] given that lower values for this specific metric are desirable. Normalised multidimensional impact was calculated as the sum of three key normalised metrics with equal weighting for each metric, such that each normalised output value ranged from 0 (very low impact) to 3 (representing very high beneficial impact in each of the productivity, profitability and GHG emissions dimensions).

## Incremental, systemic, transformational and contextualised stacking of thematic interventions

The co-design process elucidated distinct adaptation/mitigation themes that were later analysed individually or as combined 'stacked' interventions (Table 1, supplementary table 12). The "Low-Hanging Fruit" (LHF) intervention consisted of simple, immediate and reversible changes to existing farm systems that were considered good management practice and may occur over time in the absence of the present study. Incremental adaptations for LHF included changes in animal management/genetics, feedbase management, plant breeding and improved soil fertility (Table 1, supplementary table 12). The second thematic adaptation was co-designed with an overarching aspiration of reducing net farm GHG emissions year on year, such that the trajectory of net farm GHG emissions over time diminished: "Towards Carbon Neutral" or TCN. Incremental adaptations within TCN comprised longer-term, more difficult, higher cost and sometimes irreversible interventions imposed on top of those in LHF including, but not limited to, pasture renovation with deep-rooted genotypes, injecting livestock with an enteric $CH_4$ inhibition vaccine and planting regionally appropriate trees on a portion of existing farmland or on newly purchased land. A third thematic adaptation 'Income Diversification' (ID) was co-designed with the RRG in which income is derived from sources other than the current livestock farm system through options such as buying additional land in a different agroclimatic region (climate diversification), leasing land to host a wind turbine farm or diversifying part of the farm area with grapes (climate diversification, reduce the vulnerability to market fluctuations). The fourth thematic adaptation/mitigation bundle, described as 'Carbon Neutral' or CN, was created after co-designing pathways designed to reach net-zero emissions (supplementary fig. 8). A summary of each adaptation theme together with subset incremental adaptations are shown in Table 1 (further details in Supplementary Tables 5, 12, 14, 15, supplementary fig. 8).

## Sensitivity of enteric methane mitigation and vegetation carbon sequestration

The mitigation we attributed to the use of red seaweed (*Asparagopsis taxiformis*) as a feed supplement (80%) was designed to examine the impacts of an intervention with transformational potential for enteric $CH_4$ inhibition. Under controlled conditions, enteric $CH_4$ reductions of 80-99% have been observed using *Asparagopsis taxiformis*[41–44,81,82]. In grazing systems, however, $CH_4$ abatement can be more variable due to differential forage quality and supplement intake between animals and seasons[20,81]. To quantify the impact of variability in enteric $CH_4$ inhibition and carbon sequestration in vegetation on net farm GHG and profit, we conducted sensitivity analyses. Following ranges reported in peer-reviewed literature, $CH_4$ mitigation was varied from 10% to 99%[82] while temporal carbon sequestration simulated using FullCAM was perturbed by ±20% based on 95% confidence intervals of applicable tree species in temperate regions[116].

## Reporting summary

Further information on research design is available in the Nature Portfolio Reporting Summary linked to this article.

## Data availability

Data generated in this study are available at https://doi.org/10.5281/zenodo.15054102. Greenhouse gas emissions, production, economic, and climatic data generated or used in this study are provided in the corresponding Supplementary Information/Source Data file. Source data are provided with this paper.

## Code availability

Scripts used to process input data to reproduce the plots developed for the sensitivity analysis is publicly available at https://zenodo.org/records/15054102.

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

## Acknowledgements

We thank Meat and Livestock Australia Limited (MLA). The University of Tasmania, the Tasmanian Institute of Agriculture and University of Melbourne for financial support (F.B., M.T.H., K.M.C-W. and B.M. received funding from P.PSH.1219; B.C., B.M., and M.A. received funding from project P.PSH.1248). We acknowledge the important contributions of the Regional Reference Group members, in particular the two case study farmers, to the success of this study.

## Author contributions

F.B., M.T.H., K.M.C-W., B.M., B.C. and M.A. designed the project. F.B., M.T.H., K.M.C-W. and N.B. acquired research data where the acquisition required significant intellectual judgement, design or delivery. F.B., M.T.H., K.M.C-W., B.M. and N.B. analysed the research data. F.B., M.T.H., K.M.C-W., B.M. and M.A. drafted the paper. F.B. and M.T.H. revised interpretations, logic and defensibility of arguments posed and responded to reviewer's critique. M.T.H. sourced case study farmers, regional reference group, acquired funding and was chief investigator of the project.

## Competing interests

The authors declare no competing interests.
