## [Transparent Peer Review file · Nature Communications]

Costs of transitioning the livestock sector to net-zero emissions under future climates

Corresponding Author: Professor Matthew Harrison

Version 0:

Reviewer comments:

Reviewer #1

(Remarks to the Author)

This study aimed to develop pathways for achieving net-zero emissions in livestock farming systems under future climates in Australia. Agriculture, including both livestock and croplands, is a complex field involving various disciplines. Therefore, it is important to clarify that this study specifically focuses on the livestock sector in Australia in the title, abstract, introduction, and discussion sections. Additionally, it is worth noting that a recent review study by Rosa and Gabrielli (2023) provides valuable insights into net-zero emissions in agriculture as a whole.

To enhance the introduction, it would be beneficial to incorporate the findings of the aforementioned review study, which covers net-zero emissions in agriculture, including both croplands and livestock. This will provide a comprehensive overview of previous work and help contextualize the novel analyses conducted in this study. Previous work and the novelty of this study should be explained.

The review study can be accessed at this link: <https://iopscience.iop.org/article/10.1088/1748-9326/acd5e8/meta>.

In terms of the conclusion, instead of using bullet points, it should be restructured as a paragraph following the format commonly used in Nature journals. This will provide a more cohesive and concise summary of the key findings and their implications.

Reviewer #2

(Remarks to the Author)

Main manuscript

This manuscript describes a transdisciplinary study of pathways toward “carbon neutrality” for beef and sheep farms in southern Australia. The authors are to be congratulated on their integrative visioning. The scope and approach is valid, using appropriate models. However, as it stands, the manuscript is let down by: (i) a lack of clarity on precisely how the multiple strands data were integrated; (ii) a loose discussion section somewhat disconnected from the more solid results, with many assertions and too few citations. A dive into the supplementary information raises some questions. In particular, the treatment of carbon stock changes through time is not framed clearly. Forestry ramps up to very large (>30 Mg CO₂e ha⁻¹ yr⁻¹) offsets by 2030 - is this based on immediate planting and steep forest growth curves? Feeding biochar appears to be attributed live weight gain improvement and enteric methane reduction without any evidence for these effects (the soil effects are better justified). The enteric methane abatement of 80% achieved by asparagopsis is incredibly optimistic and beyond any likely effect in reality. I appreciate the authors are visioning ahead to 2050, which is very valuable and will necessarily involve some “what-if” extrapolations. But at the moment these extrapolations are treated as central estimates and used as the basis for concluding on packages of measures, without any sensitivity or uncertainty analyses around what the impacts of reduced efficacy in the aforementioned dominant measures would be. In summary, this study is an interesting and impressive effort with a sound overarching approach, based on relevant data sources and models. However, in my view, the method needs further explanation and results need to be more carefully and precisely interpreted in order to support robust conclusions.

Specific comments (main manuscript)

L35 “evoke” diminishes the causal effect

L50 “collectively” typo

L86-89: wording is imprecise – meaning unclear. How are impacts of production system, intervention and climate change directly compared?

46-72% reduction from seaweed supplement (in SI it is described how this derives from an 80% reduction in treated animals). L351-367 discuss the high uncertainty about the efficacy of asparagopsis in practise. Maintaining such high efficacy rates in the core analysis, without any sensitivity analysis, is misleading given the emphasis that is placed on this measure in the conclusions.

L99-100 and L162-165: what exactly is the scope of the analysis in relation to capturing (or not) all the effects of buying farm in a different region for breeding animals? Is the analysis strictly “farm level”? The scope of “net zero” accounting must be very clearly described early on.

L104 typo

L110-114: Transformational improvement in animal genetic feed conversion efficiencies (TFCE) promises increases in livestock production and farm profits by 8-39% and reducing net GHG emissions by 11-17%, though is aspirational according to the RRG because further livestock genetic science is needed to improve FCE before such genotypes could be widely available (Fig. 3a, 3d, Fig. 4a, 4d).

How was this parameterisation grounded (20-30% animal level FCE seems high)?

L117-120: wording implies reduction in GHG removals owing to climate change impact. Some context of the magnitude of these GHG removals (in 2030) should be introduced first. See comments on soil carbon calculations in relation to the SI. There is a lack of detail on tree planting rates and growth curves – critical to offset calculations. Was new forestry assumed not to be harvested (no revenue)?

Fig 1a+b & 2a+b: legend should be moved out of graph area where it obscures data pattern

L145-155: Reference to anecdotal evidence made here to justify stakeholder (RRG) prioritisation of biochar(OK). Experiments showed no effect on live weight gain, yet in Table S17 it is specified that a 10% beef weight gain and a 10% reduction in enteric CH₄ is attributed to biochar(in addition to better justified soil C effects). These animal performance effects of biochar are not justified, undermining the validity of results and conclusions.

L175-177: case study example farms are used to justify widespread deployment in future. Care and much more careful discussion needed around this. This invokes a critical question: is the farm scale appropriate to consider net zero targets, or should these targets be determined at landscape level, acknowledging diversification across, rather than within, farms (may be more realistic).

L201-210: multiple references to net zero emissions and carbon neutrality. These terms must be explained clearly from the beginning. In particular, carbon removals and temporal scope require further justification for the few CN scenarios that achieve negative GHG emissions.

Fig. 3 inadequate in current format. Requires legends and elaboration. Does profitability include future carbon tax mentioned in text?

L222-243: some interesting insight here, but the overall picture is not clearly presented because of the changing reference situation – with/without carbon tax. How does the profitability of mitigation options packages compare (to baseline) without a carbon tax?

L247: which scenario does “do nothing different” refer to?

L267-272: authors need to critically reflect on the genuine strengths and limitations of this study.

L291: typo

What is the difference between a farmer buying land elsewhere to plant with trees, vs purchasing carbon credits from e.g. a forestry company that does the same (and might manage trees and wood better to achieve downstream mitigation from wood use?)

L324: 25 years cannot be regarded as permanent in terms of climate mitigation!

L325-326: potentially consuming arable zones that go a long way towards fulfilling the growing global need for protein, fibre and starch. What does this mean?

The conclusion that individual actions have little impact on GHG emissions or productivity is not supported by the results which indicate very large GHG reductions from asparagopsis feeding and tree planting (for example).

Appropriately contextualised bundles of interventions to local conditions like planting trees, renovating pastures with deep rooted legumes and adopting high FCE animal genotypes not only have the potential to reduce farm business GHG emissions to net zero, but also improved profitability and productivity gains; Did any scenarios achieve net zero without tree planting elsewhere?

L423: typo

SI document

Asparagopsis: Enteric CH₄ fermentation is reduced by 80% in SB-GAF model outputs, for weaned animals. This is a massively ambitious effect.

Feeding biochar: “For this adaptation options, we assumed an increasing liveweight production about 10% for the beef farm and altered enteric CH₄ fermentation in SB-GAF by 10% in both case studies to reflect an intervention to reduce emissions.” This is not justified by any evidence, and in fact contradicts a study referred to where biochar feeding showed no significant influence on animal performance. This needs to be corrected in the modelling.

Transformational feed conversion efficiency: 20-30% improved FCE by 2050!

Soil carbon cycling is modelled using RothC and biochar parameters: biochar feed additive. 3% of biochar assumed to be labile manure C, 97% recalcitrant – of which 65% is carbon with a decay of <12% over 100 yrs. OK.

In general, the soil carbon changes are modest and look plausible, but a few stick out and require further justification. E.g. for the 2030 climate beef system: lucern renovation of pastures contributes a large effect of 1.36 Mg CO₂e ha⁻¹ yr⁻¹. This seems surprising. How is lucern renovation managed (I understand that lucern will crop for 4-6 years, but then the field requires replanting, which will have soil carbon costs!) Forestry leads to >30 Mg CO₂e ha⁻¹ yr⁻¹ by 2030. Given realistic planting rates and slow initial tree growth curves, this seems very high and requires further explanation.

Table S5 & S6: some of the pathway SOC accumulation rates seem high, e.g. 1.65 and 0.77 Mg CO₂e ha⁻¹ yr⁻¹ for TCN in

2030 or 2050, respectively. Are these entire farm averages, and if so, how are they higher than individual values displayed in Tables S1 & S2. Are some effects additive (and is this justified, e.g. through Roth-C modelling of multiple changes)? Table S10. How does asparagopsis increase ocean C sink? Please explain Tables S15 & S16: please specify daily live weight gain (margins > 0.01-0.025 specified, but these are low and LWG is a critical parameter determining both productivity and GHG emissions, so should be specified).

Version 1:

Reviewer comments:

Reviewer #1

(Remarks to the Author)

The authors have revised the manuscript appropriately. I recommend acceptance of the study.

I still think that the title is too broad and generic but I leave this decision to you and the editors. My is just a constructive feedback. The current title is "Costs of transitioning to net-zero emissions under future climates", where is this article doing the transition? In what sector? It can be a transition in the steel or cement sector from the current title. This should be clarified in my opinion. Please also be specific about 'future climates' in the title as I do not see climate projection analyses.

Reviewer #2

(Remarks to the Author)

The authors have addressed many of the issues raised in the last round of review, and have provided a lot of clarification that really helped to understand the diverse data sets and methods employed. However, a few serious issues remain to be addressed, in particular around the overly optimistic default assumption on asparagopsis methane inhibition in grazing animals, and high (and time-limited) carbon dioxide removal values for planted trees. These two measures do much of the "heavy lifting" for the net zero pathways, thus conclusions are extremely sensitive to related assumptions. The new sensitivity analysis re asparagopsis efficacy is a step in the right direction, but remains optimistic overall (with a 40% efficacy base) and does not appear to address implications of additional tree planting areas required to achieve net zero. These implications need to be fully explained to make the sensitivity analysis meaningful. Furthermore, the default assumption used for main scenarios should also be reduced to ensure results are more credible. Some evidence around important measures such as biochar feeding is also patchy and at least needs more rigorous evidence. Overall, the discussion section needs to be tightened up from a scientific perspective, with more critical reflection on the selected level of analysis (farm enterprise), and on the definition and time frame of net zero. The plausibility of farmers buying remote land to meet net zero targets at an enterprise level did not seem high based on stakeholder feedback, and the cost of the alternative approach of buying offsets is not clearly presented anywhere (despite being stated to be higher than enterprise action). Overall, the manuscript has merit and the ambition of the approach is commendable, but the execution and description requires some refinement.

Specific comments

Agriculture" or "Farms" or "Farming" or similar should be mentioned in the title

L42-43: did the UN FCCC from 1992 already target stabilised concentrations, or emissions? Better to start with emission reduction targets of Kyoto, and delete former lines to save space.

L45-46: the terms "carbon neutrality" and "climate neutrality" are implicitly distinguished, without elaborating what either of them means. A clear definition of carbon neutrality is crucial for this manuscript.

L63: numerous studies have looked at stacked interventions. More accurate to say that few have studied the combinations of mitigation actions and CDR needed to achieve net zero at farm enterprise level.

L112-113: Presumably only climate data are for 1986-2005, and baseline farm performance reflects more recent performance? Please clarify

L128 and Fig. S3: The authors have included sensitivity analyses around the efficacy of asparagopsis in figures S3 and S4. However, the main results still appear to be based on the extremely ambitious assumption of an 80% reduction. The references used to justify this appear to be based on continuous indoor feeding. Recent work on grazing animals pertinent to the systems being modelled indicates much lower efficacy of methane inhibitors (in region of 10% emission reduction). Using 80% as a central estimate in the main analysis for this major emission source undermines the results of this study. A more conservative and defensible estimate for grazing systems should be used here. Sensitivity analysis should also start from a much lower minimum, e.g. 10%, rather than 40%.

L148-151: Good to see the additional clarification and references here - the FCE gain over the time period in question seems plausible.

Fig. 1 & 2: where is the measures of buying external carbon offsets on the mitigation cost charts? This is mentioned to be expensive elsewhere (e.g. L327), but no data shown?

L176-178: Confusion between opinions of RRG and scientifically-derived mitigation performance in this sentence.

Presumably RRG provided information regarding adoptability only? It seems from next sentences that LWG, enteric CH₄ and SOC effects were based on an as-yet unpublished experiment, an article from 2012 published in non-ISI journal (Livestock Research for Rural Development), a non-peer-reviewed report. This is not robust evidence. A brief perusal of recent peer-reviewed literature indicates lack of conclusive evidence on efficacy of biochar for methane mitigation and weight gain, e.g. <https://www.frontiersin.org/articles/10.3389/fvets.2019.00308/full>, <https://www.publish.csiro.au/an/an20295> This needs better justification or else modification.

L197-209: It seems that RRG feedback was somewhat negative on numerous interventions here (buying remote land, diversifying into grapes and wind turbines), yet the authors assert that these options are worth considering. Why is the "extra

farm" in Fig. 1b not "low" ease of adoption in light of these comments?

L261: Should be 38-87% of "gross" emissions? Net emissions implies offsets already subtracted.

L278: unclear what adoption of mitigation %s refer to...

L314-316: I don't think it is sufficient to simply say that landscape scale analyses were outside the scope of this study. More critical reflection on whether landscape analyses would yield different conclusions to farm-enterprise level analysis is needed.

L334: tree areas should be presented as % farm areas to provide proper context

L370-371: The author response regarding permanence of carbon "insetting" requires more critical insight, given that the lower end of the scale suggested, 25 years, implies serious lack of durability to "net zero" solutions identified. This is an issue for critical discussion.

L379: would higher food prices be a perverse outcome? Or reflect a "fair price" that internalises the many externalities of current food production?

L398-412: The text doesn't accurately reflect the extremely optimistic default assumption of 80% enteric CH4 mitigation efficacy for asparagopsis (mentioned previously). A major impact of reduced efficacy is not achieving net zero with the listed measures, even for the CN scenarios (as shown in Figs S3 and S4). Presumably achieving net zero at 40 or 60% efficacy would require a lot more tree planting (and thus cost). This is so important for results it needs more rigorous explanation. Methods

The methods still lack a basic definition of the time frame over which "net zero" or "carbon neutrality" was defined, and how. This appears to be a balance in all GHGs, based on GWP100 values, at the year 2030 and 2050?

The authors have clarified the "farm enterprise" level of analysis, with possibilities for farmer enterprises to buy geographically distant parcels of land for other uses such as tree planting.

Fig. S1: The authors have elaborated the tree C sequestration values derived from the model they ran, and propose that 8 t C / ha / yr is a reasonable average for eucalypts in Tasmania. However, this is over 25% higher than eucalyptus growth rates cited for southern Australia. Data in Fig. S1 are presumably from the model runs (clarification needed). Panel (a) displays an annual C accumulation of up to 15 t C / ha for beef farms. This would equate to >60 m³ /ha annual peak growth! Possible Mg C has been confounded with Mg biomass dry matter on the right hand side axis of the graph? Further clarification and justification is needed for the growth rates assumed.

R#2: L201-210: Original comment not addressed: "In particular, carbon removals and temporal scope require further justification for the few CN scenarios that achieve negative GHG emissions."

Version 2:

Reviewer comments:

Reviewer #1

(Remarks to the Author)

I recommend acceptance of the manuscript

Reviewer #3

(Remarks to the Author)

Review of Manuscript #NCOMMS-23-21332B Costs of transitioning the livestock sector to net-zero emissions under future climates

Submitted to Nature Communications

General Comments:

This study's approach, incorporating input from a group of industry practitioners to codesign pathways for achieving net zero emissions in sheep and beef enterprises in Australia, represents an innovative action-oriented partnership that deserves a wide audience.

The paper has strong merit as an example of much needed transdisciplinary research, with both biophysical scientists and social scientists engaging with stakeholders to co-develop research scenarios, approaches, and criteria. The authors have addressed most previous concerns, although the discussion and conclusions can still be improved to better convey the study's assumptions, scope, and system boundaries. In particular, the terminology about carbon mitigation and carbon prices is inconsistent, as further discussed below.

This manuscript bridges several distinct domains of science, policy, and practice where some of the key terminology is used in different ways. For the most part the manuscript uses these terms in accordance with the emerging consensus definitions in the carbon dioxide removal (CDR) and emissions reduction community, although there are a few minor exceptions pointed out in the specific comments. However, it would be helpful to provide a different set of citations to support the overview of these terms in lines 46 to 50, as the existing citation (Pan et al. 2023) is not consistent with these consensus definitions, calling "removals" via carbon sequestration an "emissions reduction." The terms where definitions should be clear and consistent include:

- mitigation: activities that limit greenhouse gas emissions from entering the atmosphere and/or reduce their levels in the atmosphere. (IPCC 2022 https://www.ipcc.ch/report/ar6/wg3/downloads/faqs/IPCC_AR6_WGIII_FAQ_Chapter_01.pdf);
- reduction and avoidance: reducing or eliminating the amount of greenhouse gases entering the atmosphere.
- removal: the withdrawal of greenhouse gases from the atmosphere as a result of deliberate human activities (IPCC 2024 <https://apps.ipcc.ch/glossary/>);

- insetting: activities that reduce emissions or achieve carbon removal within a farm, enterprise or organization (WEF 2022 <https://www.weforum.org/stories/2022/03/carbon-insetting-vs-offsetting-an-explainer/>);
- offsetting: the compensation for emissions from a given source (e.g., within a farm, enterprise, or organization) with a carbon credit that represents and equivalent emissions reduction or carbon removal accomplished elsewhere outside of the farm, enterprise or organization. (IPCC 2024: The reduction, avoidance or removal of a unit of greenhouse gas (GHG) emissions by one entity, purchased by another entity to counterbalance a unit of GHG emissions by that other entity. Offsets are commonly subject to rules and environmental integrity criteria intended to ensure that offsets achieve their stated mitigation outcome. Relevant criteria include, but are not limited to, the avoidance of double counting and leakage, use of appropriate baselines, additionality, and permanence or measures to address impermanence. <https://apps.ipcc.ch/glossary/>);
- carbon credit: a market instrument in the form of a tradable certificate representing one metric ton of carbon dioxide equivalent emission reduction, avoidance, or removal as a result of a project, intervention, or activity. (WEF, 2020 <https://www.weforum.org/stories/2020/11/carbon-credits-what-how-fight-climate-change/>)

While many analyses equate reduction, avoidance, and removal, those are not the same in terms of long-term climate solutions. Reduction or avoidance of emissions can slow the trajectory of climate change but cannot reverse it. Actual removals are needed both to counterbalance residual emissions that cannot be reduced or avoided, and to draw down atmospheric carbon dioxide to return the climate to a safe operating space. Because the range of mitigation practices evaluated in this study includes reductions as well as removals, insets, offsets, and credits the authors should make sure they provide clear definitions of these terms early in the manuscript and then use them consistently throughout. As discussed further in the specific comments, the authors should also clarify the prices associated with carbon taxes and/or carbon credits, which appear to currently be used interchangeably.

While the authors have provided substantial additional justification for the use of *Asparagopsis taxiformis* to reduce enteric methane production, two of the other interventions that are combined in the carbon neutral packages have important limitations that are not discussed. Planting lucerne, which appears to be the least cost approach to carbon removal in several scenarios (see Figures 1 and 2), does not appear to adjust its carbon removal for the nitrous oxide emissions associated with this and other legumes (e.g., the clover intervention). While these emissions may not be addressed in the models used by this research team and therefore be beyond the scope of this study, there are published studies in the literature, so this impact should at least be discussed.

Similarly, the discussion of tree planting interventions does not mention the risk of wildfire. Given the increasingly high likelihood of wildfire in Australia as well as most other terrestrial ecosystems, a buffer pool (i.e., planting trees in many locations and setting credits aside as insurance against risk of losing forest carbon due to wildfire) might be needed to ensure these insetting interventions are able to continue to counterbalance emissions in the farm enterprises. While assessing fire risk and calculating the needed buffer size and cost is beyond the scope of this study, this limitation and its directional impacts on cost and carbon removal should be discussed.

Specific Comments:

Line 46. Replace “pathways to net zero carbon emissions” with “pathways to net zero GHG emissions” as net zero includes non-carbon solutions. Remove the word “mitigation”. As defined above by the IPCC, “mitigation” is an overarching term that includes both reductions and removals, The syntax of “reduce, avoid or mitigate” implies mitigation is a third option in this set, that is somehow different from reduction and avoidance, and also distinct from sequestering and removing carbon mentioned in the following phrase in the sentence. Incorporate references to support this terminology, such as those suggested in the general comments above.

Line 48. Replace “offset against” with “used to counterbalance” to avoid using the word offset within the definition for inset. Lines 50-52. Consider specifying “anthropogenic”, as in “global anthropogenic GHG emissions”.

Lines 55-56. Saying 40% of “global *agricultural* GHG emissions” here after saying the *agrifood* sector contributes 30% (lines 50-52) is somewhat misleading. Enteric methane represents 18% of agrifood sector emissions and 37% of farm gate emissions (based on FAO data). Suggest 1) adding text to clarify that “agricultural emissions” represent a portion of “agrifood sector” emissions, and 2) include Pg CO₂e per year for the agrifood sector, agricultural emissions (presumably farm gate), and enteric methane.

Line 59. Specify “GHG emissions reduction” not just “reduction” since not only carbon dioxide (the term used immediately prior) is involved.

Lines 66-67 and Lines 85-86. Again, “mitigation” seems to be used as a subset of its consensus definition, distinct from reduction, avoidance and/or removal. In line 66 suggest replacing “mitigation” with “GHG emissions reduction, avoidance,” In lines 85-86 suggest swapping “reduce” and “mitigation” so that it reads “*mitigate* GHG emissions (e.g., through carbon removals, *and* GHG emissions avoidance *and reduction*)”

Lines 154-155. Was the impact on nitrous oxide emissions from lucerne included in this analysis? Continuous lucerne/alfalfa can have significant nitrous oxide emissions (e.g., <https://www.nature.com/articles/s41467-023-37391-2>).

Line 155. The manuscript states that “trees reduced productive pasture area and livestock carrying capacity”. It would be helpful to provide some context. While the beef scenario was largely eucalyptus and the sheep scenario included more diverse species, it was not designed for forage. However, some tree species can provide feed and forage for sheep and

cattle as well as shade. Line 722 notes “(silvopasture) was not permissible following advice from the RRG”. While the rationale for this decision is not described, this may reflect regional circumstances, and silvopasture may be a preferred practice in other regions. Additional context here and/or at line 722 would minimize the potential for this result to be incorrectly generalized and discourage silvopasture where it is appropriate.

Line 167. In the legend for figure 2, which is about sheep farms, CCD (changing calving date) is defined but not shown, and the figure instead shows Alt LD (alternative lambing date?) which is not defined. Please clarify.

Line 172. Reviewer #1 pointed out “evidence around important measures such as biochar feeding is also patchy and at least needs more rigorous evidence.” In response the authors added citations (Line 172) but should also mention that there are upstream emissions associated with biochar production that were not considered in this study. These upstream emissions could potentially reduce the climate benefits of this intervention.

Lines 182-184. Given the 15% to 20% increase in protein emissions intensity from the purchase of additional land in a diverse agro-climatic region, it would be helpful if the authors summarized the primary driver(s) of these increased GHG emissions. It would also be helpful to point the reader to the supplemental information about this particularly complex intervention, especially table S17 detailing this strategy for adaptation.

Lines 234-242, caption for Figure 3 and (see comment immediately below)

Lines 274-281, caption for Figure 4: These figures are central to reporting the results of the holistic analysis for various “stacked” combinations of practices. The left side of the figures includes modeled quantities (production, profit, emissions) with explicit units, while the right side of the figures aggregates the three variables in (a) and (d) into the stacked horizontal bar charts in (b) and (e). Because of the structure of this journal your description of that conversion (the normalization in the section titled “Normalised multidimensional impact assessments” is not anywhere near the figure. Please indicate this is a normalization in the figure caption and/or in the text here, and also refer the reader to that later section. It would also be helpful to explain how (b) is converted to (c) and (e) is converted to (f), although the axis legends in the pyramids (c) and (f) stating % factor provide a hint, presumably meaning that these are the percentages each variable contributes to the totals in (b) and (e). But that may not be obvious to many readers.

Line 247. The explanation of the carbon tax is much later in the methods section, so here, where the results are reported, a quick mention of the specific tax rate would provide important context to these results. For example, “...(business as usual), carbon taxes of \$XX/Mg CO₂e would...”. As noted elsewhere in this review, a range of \$60-\$100/Mg CO₂e is provided in Line 739 but it is not clear whether the authors used an average of that range (\$80/Mg CO₂e), different prices for the 2030 and 2050 time periods, or something else. Please also see other comments about the use of the term carbon credits, apparently as a synonym for carbon taxes, without explanation.

Line 251. Should this say “inset” or “counterbalanced” instead of “offset” to differentiate from a system where the farm is purchasing offset credits from outside the farm enterprise?

Lines 251-252. A previous reviewer asked if this should be “gross” and the authors noted it was amended in their response, but the text still says “net.” I think this should be gross GHGs, not net.

Line 257. It appears a word is missing, presumably “reduce”. Should it read “significantly reduce net GHG emissions.” Line 271. Should this say carbon *credit* instead of carbon *offset*?

Lines 340-341 and Line 356. Although previously (see comments for Lines 246-247) and in later in the methods section (Lines 739-741) the manuscript references the price of a carbon tax, here the terminology seems to have shifted to carbon credits. Is the cost of a market-based carbon credit (that presumably is backed up by some sort of certified carbon removal) assumed to be the same as a government required carbon tax (noting that a carbon tax may or may not result in carbon removal)? Please clarify here and in the methods section, and then be consistent about terminology. At this point the manuscript uses the term carbon tax thirteen times and carbon credit ten times, so a manual search-and-replace (to assure correct syntax) would not be too onerous.

Line 342. Does “direct removal” mean insetting with carbon removal? If so, the authors should clarify.

Line 366. Here the authors define “insetting” with a parenthetical “(practices to reduce GHG within the value chain). The term “value chain” can refer to a single enterprise, but often extends across multiple enterprises as a product travels from producer to consumer. As previously indicated, emerging consensus defines “insetting” as within a single enterprise. Please clarify this definition for consistency.

Line 378. This sentence starts “Future market demand for carbon credits (e.g., from the transport and energy production sectors)”. It would be helpful to mention that in the future carbon credits will be needed in the future to draw down atmospheric carbon (achieve negative emissions) to bring the climate back to a normal state, in addition to counterbalancing residual emissions that are difficult or impossible to avoid entirely. The authors may also want to change the examples from “transport and energy production sectors” to “other agriculture, industry, and transport” as the scientific literature anticipates the majority of residual emissions will be from agriculture with some from industry and transport sectors (e.g., Buck et al. 2022).

Line 388-411. This discussion about permanence and temporal variation of nature-based carbon removal strategies, including planting trees, would be a good place for a brief note about the risks of losing carbon stored in trees through wildfires. As mentioned in the general comments, across the globe forests are at increasing risk of wildfire, including in the region in eastern Australia and Tasmania targeted by this study. It's important to at least mention this risk, and that buffer pools (i.e., setting aside a certain fraction of carbon credits generated by a project as insurance against unanticipated losses), which were not included in the analysis, would likely need to be included for effective implementation of this concept. Here, or elsewhere in the paper, the authors should also mention that carbon credits / offsets have requirements, including e.g., additionality and permanence, that may differ from the effects of the inseting practice assessed here (planting trees). For example, there are several biomass carbon removal and storage (BiCRS) approaches that result in permanent or geologic sequestration. For the global "livestock sector" to reach net zero emissions, permanent removals will likely be required, and while these were not explored in this study they deserve mention in the discussion.

Line 394. There appears to be a typo, where "avoiding" should be "avoid".

Line 412, Figure 5. In the figure, suggest adding sheep and beef labels to the a-d and e-h plot for increased clarity. In the caption, please indicate the price of the purchased carbon credits (\$80/Mg CO₂e?) as well as the sold carbon credits (\$28/Mg CO₂e). As mentioned elsewhere, please be consistent about terminology. For example, here in this one figure the legend includes both carbon credits and carbon tax on lines 413 and 414, and uses "post carbon tax", "bought carbon credits", and "sold carbon credits" on the color legend. If the authors are going to use the "carbon credit" terminology instead of "carbon tax", then please be consistent.

Line 437. This is another good place to include the amount of the carbon tax/carbon credit price (whichever is selected), as in "after an \$XX Mg-1 C-1 carbon tax of AU D 321-340K..."

Lines 452-453 As per the consensus definitions previously discussed, "mitigation" is the overarching term and not a subset. Suggest either eliminating the parenthesis entirely and simplify to "carbon reduction and removals given that sequestration..." or replace with something like "GHG mitigation (reduction, avoidance, inset and offset carbon sequestration) given that sequestration..."

Lines 485-486. As mentioned in the general comments, there should be a mention here of nitrous oxide implications. There is a common assumption that legumes emit little nitrous oxide, but studies have shown otherwise. For example, Anthony et al. 2023 (<https://doi.org/10.1038/s41467-023-37391-2>) report that nitrous oxide emissions reduce the potential carbon sink of alfalfa/lucerne by 14%.

Lines 492-495. This paragraph is providing an overview of "discoveries related to prospective pathways to net zero farm enterprises." These specific lines summarize several interventions including purchasing land in diversified climate zones that increased productivity and profitability. While presumably this intervention was thought to be a potential pathway to net zero, it turned out to dramatically increase GHG intensity, not just for the farm but per kg of livestock live weight and protein produced. While not as dramatic, increased GHG intensity was also consistently observed for the increased stocking rate and alternate lambing/calving date interventions (see figures 1 and 2 as well as tables S3 to S6). While the RRG proposed some interventions for income and/or climate diversification (lines 523 to 532, table S16), given the stated goal of this study (pathways to net-zero) it is important to report that some interventions actually increased emissions intensity. In the case of purchasing land in a diversified climate zone this increase was quite large. This is an important discovery, and therefore should be discussed in this section.

Lines 669-672. These lines detail the nitrogen-related GHG emissions and are reported as including: "N₂O from nitrogenous (N) fertiliser, waste management, urinary deposition and indirect N emissions via nitrate leaching and ammonia volatilisation". This list does not include direct N₂O emissions from nitrification/denitrification in cropland. It is possible that SB-GAF version 1.4 calculates cropland N₂O emissions as a function of N fertilizer inputs (a reasonable simplification for most cropland where synthetic fertilizers are used) and therefore does not consider N₂O derived from N fixed by legumes. However, as previously noted (see comments for lines 154-155 and lines 485-486), there are significant N₂O emissions associated with N-fixing legumes, including lucerne and presumably also talish clover. These emissions could reduce the carbon benefit of these crops, so if they are not reflected in this model that limitation should be discussed. This is especially important since legume interventions were included in the "low hanging fruit", "toward carbon neutral" and carbon several neutral packages.

Line 739-740. Here the carbon tax price is provided as a range (\$60-\$100/Mg CO₂e) and not a specific number, while the next sentence appears to be missing some words, stating: "Any carbon sequestration beyond net farm GHG emission were 'credited' at \$28/Mg CO₂e [missing word or words] that of a carbon tax." Because this is such an important component of the profitability analysis, it was previously suggested that carbon price be incorporated in a few key places in the results and discussion. In addition, this methods section needs to be clear about the price(s) that are used and the current language and range raises several questions. Is the stated carbon tax also used as the price for purchasing carbon credits? If there is just one value used for the carbon tax, what is it? If two or more values in this range are used, what are they and when is each used? And if the carbon sequestration "credits" are sold off-farm at only \$28/Mg CO₂e, why is that price so much lower than the carbon tax and/or carbon credit purchase price?

(Remarks to the Author)

Version 3:

Reviewer comments:

Reviewer #3

(Remarks to the Author)

Review of Manuscript #NCOMMS-23-21332B Costs of transitioning the livestock sector to net-zero emissions under future climates

Submitted to Nature Communications

General Comments:

The paper has strong merit as an example of needed transdisciplinary research, with both biophysical scientists and social scientists engaging with stakeholders to co-develop research scenarios, approaches, and criteria. The authors did an excellent job of addressing the previous suggestions. We recommend the following minor changes and subsequent acceptance of this paper.

Specific Comments:

Line 24 "Serendipitously" should be spelled serendipitously.

Line 44: There is a missing word in this line: ...aimed at limiting GHG emissions...

Line 75 Suggest changing "global agricultural GHG emissions" to "global agricultural (farm level) GHG emissions" to be consistent with the language in lines 67-70. According to recent FAO data, enteric emissions represent 35% of global agriculture (farm gate) emissions or 18% of global agrifood systems (farm gate, pre- and post- production, and land use change).

Lines 289 and 323 (figure captions): This new phrasing with an "I" is unclear and might benefit from including mention of all subparts of this figure with normalized values, as in:... Normalised values for each metric in (b), (c), "(e), and (f)," range from zero to one.

Line 390 "Serendipitously" should be spelled serendipitously.

Line 400. Unfortunately this sentence is now past tense: For example, the US government has "used" the SCC to ... (see <https://www.propublica.org/article/trump-climate-change-social-cost-of-carbon-executive-order>)

Line 593. This appears to be the first use of the acronym BNF for Biological Nitrogen Fixation, so should be defined. Reviewer #4

(Remarks to the Author)

Reviewer #1 (Remarks to the Author):

R#1: This study aimed to develop pathways for achieving net-zero emissions in livestock farming systems under future climates in Australia. Agriculture, including both livestock and croplands, is a complex field involving various disciplines. Therefore, it is important to clarify that this study specifically focuses on the livestock sector in Australia in the title, abstract, introduction, and discussion sections.

We revised the introduction, results and discussion sections, clarifying the location and type of study. We strongly believe that titles should be generalized to attract a broad readership, and that details of the study (e.g. discipline and location of analyses conducted) should be mentioned in the abstract; these sentiments are reflected both in the *Nature Communications* guide to authors and in the title of the paper R#1 suggested below.

R#1: Additionally, it is worth noting that a recent review study by Rosa and Gabrielli (2023) provides valuable insights into net-zero emissions in agriculture as a whole. To enhance the introduction, it would be beneficial to incorporate the findings of the aforementioned review study, which covers net-zero emissions in agriculture, including both croplands and livestock. This will provide a comprehensive overview of previous work and help contextualize the novel analyses conducted in this study. Previous work and the novelty of this study should be explained.

The review study can be accessed at this link: <https://iopscience.iop.org/article/10.1088/1748-9326/acd5e8/meta>.

We expanded the introduction to include findings of the study this reviewer mentions, as well as the novelty of this work:

Historically, climate targets have been articulated either as thresholds for stabilized atmospheric GHG concentrations, such as those in the 1992 United Nations Framework Convention on Climate Change, or as emissions reduction targets, such as those in the Kyoto Protocol⁶. The concept of net-zero, however, assumes a balance between GHG emissions and removals⁷ and as such, implicitly comprises ‘carbon neutrality’, rather than ‘climate neutrality’. Achieving net-zero requires alignment with Sustainable Development Goals (SDGs), ensuring an equitable transition, socio-ecological sustainability, and opportunities for productivity growth⁷. In response, bespoke decarbonization pathways have been developed by international organisations, nations, companies and industries. Pathways to net zero typically aim to reduce, avoid or mitigate GHG emissions on the one hand, while sequestering or removing carbon on the other. On an annual basis, carbon removals within a business can be ‘inset’, where new sequestration is monitored and offset against a baseline GHG of the enterprise, while remaining GHG are ‘offset’, often in the form of carbon credits purchased from entities outside the value chain⁸. To achieve net-zero emissions, the agrifood sector must reduce carbon dioxide (CO₂), methane (CH₄) and nitrous oxide (N₂O) emissions, which collectively contribute 30% of global GHG emissions⁹. Mitigation pathways should consider multiple sustainability co-benefits and trade-offs, including CO₂ removal potential, food security, nature conservation and ecosystems services, and other externalities, such as climate change^{10,83}. Putting urgent, deep and sustained cuts in enteric ruminant CH₄ emissions, which comprise one of the most dominant agricultural GHG emissions

(40%, 2.85 Pg CO₂e per year), demands development of innovations for improving livestock feed types, genetics, reproduction and health^{11,12}. While there has been much research of individual mitigation interventions in isolation (e.g. the influence of ewe fecundity or pasture type^{81,82}), few studies have explored holistic implications of stacking (or combining) GHG emissions interventions.

And:

As a corollary, few bona fide examples of food systems transformations exist, perhaps because research has traditionally progressed in a reductionist fashion, with primarily unidisciplinary and siloed foci; the present study addresses this gap through application of a multidisciplinary and holistic systems approach.

R#1: In terms of the conclusion, instead of using bullet points, it should be restructured as a paragraph following the format commonly used in Nature journals. This will provide a more cohesive and concise summary of the key findings and their implications.

Agree. We removed the title 'conclusion' (see above) and restructured as a paragraph:

We uncovered many insights relating to prospective pathways to net zero farming systems. We revealed that singular interventions realised limited concurrent improvements in productivity, profitability and GHG mitigation. Under future climates, strategies such as adoption of low-emissions livestock feed supplements (Asparagopsis) and planting of appropriate tree genotypes were most effective in reducing emissions, albeit came with higher economic costs. Diversifying farming enterprises spatially by purchasing land in diversified climatic zones, along with adoption of animal genotypes with transformative feed-conversion efficiency (TFCE) promised substantial benefits in productivity and profitability. While continuing business-as-usual (BAU) and offsetting farm emissions by purchasing carbon credits (offsetting) was perhaps the most adoptable intervention we assessed, BAU and offsetting were also the most expensive strategy, and as such should be discouraged. We underscore the triple bottom-line potential associated with appropriately contextualised bundling of interventions, particularly when they target productivity, mitigation of enteric CH₄ and carbon removals. Interventions such as planting small areas of trees on farm, renovating grass-based pastures with deep-rooted legumes, and adopting high feed conversion efficiency (FCE) animal genotypes were shown to not only realise net-zero GHG emissions, but also improve productivity. In all cases, interventions that realised productivity gains were most conducive to prosperous outcomes. We contend that purported innovations are more likely to be transformational if aspiring developers simultaneously consider multiple sustainability indicators, including environmental stewardship, food security, markets, social licence to operate and the changing climate.

Reviewer #2 (Remarks to the Author):

Main manuscript

This manuscript describes a transdisciplinary study of pathways toward “carbon neutrality” for beef and sheep farms in southern Australia. The authors are to be congratulated on their integrative visioning. The scope and approach is valid, using appropriate models. However, as it stands, the manuscript is let down by: (i) a lack of clarity on precisely how the multiple strands data were integrated; (ii) a loose discussion

section somewhat disconnected from the more solid results, with many assertions and too few citations. A dive into the supplementary information raises some questions. In particular, the treatment of carbon stock changes through time is not framed clearly. Forestry ramps up to very large (>30 Mg CO₂e ha⁻¹ yr⁻¹) offsets by 2030 - is this based on immediate planting and steep forest growth curves? Feeding biochar appears to be attributed live weight gain improvement and enteric methane reduction without any evidence for these effects (the soil effects are better justified). The enteric methane abatement of 80% achieved by asparagopsis is incredibly optimistic and beyond any likely effect in reality. I appreciate the authors are visioning ahead to 2050, which is very valuable and will necessarily involve some “what-if” extrapolations. But at the moment these extrapolations are treated as central estimates and used as the basis for concluding on packages of measures, without any sensitivity or uncertainty analyses around what the impacts of reduced efficacy in the aforementioned dominant measures would be. In summary, this study is an interesting and impressive effort with a sound overarching approach, based on relevant data sources and models. However, in my view, the method needs further explanation and results need to be more carefully and precisely interpreted in order to support robust conclusions.

We have added (1) further details of the methods, including clarity surrounding linkages between modelling approaches, (2) focussed discussion of our results, and (3) better hinged our assumptions on evidence from previous peer-reviewed literature. Including the sensitivity analyses we added, these revisions are described in the results, discussion and supplementary information (Figs S3 and S4); further details are itemised below.

Specific comments (main manuscript)

R#2: L35 “evoke” diminishes the causal effect.

We replaced “evoke” with “causes”.

R#2: L50 “collectively” typo.

Amended.

R#2: L86-89: wording is imprecise – meaning unclear. How are impacts of production system, intervention and climate change directly compared?

We revised this text as follows:

In comparing baseline (status quo) beef and sheep production systems under historical climates (1986-2005) with baseline management of the same farm in 2030 and 2050 (i.e. cf. Figs 1 and 2 panels (a) and (c) with panels (b) and (d)), we show that (1) few individual interventions elicited significant simultaneous benefit on all indicators (productivity, profitability, GHG emissions mitigation, adoptability) and (2) interventions caused greater effects on these indicators compared with impacts of climate change alone (cf. distance between points within panel (a) with differences between the same points between (a) and (b) in Figs 1 and 2). The combination of higher monthly temperatures (414% and lower rainfall in 2030 (3-7%) and 2050 (5-11%) with elevated atmospheric CO₂ concentrations evoked modest increases in pasture production of the beef farm (2-3%) and sheep farm

(7-8%) compared with historical climates. This translated into small increases in meat and wool production and reduced supplementary feed requirements, resulting in greater profit for the beef farm (3%) and sheep farm (36%) in 2050; the larger gain for the latter underpinned by a larger reduction in supplementary feed requirement in 2050 (cf. historical climates).

R#2: 46-72% reduction from seaweed supplement (in SI it is described how this derives from an 80% reduction in treated animals). L351-367 discuss the high uncertainty about the efficacy of asparagopsis in practise. Maintaining such high efficacy rates in the core analysis, without any sensitivity analysis, is misleading given the emphasis that is placed on this measure in the conclusions.

Agree. We added the following to the discussion:

A key insight we revealed was that Asparagopsis showed significant promise as a low-emissions feed supplement, decreasing net farm GHG emissions by 46-72%. Although some in vitro studies have report up to 98% enteric CH₄ reduction associated with Asparagopsis feed supplementation^{84,85}, we suggest that enteric CH₄ mitigation in practice will be more modest, due to the ability of animals to access feed supplements, foraging behaviours, diet composition and enterprise type constrains in situ^{35,84,85}. To determine how our results are influenced by our assumed 80% enteric CH₄ mitigation per weaned animal, we conducted sensitivity analyses hinged upon empirical enteric CH₄ reductions ranging between 40% and 80%³⁶ (Figs S3 and S4 in the supplementary information). This analysis showed variation in net GHG emissions of 763-970 Mg CO₂e for the beef farm and 1,635-2,041 Mg CO₂e for the sheep farm, increasing costs by AUD 59-78,000 for the beef farm to AUD 131-148,000 for the sheep farm. Despite this, relative GHG and/or economic benefits associated with each thematic intervention remained similar, with Asparagopsis as an individual intervention and stacked interventions (CN2) being least and most beneficial, respectively (Figs S3 and S4). As well, differences between thematic adaptation/mitigation intervention bundles were greater than differences associated with assumed enteric CH₄ mitigation, demonstrating that our conclusions are robust.

We do however acknowledge that - similar to any purported GHG mitigation intervention - ultimate societal impact of feed supplementation with Asparagopsis will be dictated by a raft of wider interacting economic, social, environmental, institutional and psychological factors, such as ease of implementation, market supply and price, social licence, government regulation and animal/human health and welfare implications. Forecasts suggest a future Australian 1.5B seaweed production industry, creating up to 9,000 jobs and 10% national GHG emissions reduction per year by 2040, which would comprise a substantial contribution towards UN Sustainable Development Goals³⁷. Even so, many challenges remain to be resolved if such forecasts are to eventuate. For example, scientists have cautioned that certain seaweed species are invasive; such species may detrimentally impact native species in marine environments should they to escape their domesticated environments. Some scientists have speculated that bromoform (the compound in Asparagopsis which inhibits enteric CH₄) may have adverse connotations for animal health or ozone depletion³⁶. Such concerns highlight the need for further research, development and extension to allow careful consideration of the benefits and risks associated with large-scale adoption of seaweed as a livestock feed supplement^{35,38}.

Fig. S3. Sensitivity analysis of mitigation associated with *Asparagopsis taxiformis* as a feed supplement (40, 60, 80% enteric CH₄ reduction) individually (ASP) and combined with other interventions for the beef farm. Pathways 1 and 2 reflect net-zero farming systems attained by improving animal genetics (CN1 and CN2) or renovating grass pasture swards with lucerne (CN3 and CN4) in 2050 climates; ASP: *A. taxiformis* as livestock feed supplement; Asp+PT: *A. taxiformis* + planting trees (50 ha); CN1: carbon neutral package 1 [*A. taxiformis* + planting 50 ha trees + transformational feed conversion efficiency]; CN2: carbon neutral package 2 [*A. taxiformis* + planting 85 ha trees + transformational feed conversion efficiency]; CN3: carbon neutral package 3 [*A. taxiformis* + planting 50 ha trees + Lucerne]; CN4: carbon neutral package 4 [*A. taxiformis* + planting 85 ha trees + lucerne].

Fig. S4. Sensitivity analysis of mitigation potential associated with *Asparagopsis taxiformis* (40, 60, 80% enteric CH₄ reduction) as an individual intervention (ASP) and in combination (CN packages) for the sheep farm. Pathways 1 (a and c) and 2 (b and d) demonstrate realised net-zero farming systems via improvement in animal genetics (CN1 and CN2) or renovating grass pasture swards with lucerne (CN3 and CN4); Base: 2050 climates; ASP: *A. taxiformis* as a low-emissions livestock feed supplement; Asp+PT: *A. taxiformis* + planting 200 ha trees; CN1: carbon neutral package 1 [*A. taxiformis* + planting 200 ha trees + transformational feed conversion efficiency]; CN2: carbon neutral package 2 [*A. taxiformis* + planting 220 ha trees + transformational feed conversion efficiency]; CN3: carbon neutral package 3 [*A. taxiformis* + planting 200 ha trees + lucerne]; CN4: carbon neutral package 4 [*A. taxiformis* + planting 220 ha trees + lucerne].

R#2: L99-100 and L162-165: what exactly is the scope of the analysis in relation to capturing (or not) all the effects of buying farm in a different region for breeding animals? Is the analysis strictly “farm level”? The scope of “net zero” accounting must be very clearly described early on.

Buying a farm in a different region was suggested by the RRG to enable climate diversification; large geographical separation between land parcels would be expected to cause exposure to different

climates, thus reducing the risk that adverse events impact on both production systems simultaneously. We note that 'buying another farm' is perhaps confusing terminology; once purchased, two separate farms become the one farm enterprise. We added the following to the introduction:

For the purpose of GHG accounting, the scope of our analysis was the farm enterprise, noting that farms may have land parcels with large geographical separation between them.

We altered the terminology throughout the article from 'buying an extra beef farm' to 'buying additional land for the beef farm'.

The definition of net-zero emissions *per se* was added to the introduction (see previous responses).

R#2: L104 typo

Amended

R#2: L110-114: Transformational improvement in animal genetic feed conversion efficiencies (TFCE) promises increases in livestock production and farm profits by 8-39% and reducing net GHG emissions by 11-17%, though is aspirational according to the RRG because further livestock genetic science is needed to improve FCE before such genotypes could be widely available (Fig. 3a, 3d, Fig. 4a, 4d).

How was this parameterisation grounded (20-30% animal level FCE seems high)?

The parameterisation is grounded both with feedback from senior livestock geneticists (extensive personal communications with Dr Robert Banks of the University of New England), and peer reviewed literature: Alford et al. (2006) show that gene flow within a breeding herd over 25 years can enhance annual FCE by 0.76 to 1.14%, which translates to 26-36% over 40 years.

We stress that our FCE target is aspirational, hence our labelling of this intervention as "transformational" or "TFCE". In this way, we can contrast farm enterprise outcomes (production, profit, GHG) associated with animal genetic improvements with those of other interventions to determine the merit of research and investment in several interventions with differing modes of action. Such comparisons provide the scientific community with insight as to whether or not research into a particular intervention would be fruitful. We believe that such contrasts will have wide appeal with the scientific community and industry.

We added the following to the results:

While our TFCE target (20-30% gain in animal FCE) is aspirational, our assumptions are grounded on advice from senior livestock geneticists (Dr Rob Banks pers. comm.) and peer-reviewed literature (Alford et al. 2006), demonstrating that our results are rigorous (see methods for further details).

R#2: L117-120: wording implies reduction in GHG removals owing to climate change impact. Some context of the magnitude of these GHG removals (in 2030) should be introduced first. See comments on soil carbon calculations in relation to the SI. There is a lack of detail on tree planting rates and growth curves – critical to offset calculations. Was new forestry assumed not to be harvested (no revenue)?

We added context, new figures and data to the supplementary information to more transparently demonstrate methods used to simulate tree growth, including carbon allocation to different plant components and soil fractions. This reviewer commented above that “Forestry ramps up to very large (>30 Mg CO₂e ha⁻¹ yr⁻¹) offsets by 2030”: this is because the vast bulk of tree carbon sequestration occurs within the first decade of planting (see new figures below; these have been added to the supplementary information). After the first decade, net carbon mass declines, as shown by the solid lines in the charts below (refer to right y-axes). It is important to note that we are referring to carbon dioxide equivalents (CO₂-e), which are (44/12 = 3.67) times greater than carbon per se (ie 30 Mg CO₂-e/ha is 8.3 t C/ha; see Table S1). Field surveys in Southern Australia measured average carbon sequestration of *Eucalyptus globulus* (11.8 ± 3.9 years old) at 23.6 ± 7.9 Mg CO₂e ha⁻¹ yr⁻¹, with sequestration rates of over 30 t CO₂-e/ha/yr at some sites (Fig. 9; Neumann et al. 2011). These findings suggest that our simulated rates of tree carbon sequestration in the higher rainfall zones of Tasmania are reasonable (note also declining sequestration after the first decade). We assumed that planted trees would not be harvested; this perhaps reflects a confusion in terminology, where the reviewer understood that ‘forestry’ implies harvesting. We have clarified this throughout the article. We assumed that carbon sequestration exceeding net farm greenhouse gas (GHG) emissions would be sold, so the enterprise would obtain revenue from surplus carbon credits, not from harvested timber. This has been clarified in the methods.

Neumann, C. R., Hobbs, T. J., Tucker, M. Carbon sequestration and biomass production rates from agroforestry in lower rainfall zones (300-650 mm) of South Australia: Southern Murray-Darling Basin Region (2011) Available at:

<https://data.environment.sa.gov.au/Content/Publications/carbonsequestrationbiomassagroforestry.pdf>

R#2: Fig 1a+b & 2a+b: legend should be moved out of graph area where it obscures data pattern

Agree – we shifted the legend as suggested.

Fig. S1. Temporal carbon stocks by fraction (trees, debris and soil; left y-axes) and annual change in C stocks (solid lines, right y-axes) for tree plantations on beef (a) and sheep (b) farms.

R#2: L145-155: Reference to anecdotal evidence made here to justify stakeholder (RRG) prioritisation of biochar(OK). Experiments showed no effect on live weight gain, yet in Table S17 it is specified that a 10% beef weight gain and a 10% reduction in enteric CH₄ is attributed to biochar(in addition to better justified soil C effects). These animal performance effects of biochar are not justified, undermining the validity of results and conclusions.

These data were drawn from a scientific trial underway at the time the manuscript was submitted; the experiment has since concluded and is under review (Bilotto et al. under review). The values we adopt for animal liveweight were measured in field trials containing 75 steers per treatment (control vs biochar supplemented treatment) demonstrating high levels of replication and rigour in the values we adopted in the modelling. It is also worth pointing out that the liveweight gain and enteric methane mitigation values we assumed are conservative and supported by previous reports. Scientific literature has shown that cattle supplemented with biochar over 98 days had average liveweight gain 25% greater than that of the unsupplemented control and 24% lower enteric methane (Leng et al. 2012). More recent studies have shown that *in vitro* enteric methane mitigation is less than that *in vitro* (9-13% vs 23-33%; Fernandez et al. 2022). Taken together, these data lend credibility to the values we assumed here.

We added the following to the manuscript:

The RRG considered biochar feed supplementation highly adoptable based on anecdotal evidence suggesting that use of biochar in this way could (1) improve liveweight gain, (2) reduce enteric CH₄ emissions and (3) enrich organic carbon content of manure. We assumed liveweight gains and enteric methane mitigation associated with biochar supplementation of 10% based on experiments we conducted in 2024, as well as other scientific literature^{86,87,88} (Fig. S2).

Bilotto F, Christie-Whitehead KM, Barnes N and Harrison MT (under review). Operationalising net-zero with biochar: black gold or red herring?

Leng R A, Preston T R and Inthapanya S (2012) Biochar reduces enteric methane and improves growth and feed conversion in local Yellow cattle fed cassava root chips and fresh cassava foliage. Livestock Research for Rural Development. 24, 199 [Accessed 11 March 2024].

Fernandez GM, Durmic Z, Vercoe P and Joseph S (2022). Fit-for-purpose biochar to improve efficiency in Ruminants. Meat & Livestock Australia. Available: https://www.mla.com.au/contentassets/d90d2eba9b7d4ccba407be2ecbf2fdde/bgbp_0032-biochar-final-report_mla-website-.pdf [Accessed 11 March 2024].

Fig. S2. Cattle liveweight measured (a) and modelled (b) for field experiments conducted at Deloraine, Tasmania, Australia (a) mean liveweight of steers fed biochar *ad libitum* (n=75) and no biochar (control, n=75). (b) Regression analysis fitted to data in (a) to project differences between the control and biochar treatment to liveweight values used in the modelling. Error bars depict standard error of the mean.

R#2: L175-177: case study example farms are used to justify widespread deployment in future. Care and much more careful discussion needed around this. This invokes a critical question: is the farm scale appropriate to consider net zero targets, or should these targets be determined at landscape level, acknowledging diversification across, rather than within, farms (may be more realistic).

This is a good question, but not the question we set out to answer. The question we address was that pertaining to sustainable pathways to net-zero farm enterprise GHG emissions. Nevertheless, the reviewer raises a relevant point. We added the following to the discussion:

While landscape-level assessments offer macroscopic insights on emissions mitigation potential across farms, our purpose here was to derive sustainable pathways to net-zero emissions within farms, in consultation with regional practitioners. This participatory approach helped determine the feasibility of purported interventions, validate assumptions and simulations and, through iteration, allow practitioners to gain confidence in methods we invoked. Indeed, the namesake of our study (costs of transitioning to net zero emissions) was put to us by the RRG, demonstrating that this work is demand driven. Landscape-scale assessments capturing diversity in enterprise mixes across farms and value chains, as well as carbon sequestration potential between farms⁸⁹ were out of scope of the present study, but are worthy of further investigation. Such assessments are typically more granular than farm-scale analyses, due to computational expense⁸⁹. While perceived attainment of ‘net-zero emissions’ depends on the sector or land area in question⁷, many studies agree that action at the farm level must be taken to reduce GHG emissions and avoid dangerous climate change^{77,78,82,89}.

⁸⁹Duffy, C., Prudhomme, R., Duffy, B. et al. Randomized national land management strategies for net-zero emissions. *Nat Sustain* 5, 973–980 (2022). <https://doi.org/10.1038/s41893-022-00946-0>

R#2: L201-210: multiple references to net zero emissions and carbon neutrality. These terms must be explained clearly from the beginning. In particular, carbon removals and temporal scope require further justification for the few CN scenarios that achieve negative GHG emissions.

We added definitions of net zero and carbon neutrality to the introduction, and clarity on the study boundaries to the introduction (see previous responses).

R#2: Fig. 3 inadequate in current format. Requires legends and elaboration. Does profitability include future carbon tax mentioned in text?

We clarify this question in the legend, and dimensions referred to in Figs 3 and 4 (3b, 3d, 4b and 4d).

R#2: L222-243: some interesting insight here, but the overall picture is not clearly presented because of the changing reference situation – with/without carbon tax. How does the profitability of mitigation options packages compare (to baseline) without a carbon tax?

We added the following:

While use of Asparagopsis as a feed supplement decreased operating profit by 7-8%, operating profit (post carbon tax) was significantly greater compared with paying carbon taxes and continuing business as usual (Fig. 5c, d, g, h). When feeding of Asparagopsis was stacked with purchasing an extra farmland that was planted with trees (ASP_i-PT), a further 38-87% net GHG emissions were offset (Fig. 5a, b, e, f). Relative to the baseline farm in which all residual GHG emissions were taxed, ASP_i-PT improved profits by 34% and 68% for the sheep and beef farms, respectively.

R#2: L247: which scenario does “do nothing different” refer to?

To clarify, we replaced, “do nothing different” with “relative to the baseline scenario”.

R#2: L267-272: authors need to critically reflect on the genuine strengths and limitations of this study.

We added several strengths and limitations of the study. Examples:

This study was conducted using participatory research encompassing nascent science (in models, and from empirical studies), along with a transdisciplinary lens, allowing benefits in one dimension (e.g. GHG emissions mitigation) to be holistically quantified against trade-offs in other dimensions, such as food security or prosperity. In our experience, such assessments are more difficult to operationalise cf. reductionist studies due to high levels of engagement and range of research disciplines required, but arguably more amendable to impact, because they capture more key sustainability indicators that are likely to inhibit or enable behaviour change in practice¹⁵. Use of people-centric design meant engaging end-users who would be directly or indirectly affected by the climate crisis to develop fit-for-purpose farm interventions and thematic innovation bundles to adapt to future climates and/or to mitigate GHG emissions^{15,27,28}. Such modus operandi builds end-user trust in research methods invoked, and affords researchers with the opportunity to validate model outputs while focusing on contemporary problems faced by industry²⁹.

And:

As for any scientific study, impact ultimately realised following any intervention will be influenced by manifold factors, including the rate of technology development, market prices/availability, regulation, consumer expectations and government policies. Such externalities influence practitioner behaviours and thus the extent and duration of adoption, as shown by feedback from the RRG on the credibility, legitimacy and salience of the mitigation/adaptation bundles examined here.

And:

While landscape-level assessments offer macroscopic insights on emissions mitigation potential across farms, our purpose here was to derive sustainable pathways to net-zero emissions within farms, in consultation with regional practitioners. This participatory approach helped determine the feasibility of purported interventions, validate assumptions and simulations and, through iteration, allow practitioners to gain confidence in methods we invoked. Indeed, the namesake of our study (costs of transitioning to net zero emissions) was put to us by the RRG, demonstrating that this work is demand driven. Landscape-scale assessments capturing diversity in enterprise mixes across farms and value chains, as well as carbon sequestration potential between farms⁸⁹ were out of scope of the present study, but are worthy of further investigation. Such assessments are typically more granular than farm-scale analyses, due to computational expense⁸⁹. While perceived attainment of ‘net-zero emissions’ depends on the sector or land area in question⁷, many studies agree that action at the farm level must be taken to reduce GHG emissions and avoid dangerous climate change^{77,78,82,89}.

R#2: L291: typo

Amended

R#2: What is the difference between a farmer buying land elsewhere to plant with trees, vs purchasing carbon credits from e.g. a forestry company that does the same (and might manage trees and wood better to achieve downstream mitigation from wood use?)

This is a good question, and while not the question we sought to directly address, our results do provide some insights on the pros and cons of each approach. For the beef farm, tree carbon sequestration required the purchase of new land to grow trees, which eroded profit and return on capital over the 20 year analytical period. For the sheep farm, we modelled woody-thickening associated with current bushland not used for grazing, obviating the need for purchase new land, although this scenario also eroded profit (each co-designed with the RRG). Both approaches comprise ‘insetting’, where carbon is sequestered within the farm business, and as trees mature, carbon sequestration gradually diminishes over time (see Fig. S1). In contrast, carbon offsetting, where credits are purchased externally, would not be expected to diminish in quantum over time, but would be subject to volatile carbon markets. For the farmer, the opportunity cost would be the difference between income from carbon credits (from trees) vs the highest potential income scenario (in our work, commodity based production was more economically viable than income from carbon). We made no contrasts with forestry companies generating revenue via carbon and/or timber; this comprises a different economic proposition to that assessed here, but would be worthy of future research.

R#2: L324: 25 years cannot be regarded as permanent in terms of climate mitigation! The majority of carbon markets are underpinned by consistent instruments, including the need for carbon to be additional, permanent, measurable and verifiable (see Henry et al. 2022: creating frameworks to foster soil carbon sequestration). The “permanence” period of carbon markets typically ranges from 5-100 years (e.g. see Table 1 in <https://www.tandfonline.com/doi/full/10.1080/17583004.2023.2298725>). We have clarified our usage of ‘permanence’ in this context in the paper.

The text on page 20 was modified as follows:

Management and system flexibility determines response agility to externalities, such as market volatility or climate change. Tactical and strategic farm management plans should thus encapsulate new and available technologies, together with market and climatic trends³². Tree carbon projects are typically underpinned by ‘permanence’ obligation periods of 25-100 years^{33,91}, where carbon sequestered must be prevented from re-entering the atmosphere for at least the permanence period. Future market demand for carbon credits (e.g. from the transport and energy production sectors) could lead to appropriation of land currently used for commodity production and/or higher carbon prices, which may impact on food and/or fibre supply, particularly given that markets often dictate land-use decisions⁹⁰. It is plausible that land conversion from commodity-based production to that for ecosystems services may have perverse outcomes, such as higher food prices, diminished food security or increased poverty, particularly as the population grows. Jurisdictions that prioritise carbon and/or environmental outcomes may cause carbon and/or biodiversity leakage³⁴, where commodity-based production shifts to other regions, potentially causing land clearing and loss of biodiversity in regions that have long acted as global carbon sinks or habitat assemblages (e.g., Amazon rainforest in Brazil).

R#2: L325-326: potentially consuming arable zones that go a long way towards fulfilling the growing global need for protein, fibre and starch. What does this mean?

This text has been removed in lieu of revisions articulated above.

R#2: The conclusion that individual actions have little impact on GHG emissions or productivity is not supported by the results which indicate very large GHG reductions from asparagopsis feeding and tree planting (for example).

Agree. We have revised the conclusions accordingly.

R#2: Appropriately contextualised bundles of interventions to local conditions like planting trees, renovating pastures with deep rooted legumes and adopting high FCE animal genotypes not only have the potential to reduce farm business GHG emissions to net zero, but also improved profitability and productivity gains; Did any scenarios achieve net zero without tree planting elsewhere?

Both farms required carbon sequestration from tree planting to attain net zero status (see Fig 5).

R#2: L423: typo.

Amended.

R#2: SI document

Asparagopsis: Enteric CH₄ fermentation is reduced by 80% in SB-GAF model outputs, for weaned animals. This is a massively ambitious effect.

Please see our responses above.

R#2: Transformational feed conversion efficiency: 20-30% improved FCE by

2050! Please see our responses above.

R#2: Soil carbon cycling is modelled using RothC and biochar parameters: biochar feed additive. 3% of biochar assumed to be labile manure C, 97% recalcitrant – of which 65% is carbon with a decay of <12% over 100 yrs. OK.

In general, the soil carbon changes are modest and look plausible, but a few stick out and require further justification. E.g. for the 2030 climate beef system: lucern renovation of pastures contributes a large effect of 1.36 Mg CO₂e ha⁻¹ yr⁻¹. This seems surprising. How is lucern renovation managed (I understand that lucern will crop for 4-6 years, but then the field requires replanting, which will have soil carbon costs!) Forestry leads to >30 Mg Mg CO₂e ha⁻¹ yr⁻¹ by 2030. Given realistic planting rates and slow initial tree growth curves, this seems very high and requires further explanation.

Our modelling suggests SOC sequestration rates averaged over the simulation of 0.17-0.39 Mg C ha⁻¹ yr⁻¹, which is aligned with empirical evidence reported in field studies and reviews. For example, Cotching (2018) conducted experimental studies and showed that perennial pasture on Ferrosol soils in Tasmania may sequester 98 kg C ha⁻¹ yr⁻¹, equivalent to 0.37 Mg CO₂e ha⁻¹ yr⁻¹. Scientific literature reviews of the effects of practice change on soil carbon in grasslands have measured accrual rates of 0.1-1.0 Mg C ha⁻¹ yr⁻¹ (Conant et al. 2017). Deep-rooted legumes species such as lucerne have the opportunity to grow deeper than 30 cm (25% of roots); some studies have reported lucerne roots below 100 cm (Peixoto et al 2022). To avoid adding uncertainty associated with depth and frequency of cultivation, and extent of soil carbon loss during cultivation events, we assumed pasture renovation occurred via broadcasting (zero cultivation), and have made a note of this in the methods. We have detailed our response to this reviewer's comments regarding carbon sequestration in trees in our previous responses.

Peixoto L, Olesen JE, Elsgaard L, Enggrob KL, Banfield CC, Dippold MA, Nicolaisen MH, Bak F, Zang H, Dresbøll DB, Thorup-Kristensen K, Rasmussen J. Deep-rooted perennial crops differ in capacity to stabilize C inputs in deep soil layers. Sci Rep. 2022; 12(1):5952. doi: 10.1038/s41598-022-09737-1.

Conant, R.T., Cerri, C.E.P., Osborne, B.B., Paustian, K., 2017. Grassland management impacts on soil carbon stocks: a new synthesis. Ecological Applications 27, 662-668.

R#2: Table S5 & S6: some of the pathway SOC accumulation rates seem high, e.g. 1.65 and 0.77 Mg CO₂e ha⁻¹ yr⁻¹ for TCN in 2030 or 2050, respectively. Are these entire farm averages, and if so, how are they higher than individual values displayed in Tables S1 & S2. Are some effects additive (and is this justified, e.g. through Roth-C modelling of multiple changes)?

Effects of stacking are indeed additive for many interventions (where they were combined into thematic interventions, such as TCN). This is justified, since multiple practice changes were modelled (RothC, GrassGro and FullCAM) on the same land parcel e.g. effects caused by improved FCE are added to those associated with pasture renovation; individual effects are shown in the supplementary information.

R#2: Table S10. How does asparagopsis increase ocean C sink? Please explain.

Asparagopsis is farmed in marine environments around the world; as demand for low-emissions livestock feed supplements increases in future, it would be reasonable to expect that so too will production of *Asparagopsis* spp. Some scientists have speculated that seaweed aquaculture – being one of the fastest growing components of global food production systems - can sequester up to 1,500 Mg CO₂ km⁻² yr⁻¹ due to the rapid growth of seaweed (Duarte et al. 2017). While part of the seaweed is harvested (and thus carbon is removed), residual components of seaweed remain in deep sea sediment, contributing to marine carbon stocks viz. “blue carbon” (Duarte et al. 2017).

Duarte, C. M., Wu, J., Xiao, X., Bruhn, A., Krause-Jensen, D. Can Seaweed Farming Play a Role in Climate Change Mitigation and Adaptation? *Front. Mar. Sci.* 4, (2017). <https://doi.org/10.3389/fmars.2017.00100>

We have edited Table S10 to clarify this point.

R#2: Tables S15 & S16: please specify daily live weight gain (margins > 0.01-0.025 specified, but these are low and LWG is a critical parameter determining both productivity and GHG emissions, so should be specified).

Daily weight gain is a model output, rather than a model input. Daily liveweight gain varies dynamically, according to production system, animal cohort, season, pasture digestibility and supplementary feed over the simulation duration. Aggregated liveweight values for each cohort are reported in Tables S1-S8. The liveweight values the reviewer refers to here are threshold parameters to ensure uniformity of pasture residuals across the farm and comparability of simulated treatments.

Response to reviewers

Reviewer #1 (Remarks to the Author):

The authors have revised the manuscript appropriately. I recommend acceptance of the study.

Thank you.

Reviewer #1: I still think that the title is too broad and generic but I leave this decision to you and the editors. This just a constructive feedback. The current title is "Costs of transitioning to net-zero emissions under future climates", where is this article doing the transition? In what sector? It can be a transition in the steel or cement sector from the current title. This should be clarified in my opinion. Please also be specific about 'future climates' in the title as I do not see climate projection analyses.

We revised the title to include "the livestock sector".

The revised title "*Costs of transitioning the livestock sector to net-zero emissions under future climates*" will appeal to a wide audience, ultimately benefitting the journal and the article.

Reviewer #2 (Remarks to the Author):

The authors have addressed many of the issues raised in the last round of review, and have provided a lot of clarification that really helped to understand the diverse data sets and methods employed.

Thank you for your positive feedback on our manuscript.

However, a few serious issues remain to be addressed, in particular around the overly optimistic default assumption on asparagopsis methane inhibition in grazing animals, and high (and time-limited) carbon dioxide removal values for planted trees. These two measures do much of the "heavy lifting" for the net zero pathways, thus conclusions are extremely sensitive to related assumptions. The new sensitivity analysis re asparagopsis efficacy is a step in the right direction, but remains optimistic overall (with a 40% efficacy base) and does not appear to address implications of additional tree planting areas required to achieve net zero. These implications need to be fully explained to make the sensitivity analysis meaningful. Furthermore, the default assumption used for main scenarios should also be reduced to ensure results are more credible. Some evidence around important measures such as biochar feeding is also patchy and at least needs more rigorous evidence. Overall, the discussion section needs to be tightened up from a scientific perspective, with more critical reflection on the selected level of analysis (farm enterprise), and on the definition and time frame of net zero. The plausibility of farmers buying remote land to meet net zero targets at an enterprise level did not seem high based on stakeholder

feedback, and the cost of the alternative approach of buying offsets is not clearly presented anywhere (despite being stated to be higher than enterprise action). Overall, the manuscript has merit and the ambition of the approach is commendable, but the execution and description requires some refinement.

We revised the sensitivity analysis for a second time following suggestions from this reviewer. We now examine enteric methane mitigation ranging from 10% to 99% to capture the entire plausible range in mitigation potentially realised at the whole farm level. We further examine the impact of tree plantings via sensitivity analyses by perturbing dynamic carbon sequestration growth rates by $\pm 20\%$ and add discussion on the time limitation associated with sequestration associated with planting trees. In all cases, we hinge assumptions on empirical evidence from the peer-reviewed literature (further details provided below).

We need to be clear that the purpose of the *Asparagopsis taxiformis* feed supplement intervention was to examine the change in net farm GHG emissions and profit emanating from an intervention with *transformative potential* to reduce enteric methane, as requested of us by the group of industry practitioners we co-designed interventions with (the regional reference group, or RRG). Interventions with *modest* enteric methane mitigation were examined elsewhere in this study (e.g. 30% inhibition of enteric methane with adoption of a vaccine). Reducing the quantum of enteric methane mitigation associated with *A. taxiformis* feed supplementation would reduce the difference between this intervention and that obtained from use of an enteric methane inhibitor vaccine. This would render our assessment of an intervention with transformative potential for reducing enteric methane redundant (*viz. A. taxiformis* feed supplementation).

Given that 80% mitigation of enteric methane with *A. taxiformis* feed supplementation has been measured empirically *in vivo* in several peer-reviewed studies⁴⁹⁻⁵⁴, such abatement is not only realistic, but directly aligned with the purpose of this intervention: to elicit GHG and economic implications associated with *transformative reductions* in enteric methane emissions. We note high global interest from academia, industry and policy-makers in the development of technologies aimed at *substantially* reducing enteric methane emissions; here we examine implications of such technology at the whole farm level. We substantiate the text in the manuscript in this vein.

We conducted further sensitivity analyses based on this reviewer's commentary (Figs S3-S6). The analysis shows that net farm GHG and profitability are more sensitive to assumptions pertaining to enteric methane mitigation than temporal carbon sequestration in vegetation. In particular, the relative impact of the adaptation/mitigation bundles on net farm GHG and enterprise profit across a range of enteric methane mitigation values remained similar, evidencing robust conclusions. We detail sensitivity analyses below and add further results to the supplementary information (Figs S3-S6).

We articulated further evidence relating to biochar as a feed supplement, our definition of net-zero, clarity on stakeholder feedback relating to adaptation/mitigation bundles (including that pertaining to purchasing remote land to diversify enterprise climate exposure), and transparency underpinning methods adopted. We note that the RRG provided positive and negative feedback on all

interventions, as requested of them during co-design workshops. Interventions shown here were ultimately perceived by the RRG as prospective practices, infrastructure and/or technologies warranting further investigation, altogether considering adoptability, profit, production and GHG emissions mitigation potential.

Specific comments

Reviewer #2: Agriculture” or “Farms” or “Farming” or similar should be mentioned in the title.

See response to first reviewer.

Reviewer #2: L42-43: did the UN FCCC from 1992 already target stabilised concentrations, or emissions? Better to start with emission reduction targets of Kyoto, and delete former lines to save space.

We revised this text to better clarify the goals of the UNFCCC in 1992, then transition into the focus on emissions reduction introduced by the Kyoto Protocol in line with this reviewer’s opinion.

Reviewer #2: L45-46: the terms “carbon neutrality” and “climate neutrality” are implicitly distinguished, without elaborating what either of them means. A clear definition of carbon neutrality is crucial for this manuscript.

Carbon neutrality refers to net zero carbon emissions, while climate neutrality refers to net zero global temperature change.

The text was revised as follows:

‘Carbon neutrality’ refers to the balancing of GHG emissions with removals⁷, measured in terms of CO₂-equivalents. In contrast, ‘climate neutrality’ refers to net-zero global atmospheric temperature change⁷.

Reviewer #2: L63: numerous studies have looked at stacked interventions. More accurate to say that few have studied the combinations of mitigation actions and CDR needed to achieve net zero at farm enterprise level.

Amended as follows:

“Few studies have explored holistic implications of bundling practices simultaneously aimed at carbon dioxide removal, greenhouse gas emissions reduction and climate change adaptation”

R#2: L112-113: Presumably only climate data are for 1986-2005, and baseline farm performance reflects more recent performance? Please clarify

Climate data and baseline farm performance were both based on 1986-2005. This was a deliberate decision to enable consistency in the comparison of simulations for each intervention (profit, production etc) with the same duration for the baseline period. The period 1986-2005 was set following projections from global climate models which use the said period as a baseline. The methods have been clarified.

R#2: L128 and Fig. S3: The authors have included sensitivity analyses around the efficacy of asparagopsis in figures S3 and S4. However, the main results still appear to be based on the extremely ambitious assumption of an 80% reduction. The references used to justify this appear to be based on continuous indoor feeding. Recent work on grazing animals pertinent to the systems being modelled indicates much lower efficacy of methane inhibitors (in region of 10% emission reduction). Using 80% as a central estimate in the main analysis for this major emission source undermines the results of this study. A more conservative and defensible estimate for grazing systems should be used here. Sensitivity analysis should also start from a much lower minimum, e.g. 10%, rather than 40%.

We revised the sensitivity analysis for a second time following feedback from this reviewer. We now examine enteric methane mitigation ranging from 10% to 99% to capture the entire range in mitigation potentially realised at the whole farm level. We further examine the impact of tree plantings via sensitivity analyses by perturbing dynamic carbon sequestration growth rates by $\pm 20\%$ and add discussion on the time limitation associated with sequestration associated with planting trees. In all cases, we hinge assumptions on empirical evidence from the peer-reviewed literature (further details provided in responses below).

We need to be clear that the purpose of the *Asparagopsis taxiformis* feed supplement intervention was to examine the change in net farm GHG emissions and profit emanating from an intervention with *transformative potential* to reduce enteric methane, as requested of us by the group of industry practitioners we co-designed interventions with (the regional reference group, or RRG). Interventions with *modest* enteric methane mitigation have already been examined elsewhere in this study (e.g. 30% inhibition of enteric methane with adoption of a vaccine). Reducing the quantum of enteric methane mitigation associated with *A. taxiformis* feed supplementation would reduce the difference between this intervention and that obtained from use of an enteric methane inhibitor vaccine. This would make the modelling *A. taxiformis* feed supplementation redundant.

Given that 80% mitigation of enteric methane with *A. taxiformis* feed supplementation has been measured empirically *in vivo* in several peer-reviewed studies⁴⁹⁻⁵⁴, such abatement is not only realistic, but directly aligned with the purpose of this intervention: to elicit GHG and economic implications associated with *transformative reductions* in enteric methane emissions. We note high global interest from academia, industry and policy-makers in the development of technologies aimed at substantially reducing enteric methane emissions; here we examine implications if such technology were realised at the whole farm level.

The lower threshold of the sensitivity analyses was reduced based on this reviewer's commentary. The revised analysis shows that net farm GHG and profitability are more sensitive to assumptions pertaining to enteric methane mitigation than temporal carbon sequestration in vegetation. In particular, the relative impact of the adaptation/mitigation bundles on net farm GHG and enterprise profit across a range of enteric methane mitigation values remained similar, evidencing robust conclusions. We detail sensitivity analyses below and add further results to the supplementary information (Figs S3-S6).

- ⁴⁹ Kinley R. D., *et al.* Mitigating the carbon footprint and improving productivity of ruminant livestock agriculture using a red seaweed. *J. Clean Prod.* **259**, 120836 (2020). <https://doi.org/10.1016/j.jclepro.2020.120836>
- ⁵⁰ Machado, L., Magnusson, M., Paul, N.A., de Nys, R., Tomkins, N. Effects of Marine and Freshwater Macroalgae on In Vitro Total Gas and Methane Production. *PLoS ONE* **9**(1): e85289 (2014). <https://doi.org/10.1371/journal.pone.0085289>
- ⁵¹ Li, X., Norman H. C., Kinley, R. D., Laurence, M., Wilmot, M., Bender, H., de Nys, R., Tomkins, N.. *Asparagopsis taxiformis* decreases enteric methane production from sheep. *Animal Production Science* **58**, 681-688 (2018). <https://doi.org/10.1071/AN15883>
- ⁵² Roque, B.M., *et al.* Red seaweed (*Asparagopsis taxiformis*) supplementation reduces enteric methane by over 80 percent in beef steers. *PLoS ONE* **16**(3): e0247820 (2021). <https://doi.org/10.1371/journal.pone.0247820>
- ⁵³ Vijn, S. *et al.* Key Considerations for the Use of Seaweed to Reduce Enteric Methane Emissions From Cattle. *Front. Vet. Sci.* **7** (2020). <https://doi.org/10.3389/fvets.2020.597430>
- ⁵⁴ Wasson, D. E., Yarish, C., Hristov, A. N. Enteric methane mitigation through *Asparagopsis taxiformis* supplementation and potential algal alternatives. *Front. Anim. Sci.* **3**, (2022). <https://doi.org/10.3389/fanim.2022.999338>

The methods were revised as follows:

Sensitivity of enteric methane mitigation and vegetation carbon sequestration

The mitigation we attributed to the use of red seaweed (*Asparagopsis taxiformis*) as a feed supplement (80%) was designed to examine the impacts of an intervention with transformational potential for enteric methane inhibition. Under controlled conditions, enteric CH₄ reductions of 80-99% have been observed using *A. taxiformis*⁴⁹⁻⁵⁵. In grazing systems however, CH₄ abatement can be more variable due to differential forage quality and supplement intake between animals and seasons^{18,53}. To quantify the impact of variability in enteric methane inhibition and carbon sequestration in vegetation on net farm GHG and profit, we conducted sensitivity analyses. Following ranges reported in peer-reviewed literature, methane mitigation was varied from 10% to 99%⁵⁴, while temporal carbon sequestration simulated using FullCAM was perturbed by ±20% based on 95% confidence intervals of applicable tree species in temperate regions⁹⁵.

Details relating to sensitivity analysis of tree carbon sequestration are articulated at the end of our responses (below).

R#2: L148-151: Good to see the additional clarification and references here - the FCE gain over the time period in question seems plausible.

Thank you.

R#2: Fig. 1 & 2: where is the measures of buying external carbon offsets on the mitigation cost charts? This is mentioned to be expensive elsewhere (e.g. L327), but no data shown?

Costs associated with purchasing external carbon credits to offset farm GHG emissions are shown in Fig. 5g and 5h (light blue bars). We also clarified the text.

R#2: L176-178: Confusion between opinions of RRG and scientifically-derived mitigation performance in this sentence. Presumably RRG provided information regarding adoptability only? It seems from next sentences that LWG, enteric CH₄ and SOC effects were based on an as-yet unpublished experiment, an article from 2012 published in non-ISI journal (Livestock Research for Rural Development), a non-peer-reviewed report. This is not robust evidence. This needs better justification or else modification.

The distinction between RRG feedback and peer-reviewed evidence has been clarified.

The manuscript was revised as follows:

The RRG considered biochar feed supplementation as highly adoptable based on ease of implementation and comparison with other interventions. Grounding liveweight gains and enteric methane mitigation on peer-reviewed evidence^{31-33,96} (Fig. S2), we showed that biochar feed supplementation reduced net GHG emissions by 8% and increased profit of the cattle enterprise by 18% (Fig. 1 c, d), but reduced profit of the sheep enterprise by 10% (Fig. 2 c, d).

³¹ Bilotto, F., Christie-Whitehead, K.M., Barnes, N., Harrison, M.T. Operationalising net-zero with biochar: Black gold or red herring? *Trends Food Sci. Technol.* **150**, 104579 (2024). <https://doi.org/10.1016/j.tifs.2024.104579>

³² Winders, T.M., Jolly-Breithaupt, M.L., Wilson, H.C., MacDonald, J.C., Erickson, G.E. and Watson, A.K. Evaluation of the effects of biochar on diet digestibility and methane production from growing and finishing steers. *Transl. anim. sci.* **3**, 775-783 (2019). <https://doi.org/10.1093/tas/txz027>

³³ Cabeza, I., Waterhouse, T., Sohi, S. and Rooke, J.A. Effect of biochar produced from different biomass sources and at different process temperatures on methane production and ammonia concentrations in vitro. *Anim. Feed Sci. Technol.* **237**, 1-7 (2018). <https://doi.org/10.1016/j.anifeedsci.2018.01.003>

⁹⁶Lind, V., Sizmaz, Ö., Demirtas, A., Sudagidan, M., Weldon, S., Budai, A., O'Toole, A., Miladinovic, D.D. and Jørgensen, G.M. Biochar effect on sheep feed intake, growth rate and ruminant in vitro and in vivo methane production. *Animal* **18**, 101195 (2024).
<https://doi.org/10.1016/j.animal.2024.101195>

R#2: L197-209: It seems that RRG feedback was somewhat negative on numerous interventions here (buying remote land, diversifying into grapes and wind turbines), yet the authors assert that these options are worth considering. Why is the “extra farm” in Fig. 1b not “low” ease of adoption in light of these comments?

The RRG provided positive and negative comments on all interventions as requested of them during our co-design workshops. Interventions shown here were ultimately perceived by the RRG as prospective interventions warranting further investigation, considering adoptability, profit, production and GHG emissions mitigation potential.

We agree that the RRG contended that purchasing external land for climate diversification would have low adoptability and have revised Fig. 1b accordingly.

We revised the paper as follows:

Our participatory workshops with the RRG co-designed and co-refined interventions, including those pertaining to income diversification. For example, the RRG indicated that purchasing additional farmland to diversify enterprise climate exposure (north-eastern Tasmania, 400 km away from the existing beef cattle farm in north-western Tasmania) would require additional labour, costs of transporting cattle between regions, infrastructure on the new land, and higher management coordination across regions. As such, the RRG opined that this intervention would be difficult to manage over the long-term (Fig. 1 a, b). The RRG contended that establishment of an irrigated grapevine enterprise would require specialist input, given disparate skillsets and knowledge requirements for managing vineyards compared with livestock production (Fig. 2 a, b), including the need for micrometeorological and soil data surveys to identify suitable locations for the vineyard on farm. The RRG suggested that installing wind turbines would require proximity with three-phase powerlines (to feed electricity generated into the main grid) as well as consistent and high prevailing winds (e.g. coastal regions). We suggest that interventions enabling climatic or enterprise diversification could be generically adapted to any production system or agroecological zone, and are very much worthy of further investigation.

R#2: L261: Should be 38-87% of “gross” emissions? Net emissions implies offsets already subtracted.

Amended.

R#2: L278: unclear what adoption of mitigation %s refer to...

This text has been removed.

R#2: L314-316: I don't think it is sufficient to simply say that landscape scale analyses were outside the scope of this study. More critical reflection on whether landscape analyses would yield different conclusions to farm-enterprise level analysis is needed.

In addition to the previous text we added in response to this reviewer's commentary, we revised the text in the manuscript as follows:

While landscape-level assessments offer macro insights on emissions mitigation across farms, our purpose here was to co-design pathways for reducing GHG emissions and adapting to the changing climate in consultation with regional practitioners. This participatory approach enabled refinement and validation of farming systems assumptions and, through iteration, allowed practitioners to gain confidence in methods we invoked. Indeed, the namesake of our study (costs of transitioning to net zero emissions) was put to us by the RRG, demonstrating the demand-driven nature of this work. Landscape-scale assessments capturing diversity in enterprise mixes across farms and value chains are typically more granular than farm-scale analyses due to computational expense³⁷. While perceived 'climate resilience' or attainment of 'net-zero emissions' depends on the sector, land area and time frame in question⁷, many scholars agree that farm-level interventions must be taken to reduce GHG emissions to avoid dangerous climate change^{15,37-39}.

Transitioning agricultural enterprises to net zero emissions is a different question to transitioning landscapes to net zero emissions. Adoption of the LHF intervention at the landscape scale may result in greater food security, assuming benefits at the farm level were realised ceteris paribus at the landscape level. Increased livestock supply on the market may reduce local prices due to trade-offs between supply and demand⁴⁰. Implications of the TCN bundle at the farm scale may have similar implications for food security and market prices if adopted at the landscape level, although the latter may also influence carbon prices. For example, assuming widespread concurrent enrolment in carbon markets with adoption of the TCN intervention, available carbon credits on the market would fall, increasing carbon prices (similar to economic responses to other commodities). This may prohibit market entry for new practitioners and favour those with greater purchasing power

and/or access to financial capital. Adoption of renewable energy – such as wind turbines and/or agrivoltaics – are unlikely to be adopted en masse due to geographical prerequisites for establishing such interventions, including consistent, prevailing winds (wind turbines) or high probability of sunshine hours (agrivoltaics), as well as proximity with three-phase powerlines. In the same vein, the transformative interventions we examined are unlikely to be adopted at the landscape scale in a short period. Everett Rogers famous 1962 theorem of the diffusion of innovation suggests that adoption tends to be normally distributed, first with a few pioneering innovators, followed by the early adopters, early majority, late majority and finally the laggards⁴¹. Stacking or bundling of interventions as per the TCN, Income Diversification and Transformative interventions is arguably more difficult to realise – evidenced by Fig. 1 and 2 – implying lower rates of adoption given additional knowledge, labour and practical requirements to successfully implement, refine and benefit from such interventions. Landscape level implications pertaining to adoption of interventions examined here would also require assessment of other socio-economic factors, using for example agent-based modelling to account for social learning between peers, as well as economic, regulatory and environmental interactions influencing land use⁴². Assumptions used to invoke such frameworks and their consistency with approaches used in the present study (herd/flock dynamics, pasture growth, carbon sequestration etc) would dictate whether or not landscape scale assessments yielded similar conclusions to those drawn here.

⁴¹Muleke, A., Harrison, M. T., Yanotti, M. & Battaglia, M. Yield gains of irrigated crops in Australia have stalled: the dire need for adaptation to increasingly volatile weather and market conditions. *Current Research in Environmental Sustainability* 4, 100192 (2022).

[https://doi.org:https://doi.org/10.1016/j.crsust.2022.100192](https://doi.org/https://doi.org/10.1016/j.crsust.2022.100192)

⁴²Rogers, Everett M (1962) *Diffusion of Innovations*. Third Edition. The Free press, New York.

⁴³Shahpari, S., Allison, J., Harrison, M. T. & Stanley, R. An integrated economic, environmental and social approach to agricultural land-use planning. *Land* 10, 1-18 (2021).

<https://doi.org:10.3390/land10040364>

R#2: L334: tree areas should be presented as % farm areas to provide proper context

Amended as suggested:

...stacking together interventions improved pasture growth and soil carbon sequestration, and adopting superior animal genotypes with greater liveweight gain for the same/less feed intake (FCE, TFCE), along with planting small proportions of farms with trees (5-15%) went considerable way towards negating farm emissions...

R#2: L370-371: The author response regarding permanence of carbon “insetting” requires more critical insight, given that the lower end of the scale suggested, 25 years, implies serious lack of durability to “net zero” solutions identified. This is an issue for critical discussion.

The 25 or 100 year time scale is stipulated by those who conceive policies for carbon insetting, not by the authors. We assume the reviewer is referring to the duration with which GHG emissions mitigation vs carbon sequestration can occur (carbon reductions and removals following IPCC terminology, respectively), as there appears to be confusion between “permanence” parlance used in policy instruments and that realised by interventions modelled.

In many carbon markets, permanence obligations require project proponents to maintain the carbon stored or sequestered for a minimum period. For the Australian Government, this period is either 25 or 100 years, with discounts on carbon credits applied to projects enrolled in the 25-year period due to greater risk of carbon reversal (that is, the carbon sequestered reenters the atmosphere if trees are burnt by fire). The proponent (landholder or other) chooses the duration of the project at the outset. In terms of the reviewer’s comment, the “durability” of the project in this case can only be 25 or 100 years. The project period is fixed by the regulator (not by the authors of this paper). Further details are here: <https://cer.gov.au/schemes/australian-carbon-credit-unit-scheme/how-to-participate/permanence-obligations>

Enduring physical “permanence” - wherein GHGs are perpetually prevented from returning to the atmosphere - is an entirely different prospect to “permanence” parlance used in carbon policies. We assume the reviewer refers to the former. The time with which GHGs are removed or prevented from entering the atmosphere depends on the *modus operandi* of the intervention and GHG in question. Interventions that mitigate enteric methane emissions – such as feed supplementation with *A. taxiformis* – permanently prevent methane that would have otherwise been eructed from entering the atmosphere, assuming all other aspects of the system (stocking rates, liveweight gain, age of sale) remain unchanged. This methane is permanently avoided from entering the atmosphere. In the same vein, adopting animals with genetics that afford greater feed-conversion efficiency can permanently avoid enteric methane, assuming animals with greater FCE are sold at the same liveweight and earlier than the *ceteris paribus* system. Interventions that sequester additional carbon in soils and/or vegetation, such as renovation of pastures with lucerne or planting

trees are however characterised by diminishing longitudinal carbon sequestration (Fig. S1). Assuming other aspects remain unchanged, annual carbon emissions would be expected to vary around some constant value, while carbon removals would diminish as trees approach maturity, making prospects of attaining net zero increasingly difficult with the passage of time. This could be countered in many ways, for example via agroforestry, where portions of farm area are sequentially sown then harvested for timber as trees approach maturity. Provided carbon in harvested timber was not permitted to re-enter the atmosphere (e.g. use of timber in construction materials), longitudinal carbon removals of the farm business would no longer plateau, as some plantations would always be approaching, or in, periods of peak growth. Such asynchronous carbon removals would continue until such time as the harvested timber were burnt or decomposed. We suggest that farming systems approaches that afford such asynchronous temporal carbon sequestration, together with practices that reduce methane emissions (such as *A. taxiformis* feed supplements) in concert with practices that avoid methane emissions (improving animal growth rates and earlier sales) or CO₂ (renewable energy on farm), will be increasingly called for in future, particularly if farms are mandated to reduce GHG emissions.

We integrated the above into the paper.

R#2: L379: would higher food prices be a perverse outcome? Or reflect a “fair price” that internalises the many externalities of current food production?

This text has been removed.

R#2: L398-412: The text doesn't accurately reflect the extremely optimistic default assumption of 80% enteric CH₄ mitigation efficacy for asparagopsis (mentioned previously). A major impact of reduced efficacy is not achieving net zero with the listed measures, even for the CN scenarios (as shown in Figs S3 and S4). Presumably achieving net zero at 40 or 60% efficacy would require a lot more tree planting (and thus cost). This is so important for results it needs more rigorous explanation.

We revised the main text and methods to underscore this point. We discuss implications of diminishing sequestration later in the article (see responses below).

*The Asparagopsis taxiformis feed supplement intervention was examined through the lens of a transformative intervention for enabling deep cuts in enteric CH₄ emissions, with the 80% enteric methane mitigation value calibrated based on empirical evidence from several peer-reviewed studies⁴⁹⁻⁵⁴. Use of *A. taxiformis* in this way showed significant promise at the whole farm scale, decreasing net farm GHG emissions by 46-72%. Although some in vitro studies report greater CH₄ inhibition than we assumed (up to 98% enteric CH₄ reduction^{49,50,54}), enteric CH₄ mitigation in*

grazing systems may be more modest due to animal access to feed supplements, foraging behaviours, seasonal variation in forage quality, diet composition and other enterprise constraints in situ^{18,53}. As such, to determine the sensitivity of net farm GHG emissions and enterprise profit associated with enteric CH₄ mitigation, we conduct sensitivity analyses with enteric methane mitigation ranging from 10% to 99%^{52,54} (Figs S3 and S4). For the beef farm, net emissions ranged from 1,484-1,709 Mg CO₂e with a 99% CH₄ reduction, to 3,533-3,761 Mg CO₂e with 10% CH₄ reduction, corresponding to profit after carbon tax of AUD 321-340K and AUD 157-176K, respectively (Figs. S3 and S5). For the sheep farm, net GHG varied from 596-871 Mg CO₂e (99% CH₄ reduction) to 4,726-4,997 Mg CO₂e (10% CH₄ reduction), with profit ranging between AUD 1,088-1,113K and AUD 758-783K (Figs. S4 and S6). This analysis demonstrates that adaptation/mitigation bundles are sensitive to the quantum of CH₄ reduction, and while we acknowledge that enteric CH₄ emissions in any production system will vary between animals, seasons and farms, these results show farm-level implications should some transformative intervention for reducing enteric CH₄ eventually be realised. If enteric CH₄ mitigation were lower than assumed here, further measures would be required to negate residual GHG emissions. One way for this may be through additional tree plantations. If C sequestration were improved by 20%, enteric CH₄ mitigation required to realise net-zero status would fall from 80% to 60% for the beef farm (CN2 and CN4) and from 80% to 70% for the sheep farm (CN3 and CN4). If tree C sequestration were 20% lower, either enteric CH₄ reduction would need to be greater than 80% or further measures would be necessary to achieve carbon neutrality. Taken together, these results demonstrate that sustained maintenance of net-zero emissions for livestock businesses will be challenging. Any aspiration for GHG abatement would need to be conducted by combining a range of technologies, practices and infrastructure for carbon reduction and removals (mitigation, avoidance, sequestration and offsetting) given that sequestration in soils and vegetation tends to diminish over time.

To the methods, we added:

Sensitivity of enteric methane mitigation and vegetation carbon sequestration

The mitigation we attributed to the use of red seaweed (*Asparagopsis taxiformis*) as a feed supplement (80%) was designed to examine the impacts of an intervention with transformational potential for enteric methane inhibition. Under controlled conditions, enteric CH₄ reductions of 80-99% have been observed using *A. taxiformis*⁴⁹⁻⁵⁵. In grazing systems however, CH₄ abatement can be more variable due to differential forage quality and supplement intake between animals and

seasons^{18,53}. To quantify the impact of variability in enteric methane inhibition and carbon sequestration in vegetation on net farm GHG and profit, we conducted sensitivity analyses. Following ranges reported in peer-reviewed literature, methane mitigation was varied from 10% to 99%⁵⁴, while temporal carbon sequestration simulated using FullCAM was perturbed by $\pm 20\%$ based on 95% confidence intervals of applicable tree species in temperate regions⁹⁵.

In the supplementary material, we replaced the previous Figures S3 and S4 with four new plots that account for both time horizons (2030 and 2050) in the beef and sheep farms. These plots expand on the methane reduction potential from feeding *Asparagopsis* (10-99%) and incorporate uncertainty in carbon sequestration by trees (-20% and +20%) as follows:

Fig. S3. Sensitivity of net farm GHG emissions to mitigation of *Asparagopsis taxiformis* (10-99% CH₄ reduction) as individual intervention, combined with planting trees (sensitivity of vegetation C sequestration) and stacked in carbon neutral packages for a beef cattle production system. Carbon neutral packages were attained by improving animal genetics (CN1 and CN2) or renovating pasture swards with lucerne (CN3 and CN4) in 2030 and 2050 climates. Mod C seq: modelled C sequestration in planting trees; +20 C seq: 20% increase in modelled C sequestration; -20 C seq: 20% decrease in modelled C sequestration; ASP: *A. taxiformis* as livestock feed supplement; ASP+PT: *A. taxiformis* + planting trees (50 ha); CN1: carbon neutral package 1 [*A. taxiformis* + planting trees (50 ha) + transformational feed conversion efficiency]; CN2: carbon neutral package 2 [*A. taxiformis* + planting trees (55 ha in 2030 and 110 ha in 2050) + transformational feed conversion efficiency]; CN3: carbon neutral package 3 [*A. taxiformis* + planting trees (50 ha) +

Lucerne]; CN4: carbon neutral package 4 [*A. taxiformis* + planting trees (55 ha in 2030 and 110 ha in 2050) + Lucerne].

Fig. S4. Sensitivity of net farm GHG emissions to mitigation of *A. taxiformis* (10-99% CH₄ reduction) as individual intervention, combined with planting trees (sensitivity of vegetation C sequestration) and stacked in carbon neutral packages for a sheep production system. Carbon neutral packages were attained by improving animal genetics (CN1 and CN2) or renovating pasture swards with lucerne (CN3 and CN4) in 2030 and 2050 climates. Mod C seq: modelled C sequestration in planting trees; +20 C seq: 20% increase in modelled C sequestration; -20 C seq: 20% decrease in modelled C sequestration; ASP: *A. taxiformis* as livestock feed supplement; ASP+PT: *A. taxiformis* + planting trees (200 ha); CN1: carbon neutral package 1 [*A. taxiformis* + planting trees (200 ha) +

transformational feed conversion efficiency]; CN2: carbon neutral package 2 [*A. taxiformis* + planting trees (220 ha) + transformational feed conversion efficiency]; CN3: carbon neutral package 3 [*A. taxiformis* + planting trees (200 ha) + Lucerne]; CN4: carbon neutral package 4 [*A. taxiformis* + planting trees (220 ha) + Lucerne].

Fig. S5. Sensitivity of costs of transitioning to net zero associated with mitigation of *A. taxiformis* (10-99% CH₄ reduction) as individual intervention, combined with planting trees (sensitivity of vegetation C sequestration) and stacked in carbon neutral packages for a beef cattle production system. Carbon neutral packages were attained by improving animal genetics (CN1 and CN2) or renovating pasture swards with lucerne (CN3 and CN4) in 2030 and 2050 climates. Mod C seq: modelled C sequestration in planting trees; +20 C seq: 20% increase in modelled C sequestration; -20 C seq: 20% decrease in modelled C sequestration; ASP: *A. taxiformis* as livestock feed supplement; ASP+PT: *A. taxiformis* + planting trees (50 ha); CN1: carbon neutral package 1 [*A. taxiformis* + planting trees (50 ha) + transformational feed conversion efficiency]; CN2: carbon

neutral package 2 [*A. taxiformis* + planting trees (55 ha in 2030 and 110 ha in 2050) + transformational feed conversion efficiency]; CN3: carbon neutral package 3 [*A. taxiformis* + planting trees (50 ha) + Lucerne]; CN4: carbon neutral package 4 [*A. taxiformis* + planting trees (55 ha in 2030 and 110 ha in 2050) + Lucerne].

Fig. S6. Sensitivity of costs of transitioning to net zero associated with *A. taxiformis* mitigation (1099% CH₄ reduction) as individual intervention, combined with planting trees (sensitivity of vegetation C sequestration) and stacked in carbon neutral packages for a sheep production system. Carbon neutral packages were attained by improving animal genetics (CN1 and CN2) or renovating pasture swards with lucerne (CN3 and CN4) in 2030 and 2050 climates. Mod C seq: modelled C sequestration in planting trees; +20 C seq: 20% increase in modelled C sequestration; -20 C seq: 20% decrease in modelled C sequestration; ASP: *A. taxiformis* as livestock feed supplement; ASP+PT: *A. taxiformis* + planting trees 200 ha); CN1: carbon neutral package 1 [*A.*

taxiformis + planting trees (200 ha) + transformational feed conversion efficiency]; CN2: carbon neutral package 2 [*A. taxiformis* + planting trees (220 ha) + transformational feed conversion efficiency]; CN3: carbon neutral package 3 [*A. taxiformis* + planting trees (200 ha) + Lucerne]; CN4: carbon neutral package 4 [*A. taxiformis* + planting trees (220 ha) + Lucerne].

R#2:

Methods

The methods still lack a basic definition of the time frame over which “net zero” or “carbon neutrality” was defined, and how. This appears to be a balance in all GHGs, based on GWP100 values, at the year 2030 and 2050?

Net farm GHG emissions were based on GWP100 computed on an annual basis using SB-GAF as the sum of carbon sequestration in soils and vegetation with GHG emissions (sequestration being negative; CO₂, CH₄ and N₂O emissions being positive). Annual net GHG emissions reported here represent 20-year averages for each of the historical, 2030 and 2050 climate horizons (viz. Tables S3-S10). Net-zero GHG emissions was defined as the point at which annual net GHG emissions averaged over each 20-year simulation equated to zero.

We added the text above to the ‘Net farm greenhouse gas emissions’ section in the methods.

R#2: Fig. S1: The authors have elaborated the tree C sequestration values derived from the model they ran, and propose that 8 t C / ha / yr is a reasonable average for eucalypts in Tasmania. However, this is over 25% higher than eucalyptus growth rates cited for southern Australia. Data in Fig. S1 are presumably from the model runs (clarification needed). Panel (a) displays an annual C accumulation of up to 15 t C / ha for beef farms. This would equate to >60 m³ /ha annual peak growth! Possible Mg C has been confounded with Mg biomass dry matter on the right hand side axis of the graph? Further clarification and justification is needed for the growth rates assumed.

Our simulated sequestration values in Fig S1 are consistent with measured values published for temperate zones in southern Australia, evidenced by empirical data from peer-reviewed literature (figure caption clarified). For the beef farm, we assumed Eucalyptus spp. were planted, which are prolific in the region. For the sheep farm, we assumed plantings of endemic tree species within regions of the farm where existing tree vegetation already comprised less than 50% of the canopy (woody thickening).

We demonstrate that published values of tree growth for various tree ages align well with those simulated, giving confidence that our results are robust (see Fig. S1, Tables S1, Table S2):

1. The beef farm is situated in a region with high and consistent seasonal rainfall (807 ± 139 mm) with fertile free draining clays (Ferrosols) conducive to high growth rates. We note, too, that simulated peak growth rates are transient, peaking 10-15 years after planting, then later subsiding (Fig S1). Our simulated C sequestration rates for both regions align well with

empirical data for the same species compositions and climatic zones (Fig. S1, Table S1 and Table S2).

2. We show that simulated and measured tree carbon sequestration for the sheep farm is significantly lower than that for the beef farm (cf. Figs S1a and b), such that all simulated values for the sheep farm are less than 5 t C/ha/year. These values are also consistent with tree growth in low rainfall zones (440-600 mm/year) in published literature (Hobbs et al. 2013; Neumann et al. 2011).
3. Our annual average CO₂ sequestration rates per hectare over 20 years are supported by empirical evidence for *Eucalyptus* spp. grown in temperate climates. In a review of the global literature, Bernal et al. (2018) report average sequestration rates of 37.9 Mg CO₂ ha⁻¹ yr⁻¹ (half confidence interval of 5.5 Mg CO₂ ha⁻¹ yr⁻¹) *Eucalyptus* spp. grown in temperate climates. These values align with our simulations (Fig. S1).
4. In terms of volumetric growth. The FAO (2001) show that *Eucalyptus globulus* in Tasmania can have a Mean Annual Increment (MAI) of 35 m³ ha⁻¹ yr⁻¹, with peak growth at around 10-15 years. While MAI represents the average annual biomass growth, the Current Annual Increment (CAI) reflects instantaneous biomass flux. Using data from the FAO (2001), we calculate CAI of 55-64 m³ ha⁻¹ yr⁻¹, of which quanta are aligned with other published values of destructively-sampled aboveground biomass (Hobbs et al., 2013).
5. To examine the impact of new tree plantings (not growth of existing vegetation) on net farm GHG emissions and profitability, we conducted further sensitivity analyses, wherein tree carbon sequestration was varied by ±20%. Compared with enteric methane, we show that tree carbon sequestration had little impact on either net farm GHG nor profit when stacked with other practices, with the relative order of stacked interventions remaining similar (Figs S3-S6).

Taken together, the peer-reviewed empirical data and sensitivity analyses demonstrate that interpretations we draw are defensible.

Bernal B, Murray LT, Pearson TRH. Global carbon dioxide removal rates from forest landscape restoration activities. *Carbon Balance and Management* 13, 22 (2018).
<https://doi.org/10.1186/s13021-018-0110-8>

FAO (2001). Mean annual volume increment of selected industrial forest plantation species by L Ugalde & O Pérez. Forest Plantation Thematic Papers, Working Paper 1. Forest Resources Development Service, Forest Resources Division. FAO, Rome.
<https://www.fao.org/4/AC121E/ac121e04.htm#bm04.3>

Hobbs T, Neumann C, Tucker M, Ryan K. Carbon sequestration from revegetation: South Australian agricultural regions. *Adelaide: Department of Environment, Water and Natural Resources, The Government of South Australia & Future Farm Industries Cooperative Research Centre*, (2013).

Neumann CR, Hobbs TJ, Tucker M. Carbon sequestration and biomass production rates from agroforestry in lower rainfall zones (300-650 mm) of South Australia: Southern Murray-Darling Basin Region. *Adelaide & Future Farm Industries Cooperative Research Centre* 32, (2011).

Fig. S1. Carbon stocks by fraction (trees, debris, and soil; left axes) and annual carbon flux (solid lines, right axes) for tree plantations on the beef (a) and sheep (b) case study farms with annual precipitation >660 mm and between 400 mm and 660 mm, respectively. White circles depict field observations derived from Hobbs et al.³ and Neumann et al.⁴ of carbon sequestered by trees grown in temperate regions of southern Australia (Tables S1 and S2).

Table S1. Measured carbon sequestration of tree vegetation with recorded ages in temperate zones of southern Australia having more than 660 mm/year precipitation. Plantations comprise more than 50% Eucalyptus spp. Adapted from Hobbs et al.³ and Neumann et al.⁴

Species	Precipitation (mm/year)	Average canopy age	C seq (Mg C ha ⁻¹ yr ⁻¹)	CO ₂ e seq (CO ₂ e ha ⁻¹ yr ⁻¹)	Location
E. viminalis ssp. cygnetensis (80.1%), E. ovata var. (9.3%), E. camaldulensis var. (3.9%), E. cladocalyx (3.2%), E. obliqua (2.8%), Ac. dodonaeifolia (0.3%), Mel. gibbosa (0.2%), Al. verticillata (0.1%), Ac. retinodes (0.1%), Mel. uncinata (<0.1%)	660	11.6	12.6	46.1	Kangaroo Island
E. camaldulensis/viminalis , Ac. Retinodes	843	12.9	7.4	27.1	Adelaide & Mt Lofty Ranges
Eucalyptus globulus ssp. Globulus	826	13.9	13.8	50.6	Adelaide & Mt Lofty Ranges
E. camaldulensis var. (99.2%), Mel. decussata (0.5%), Mel. gibbosa (0.2%), Mel. uncinata (0.1%), Callistemon rugulosus (<0.1%)	660	14.6	12.3	45.2	Kangaroo Island
E. camaldulensis , Ac. retinodes	843	28.0	8.4	30.4	Adelaide & Mt Lofty Ranges
E. camaldulensis var. camaldulensis (95.0%), Banksia marginata (3.7%), Ac. melanoxylon (1.3%)	727	34.0	2.8	10.2	Adelaide & Mt Lofty Ranges
Eucalyptus leucoxydon ssp.	672	96.9	2.2	8.1	Adelaide & Mt Lofty Ranges

E. goniocalyx ssp. goniocalyx (56.6%), E. camaldulensis var. camaldulensis (28.2%), E. fasciculosa (10.8%), Ac. pycnantha (2.9%), Al. verticillata (1.4%)	727	120.0	2.2	8.0	Adelaide & Mt Lofty Ranges
--	-----	-------	-----	-----	----------------------------

Table S2. Measured carbon sequestration of tree vegetation with recorded ages in temperate zones of southern Australia having 400 - 660 mm/year precipitation. Plantations comprise less than 50% Eucalyptus spp. and native species. Adapted from Hobbs et al.³ and Neumann et al.⁴

Species	Precipitation (>400 & <660 mm/year)	Average canopy age	Cseq (Mg C ha ⁻¹ yr ⁻¹)	CO ₂ e seq (CO ₂ e ha ⁻¹ yr ⁻¹) ₁	Location
Eucalypt species present Ac. mearnsii (55.0%), Ac. melanoxylon (19.7%), E. viminalis ssp. cygnetensis (14.0%), E. leucoxyton ssp. (7.3%), E. fasciculosa (4.0%), Al. verticillata (0.2%)	578	11.9	7.8	28.5	South East
Dodonaea viscosa ssp. (24.5%), Ac. pycnantha (23.4%), Ac. ligulata (22.2%), Ac. wattiana (10.3%), E. leptophylla (8.4%), E. cyanophylla (4.3%), E. socialis ssp. (2.9%), Ac. brachy botrya (1.7%), Ac. anceps (1.0%), Al. verticillata (0.7%), Ac. notabilis (0.7%)	418	13.4	1.0	3.5	Kangaroo Island

E. leucoxylon ssp. (32.3%), Ac. mearnsii (30.6%), E. fasciculosa (12.5%), E. ovata var. (12.0%), Ac. pycnantha (12.0%), Mel. lanceolata (0.5%), Callistemon rugulosus (0.1%), E. gracilis (0.1%), E. incrassata (<0.1%)	507	14.8	6.9	25.2	South East
Corymbia maculata (50.5%), E. fasciculosa (30.2%), Ac. retinodes (13.1%), E. viminalis ssp. cygnetensis (4.8%), Ac. pycnantha (1.1%), Al. verticillata (0.3%), Mel. lanceolata (<0.1%)	592	14.9	2.6	9.5	Adelaide & Mt Lofty Ranges
Casuarina cunninghamiana (66.1%), E. cladocalyx (17.8%), E. leucoxylon ssp. (7.5%), Corymbia maculata (4.0%), Ac. pycnantha (2.4%), Al. verticillata (2.3%)	515	14.9	0.8	2.8	SA Murray-Darling Basin
Al. verticillata (99.5%), E. cneorifolia (0.4%), E. diversifolia ssp. diversifolia (0.1%)	503	16.6	0.9	3.4	Kangaroo Island
Al. verticillata (44.1%), E. fasciculosa (21.2%), E. diversifolia ssp. diversifolia (18.2%), E. cneorifolia (6.9%), Mel. halmaturorum (6.7%), E. cosmophylla (1.5%), Mel. gibbosa (1.3%), Ac. retinodes var. uncifolia (0.1%)	503	16.6	1.7	6.0	Kangaroo Island

E. porosa (47.2%), E. odorata (24.0%), Callitris gracilis (8.4%), Ac. notabilis (7.4%), E. incrassata (4.3%), Ac. oswaldii (3.3%), Ac. brachybotrya (2.1%), Ac. sclerophylla var. sclerophylla (1.4%), E. socialis ssp. (0.8%), Ac. acinacea (0.7%), Ac. ligulata (0.3%), Mel. lanceolata (0.1%)	414	18.8	0.6	2.0	Adelaide & Mt Lofty Ranges
---	-----	------	-----	-----	----------------------------

Non-Eucalypt species present

Corymbia maculata	492	6.9	3.9	14.3	SA Murray-Darling Basin
Callitris gracilis	478	7.4	0.1	0.3	SA Murray-Darling Basin
Corymbia maculata	495	7.4	1.1	4.0	SA Murray-Darling Basin
Ac. implexa	478	7.4	1.9	6.9	SA Murray-Darling Basin
Corymbia maculata	655	8.4	2.8	10.3	SA Murray-Darling Basin
Corymbia maculata	492	10.8	1.8	6.6	SA Murray-Darling Basin
Allocasuarina verticillata	492	10.9	3.1	11.3	SA Murray-Darling Basin
Ac. mearnsii	492	12.5	9.2	33.8	SA Murray-Darling Basin
Casuarina cunninghamiana	585	14.9	0.7	2.6	SA Murray-Darling Basin
Casuarina cunninghamiana	585	14.9	0.9	3.5	SA Murray-Darling Basin
Casuarina cunninghamiana	465	14.9	1.2	4.3	SA Murray-Darling Basin

Allocasuarina verticillata

403

33.0

0.4

1.4

SA Murray-Darling Basin

R#2: L201-210: Original comment not addressed: “In particular, carbon removals and temporal scope require further justification for the few CN scenarios that achieve negative GHG emissions.”

See aforementioned replies relating to (1) definition of net zero, (2) sensitivity analyses of enteric methane and tree carbon vegetation, (3) critical reflection of study assumptions, and (4) dialogue relating to mismatch between constant temporal GHG emissions and diminishing sequestration articulated above.

REVIEWER COMMENTS

Reviewer #1 (Remarks to the Author):

I recommend acceptance of the manuscript

Au: We thank the reviewer for their positive feedback.

Reviewer #3 (Remarks to the Author):

Review of Manuscript #NCOMMS-23-21332B Costs of transitioning the livestock sector to net-zero emissions under future climates
Submitted to Nature Communications

General Comments:

This study's approach, incorporating input from a group of industry practitioners to codesign pathways for achieving net zero emissions in sheep and beef enterprises in Australia, represents an innovative action-oriented partnership that deserves a wide audience.

Au: Thank you for your positive feedback on the approach used in our manuscript. We agree that the study will appeal to a wide audience.

The paper has strong merit as an example of much needed transdisciplinary research, with both biophysical scientists and social scientists engaging with stakeholders to co-develop research scenarios, approaches, and criteria. The authors have addressed most previous concerns, although the discussion and conclusions can still be improved to better convey the study's assumptions, scope, and system boundaries. In particular, the terminology about carbon mitigation and carbon prices is inconsistent, as further discussed below.

This manuscript bridges several distinct domains of science, policy, and practice where some of the key terminology is used in different ways. For the most part the manuscript uses these terms in accordance with the emerging consensus definitions in the carbon dioxide removal (CDR) and emissions reduction community, although there are a few minor exceptions pointed out in the specific comments. However, it would be helpful to provide a different set of citations to support the overview of these terms in lines 46 to 50, as the existing citation (Pan et al. 2023) is not consistent with these consensus definitions, calling "removals" via carbon sequestration an "emissions reduction". The terms where definitions should be clear and consistent include:

- mitigation: activities that limit greenhouse gas emissions from entering the atmosphere and/or reduce their levels in the atmosphere. (IPCC

2022 https://www.ipcc.ch/report/ar6/wg3/downloads/faqs/IPCC_AR6_WGIII_FAQ_Chapter_01.pdf);

- reduction and avoidance: reducing or eliminating the amount of greenhouse gases entering the atmosphere.
- removal: the withdrawal of greenhouse gases from the atmosphere as a result of deliberate human activities (IPCC 2024 <https://apps.ipcc.ch/glossary/>);
- insetting: activities that reduce emissions or achieve carbon removal within a farm, enterprise or organization (WEF 2022 <https://www.weforum.org/stories/2022/03/carbon-insetting-vs-offsetting-an-explainer/>);
- offsetting: the compensation for emissions from a given source (e.g., within a farm, enterprise, or organization) with a carbon credit that represents an equivalent emissions reduction or carbon removal accomplished elsewhere outside of the farm, enterprise or organization. (IPCC 2024: The reduction, avoidance or removal of a unit of greenhouse gas (GHG) emissions by one entity, purchased by another entity to counterbalance a unit of GHG emissions by that other entity. Offsets are commonly subject to rules and environmental integrity criteria intended to ensure that offsets achieve their stated mitigation outcome. Relevant criteria include, but are not limited to, the avoidance of double counting and leakage, use of appropriate baselines, additionality, and permanence or measures to address impermanence. <https://apps.ipcc.ch/glossary/>);
- carbon credit: a market instrument in the form of a tradable certificate representing one metric ton of carbon dioxide equivalent emission reduction, avoidance, or removal as a result of a project, intervention, or activity. (WEF, 2020 <https://www.weforum.org/stories/2020/11/carbon-credits-what-how-fight-climate-change/>)

While many analyses equate reduction, avoidance, and removal, those are not the same in terms of long-term climate solutions. Reduction or avoidance of emissions can slow the trajectory of climate change but cannot reverse it. Actual removals are needed both to counterbalance residual emissions that cannot be reduced or avoided, and to draw down atmospheric carbon dioxide to return the climate to a safe operating space. Because the range of mitigation practices evaluated in this study includes reductions as well as removals, insets, offsets, and credits the authors should make sure they provide clear definitions of these terms early in the manuscript and then use them consistently throughout. As discussed further in the specific comments, the authors should also clarify the prices associated with carbon taxes and/or carbon credits, which appear to currently be used interchangeably.

Au: Following these suggestions, we clarified terminology regarding mitigation (reduction, avoidance, removals), insets, offsets, taxes and credits, in each case citing appropriate references.

We added the following to the introduction:

As such, pathways to net-zero GHG emissions typically involve a combination of mitigation strategies⁸ aimed limiting GHG emissions from entering the atmosphere (e.g. provision of livestock feed additives to reduce enteric methane) and/or reducing atmospheric GHG through additional removals (e.g. through additional soil organic carbon [SOC] sequestration)⁹⁻¹¹. ‘Mitigation’ thus can be either reduction and avoidance (reducing and/or eliminating the quantum of GHG entering the atmosphere) and/or removal (withdrawal of GHG from entering the atmosphere as a result of deliberate human activities)⁸⁻¹². Mitigation can be via ‘insetting’, where new removals or reduction are used to counterbalance baseline GHG within the enterprise (e.g. a practice change resulting in additional SOC sequestration¹¹) or value chain (e.g. reducing GHG emissions associated with transport of farm products)¹³. Greenhouse gas emissions from an entity may also be ‘offset’, defined as the reduction, avoidance or removal of a GHG unit by one entity, purchased by another entity to counterbalance a unit of GHG emissions by that other entity¹²⁹. Offsets are subject to rules and environmental integrity criteria to ensure that carbon credits have their stated mitigation, including avoidance of double counting and leakage, use of appropriate baselines, additionality, transparency, conservatism, permanence (or measures to address impermanence) and the extent to which they are measurable and independently verified¹²⁹. Offsets can be quantified in ‘carbon credits’, a market instrument in the form of a tradable certificate representing one metric tonne of carbon dioxide-equivalent emission reduction, avoidance, or removal as a result of a project, intervention, or activity^{14,130}. Carbon taxes are monetised levies on net positive GHG emissions of an entity, and are often greater than carbon credits to account for the true damage society causes by each incremental metric tonne of CO₂ emissions^{14,62, 124}.

- 8 IPCC. “Frequently Asked Questions”, Sixth Assessment Report, Working Group 3.
https://www.ipcc.ch/report/ar6/wg3/downloads/faqs/IPCC_AR6_WGIII_FAQ_Chapter_01.pdf (2022).
- 9 Beillouin, D. et al. A global overview of studies about land management, land-use change, and climate change effects on soil organic carbon. *Glob. Chang. Biol.* 28, 1690-1702 (2022). <https://doi.org/10.1111/qcb.15998>
- 10 del Prado, A. et al. Feed additives for methane mitigation: Assessment of feed additives as a strategy to mitigate enteric methane from ruminants - Accounting; How to quantify the mitigating potential of using antimethanogenic feed additives. *J. Dairy Sci.* 108, 411-429 (2025).
<https://doi.org/10.3168/jds.2024-25044>
- 11 Henry, B. et al. Soil carbon sequestration in rangelands: a critical review of the impacts of major management strategies. *Rangeland J.* 46, 1-27 (2024).

- 12 Kreibich, N. Toward global net zero: The voluntary carbon market on its quest to find its place in the post-Paris climate regime. *WIREs Clim Change* 15, e892 (2024). <https://doi.org/10.1002/wcc.892>
- 13 Ebersold, F., Hechelmann, R.-H., Holzapfel, P. and Meschede, H. 2023. Carbon insetting as a measure to raise supply chain energy efficiency potentials: Opportunities and challenges. *Energy Conversion and Management: X* 20, 100504.
- 14 Maestre-Andrés S, Drews S, Savin I, van den Bergh J. Carbon tax acceptability with information provision and mixed revenue uses. *Nat. Commun.* 12, 7017 (2021). <https://doi.org/10.1038/s41467-021-27380-8>
- 129 IPCC. International Panel on Climate Change Annex I: Glossary [van Diemen, R., J.B.R. Matthews, V. Möller, J.S. Fuglestvedt, V. Masson-Delmotte, C. Méndez, A. Reisinger, S. Semenov (eds)]. In IPCC, 2022: *Climate Change 2022: Mitigation of Climate Change. Contribution of Working Group III to the Sixth Assessment Report of the Intergovernmental Panel on Climate Change* [P.R. Shukla, J. Skea, R. Slade, A. Al Khourdajie, R. van Diemen, D. McCollum, M. Pathak, S. Some, P. Vyas, R. Fradera, M. Belkacemi, A. Hasija, G. Lisboa, S. Luz, J. Malley, (eds.)]. Cambridge University Press, Cambridge, UK and New York, NY, USA. doi: 10.1017/9781009157926.020. (2022).
- 130 Hartmann, T. & Broom, D. What are carbon credits and how can they help fight climate change? (2020). <https://www.weforum.org/stories/2020/11/carbon-credits-what-how-fight-climate-change/>.

We also clarified the cost of carbon taxes in the methods with the following:

We applied a hybrid mechanism to GHG emissions post-intervention, where carbon taxes were imposed on residual net-positive GHG emissions and carbon credits were issued where net farm GHG emissions were negative. Carbon taxes were defined as the cost of reducing net-positive farm GHG emissions to net-zero, with the quantum of mitigation required being the equivalent of the annual net-positive GHG averaged over the simulation duration. Farm businesses with net-negative average annual GHG emissions received one carbon credit per tonne of GHG valued at AUD 28 per tonne CO₂e following Australian Carbon Credit Unit (ACCU) spot prices¹²³. Residual GHG emissions above net-zero were taxed at AUD 80 per tonne CO₂e, reflecting actual environmental and social impacts of such emissions (often referred to as the social carbon cost^{62, 124}). The price differential between carbon credits and taxes ensures that greater emphasis is placed in realisation of net-zero, rather than further mitigation when GHG status is already net negative. The AUD 80 per tonne CO₂e value attributed to

carbon taxes reflects actual social cost of carbon⁶², while the AUD 28 per tonne CO_{2e} attributed to carbon credits reflects (ACCU) market supply and demand¹²⁴. As in voluntary and compliance markets, when an entity achieves net-negative emissions (exceeding GHG neutrality), surplus reductions or removals beyond net-zero emissions can be sold.

62 Rennert K, et al. Comprehensive evidence implies a higher social cost of CO₂. Nature 610, 687-692 (2022).

123 CER. Clean Energy Regulator. Quarterly Carbon Market Report June Quarter 2022. Available at: <https://cer.gov.au/markets/reports-and-data/quarterly-carbon-market-reports/quarterly-carbon-market-report-june-quarter-2022/australian-carbon-credit-units-accus>. (2022).

124 Hutley N. A Social Cost of Carbon for the ACT. Prepared for the ACT Government. Available: https://www.climatechoices.act.gov.au/_data/assets/pdf_file/0006/1864896/a-social-cost-of-carbon-in-the-act.pdf. (2021).

We also added a new paragraph in the discussion:

Carbon taxation is a crucial tool for combating climate change, often designed to increase public acceptance by compensating low-income households and providing funding for climate projects¹³. Carbon taxes are designed to reflect the social cost of carbon (SCC), which monetise damages caused by each incremental tonne of CO_{2e} emitted to long-term environmental effects, such as sea-level rise, extreme weather events and agricultural yield losses⁶². For example, the US government uses the SCC to measure benefits associated with reducing CO_{2e} emissions in its mandatory regulatory analysis of more than 60 economically-significant regulations, including standards for appliance energy efficiency, and vehicle and power plant emissions⁶². In contrast to SCC, voluntary carbon markets reward proponents for GHG emissions mitigation via carbon credits, promoting private investment and innovation. The lack of voluntary market mandates has resulted in wide variation in the credibility and legitimacy underpinning credit quality and climate impact⁶¹. Here, we adopt a hybrid approach including both carbon taxes and carbon credits from voluntary markets. While hybrid approaches discourage greenwashing⁶¹, disparities between the SCC and voluntary carbon market spot prices risk undervaluing emissions reductions and removals⁶². Addressing these challenges through policy alignment and more robust regulation will be key to ensuring equity in attributing carbon cost and maximising climate benefits.

62 Rennert K, et al. Comprehensive evidence implies a higher social cost of CO₂. Nature 610, 687-692 (2022). <https://doi.org/10.1038/s41586-022-05224-9>

While the authors have provided substantial additional justification for the use of *Asparagopsis taxiformis* to reduce enteric methane production, two of the other interventions that are combined in the carbon neutral packages have important limitations that are not discussed. Planting lucerne, which appears to be the least cost approach to carbon removal in several scenarios (see Figures 1 and 2), does not appear to adjust its carbon removal for the nitrous oxide emissions associated with this and other legumes (e.g., the clover intervention). While these emissions may not be addressed in the models used by this research team and therefore be beyond the scope of this study, there are published studies in the literature, so this impact should at least be discussed.

Au: This is a good insight. We clarified the methods and added the following to the discussion:

Effects of pasture renovation with lucerne modelled here account for impacts of root depth, soil moisture, SOC stocks, sward crude protein and digestibility, feed intake, and nitrogenous fertiliser, among other factors¹⁰² (Table S16). Impacts of stocking rates (urine and faecal loading per area), pasture growth (mineral N use) and nitrogenous fertiliser on N₂O emissions were accounted for using equations prescribed under the Australian National GHG Inventory (SB-GAF version 1.4¹⁰²). While we note that the Australian National GHG Inventory and IPCC exclude N₂O associated with BNF^{95,102}, we acknowledge that nascent empirical experimentation demonstrates that lucerne agroecosystems can propagate significant N₂O⁹⁴ due to BNF, among other factors⁹⁶. While inclusion of such empirical data are not within the purview of the present study, we contend that impacts of BNF on N₂O at scale is worthy of deeper analysis in future, provided robust empirical datasets across a range of agroecosystems and management options are available. Economic benefits associated with lucerne in the present study were derived from their deep root infrastructure, improved SOC accrual (Tables S3-S5), and improved pasture quality, which were together were conducive to higher stocking rates, livestock production and profit (Figs 1-2). Comparison of lucerne and Talish clover in Figure 2 (a-d) indicates that lucerne had greater economic, environmental and agronomic benefits compared with Talish clover due to superior sward quality (thus livestock carrying capacity) and SOC accrual (thus carbon removals).

94 Anthony TL, Szutu DJ, Verfaillie JG, Baldocchi DD, Silver WL. Carbon-sink potential of continuous alfalfa agriculture lowered by short-term nitrous oxide emission events. *Nat. Commun.* 14, 1926 (2023).
<https://doi.org/10.1038/s41467-023-37391-2>

95 Intergovernmental Panel on Climate Change (IPCC). 2019 Refinement to the 2006 IPCC Guidelines for National Greenhouse Gas Inventories. Calvo Buendia, E., Tanabe, K., Kranjc, A., Baasansuren, J., Fukuda, M., Ngarize, S., Osako, A., Pyrozhenko, Y., Shermanau, P., and Federici, S. (eds). IPCC, Switzerland, Volume 4: Agriculture, Forestry and Other Land Use (2019). Chapter

11: Hergoualc'h, K., et al. N₂ O emissions from managed soils, and CO₂ emissions from lime and urea application.

96 Liu Y, Wu L, Baddeley JA, Watson CA. Models of Biological Nitrogen Fixation of Legumes. In: Sustainable Agriculture Volume 2 (eds Lichtfouse E, Hamelin M, Navarrete M, Debaeke P). Springer Netherlands (2011). <https://doi.org/10.1051/agro/2010008>

102 Dunn, J., Wiedemann, S. & Eckard, R. J. A Greenhouse Accounting Framework for Beef and Sheep properties based on the Australian National Greenhouse Gas Inventory methodology. (2020). Available: <http://piccc.org.au/Tools> [Accessed 4 February 2025].

Similarly, the discussion of tree planting interventions does not mention the risk of wildfire. Given the increasingly high likelihood of wildfire in Australia as well as most other terrestrial ecosystems, a buffer pool (i.e., planting trees in many locations and setting credits aside as insurance against risk of losing forest carbon due to wildfire) might be needed to ensure these insetting interventions are able to continue to counterbalance emissions in the farm enterprises. While assessing fire risk and calculating the needed buffer size and cost is beyond the scope of this study, this limitation and its directional impacts on cost and carbon removal should be discussed.

We thank the reviewer for highlighting wildfire risk. While Tasmania generally has a lower fire risk compared with mainland Australia due to cooler and wetter climate, the State is not immune to increasingly frequent extreme weather events, such as droughts and heatwaves. In response to the reviewer's comment, we included a paragraph on wildfire risk to introduce the buffer pool concept as a mitigation measure to address uncertainty and potential carbon loss from wildfire.

The following was added to the discussion:

One consideration relating to permanence of carbon sequestration via planting trees is risk of wildfire. Warmer, drier conditions borne by climate change have increased fire propensity and seasonal duration, significantly increasing areas burnt over the last decade⁷⁴. To ensure that carbon sequestered in vegetation used for insetting is prevented from re-entering the atmosphere from fire, a buffer pool may be necessary. This may include planting trees across multiple locations and setting aside some carbon credits as insurance against potential losses due to wildfires. Buffer pools act as safeguards, ensuring that each carbon credit delivers the intended CO₂ removal or avoidance, in the event that some carbon stocks are lost⁷⁵. However, this strategy may increase costs associated with insetting, potentially undermining their economic viability, particularly if farm area is small or if access to multiple locations is constrained. For instance, in California's forest carbon offset program, wildfires have solicited retirement of millions of credits from buffer pools, highlighting financial risk associated with planting trees for carbon removals⁷⁶. Risk of fire associated with any given region

could be used to compute contextualised buffer sizes, should future studies wish to add further sophistication to analytical approaches we employed here.

74 Canadell JG, et al. Multi-decadal increase of forest burned area in Australia is linked to climate change. *Nat. Commun.* 12, 6921 (2021). <https://doi.org/10.1038/s41467-021-27225-4>

75 Badgley G. Increasingly Active Wildfire Seasons Threaten the Sustainability of Forest-Backed Carbon Offset Programs. *Glob Chang Biol.* 30, e17599 (2024). <https://doi.org/10.1111/gcb.17599>

76 Badgley G, Chay F, Chegwiddden OS, Hamman JJ, Freeman J, Cullenward D. California’s forest carbon offsets buffer pool is severely undercapitalized. *Front. For. Glob. Change.* 5, 1-15 (2022). <https://doi.org/10.3389/ffgc.2022.930426>

Specific Comments:

Line 46. Replace “pathways to net zero carbon emissions” with “pathways to net zero GHG emissions” as net zero includes non-carbon solutions.

Author: Replaced as suggested.

Remove the word “mitigation”. As defined above by the IPCC, “mitigation” is an overarching term that includes both reductions and removals, The syntax of “reduce, avoid or mitigate” implies mitigation is a third option in this set, that is somehow different from reduction and avoidance, and also distinct from sequestering and removing carbon mentioned in the following phrase in the sentence. Incorporate references to support this terminology, such as those suggested in the general comments above.

Au: We removed the term “mitigation” from the said sentence and included the following references (as described above):

- 8 IPCC. “Frequently Asked Questions”, Sixth Assessment Report, Working Group 3. https://www.ipcc.ch/report/ar6/wg3/downloads/faqs/IPCC_AR6_WGIII_FAQ_Chapter_01.pdf (2022).
- 9 Beillouin, D. et al. A global overview of studies about land management, land-use change, and climate change effects on soil organic carbon. *Glob. Chang. Biol.* 28, 1690-1702 (2022). <https://doi.org/10.1111/gcb.15998>
- 10 del Prado, A. et al. Feed additives for methane mitigation: Assessment of feed additives as a strategy to mitigate enteric methane from ruminants - Accounting; How to quantify the mitigating potential of using

antimethanogenic feed additives. *J. Dairy Sci.* 108, 411-429 (2025).
<https://doi.org/10.3168/jds.2024-25044>

- 11 Henry, B. et al. Soil carbon sequestration in rangelands: a critical review of the impacts of major management strategies. *Rangeland J.* 46, 1-27 (2024).
- 12 Kreibich, N. Toward global net zero: The voluntary carbon market on its quest to find its place in the post-Paris climate regime. *WIREs Clim Change* 15, e892 (2024). <https://doi.org/10.1002/wcc.892>
- 13 Maestre-Andrés S, Drews S, Savin I, van den Bergh J. Carbon tax acceptability with information provision and mixed revenue uses. *Nat. Commun.* 12, 7017 (2021). <https://doi.org/10.1038/s41467-021-27380-8>

Line 48. Replace “offset against” with “used to counterbalance” to avoid using the word offset within the definition for inset.

Au: Replaced as suggested.

Lines 50-52. Consider specifying “anthropogenic”, as in “global anthropogenic GHG emissions”.

Au: We included the word “anthropogenic” to clarify.

Lines 55-56. Saying 40% of “global *agricultural* GHG emissions” here after saying the *agrifood* sector contributes 30% (lines 50-52) is somewhat misleading. Enteric methane represents 18% of agrifood sector emissions and 37% of farm gate emissions (based on FAO data). Suggest 1) adding text to clarify that “agricultural emissions” represent a portion of “agrifood sector” emissions, and 2) include Pg CO₂e per year for the agrifood sector, agricultural emissions (presumably farm gate), and enteric methane.

Au: The paragraph was revised as follows:

To achieve net-zero emissions, the agrifood sector must reduce carbon dioxide (CO₂), methane (CH₄) and nitrous oxide (N₂O) emissions, which, according to the Intergovernmental Panel on Climate Change (IPCC), contribute 10.8-19.1 Gt CO₂e per year, equivalent to 21-37% of global anthropogenic GHG emissions^{14, 15}. Of those agrifood emissions - comprising GHGs from agriculture, land use, storage, transport, packaging, processing, retail, and consumption¹⁴ – farm-level crop and

livestock activities account for 9–14% of global anthropogenic GHG emissions (6.2 ± 1.4 Gt CO₂e per year, increasing to 11.1 ± 2.9 Gt CO₂e per year including relevant land use)¹⁴. Proposed mitigation innovations must however go beyond consideration of GHG reduction or removal potential to further account for co-benefits and trade-offs on food security, conservation or restoration of natural resources, ecosystem services, and climate change impacts^{16, 17}. Achieving deep, sustained reduction of enteric CH₄ emissions, which in 2019 accounted for 23% of anthropogenic GHG emissions from the agrifood sector (2.85 Gt CO₂e per year)^{19,131}, or 40% of global agricultural GHG emissions, requires innovations in livestock feed types, genetics, reproduction, and health management^{20, 21}.

14 Mbow, C. et al. Food Security in Climate Change and Land: an IPCC Special Report on Climate Change, Desertification, Land Degradation, Sustainable Land Management, Food Security, and Greenhouse Gas Fluxes in Terrestrial Ecosystems. IPCC — Intergovernmental Panel on Climate Change (2019).

15 Rosenzweig C, et al. Climate change responses benefit from a global food system approach. Nat Food 1, 94-97 (2020). <https://doi.org/10.1038/s43016-020-0031-z>

18 Jackson R. B. et al. Increasing anthropogenic methane emissions arise equally from agricultural and fossil fuel sources. Env. Res. Lett. 15, 071002 (2020). <https://doi.org/10.1088/1748-9326/ab9ed2>

19 Rosa, L., Gabrielli, P. Achieving net-zero emissions in agriculture: a review. Environ. Res. Lett. 18, 063002 (2023). <https://doi.org/10.1088/1748-9326/acd5e8>

131 Nabuurs, G.-J. et al. in IPCC: Climate Change 2022: Mitigation of Climate Change. Contribution of Working Group III to the Sixth Assessment Report of the Intergovernmental Panel on Climate Change (eds P.R Shukla et al.) Cambridge University Press, Cambridge, UK and New York, NY, USA (2022).

Line 59. Specify “GHG emissions reduction” not just “reduction” since not only carbon dioxide (the term used immediately prior) is involved.

Au: Revised as suggested:

“While there has been much research of individual mitigation interventions in isolation (e.g. the influence of ewe fecundity or pasture type^{22, 23}), few studies have explored holistic implications of bundling practices aimed at simultaneous carbon dioxide removal, GHG emissions reduction and climate change adaptation.”

Lines 66-67 and Lines 85-86. Again, “mitigation” seems to be used as a subset of its consensus definition, distinct from reduction, avoidance and/or removal. In line 66 suggest replacing “mitigation” with “GHG emissions reduction, avoidance,” In lines 85-86 suggest swapping “reduce” and “mitigation” so that it reads “*mitigate* GHG emissions (e.g., through carbon removals, *and* GHG emissions avoidance *and reduction*”)

Au: Revised as suggested:

“This has resulted in ‘carbon myopia’ phenomena¹⁷, wherein only GHG emissions reduction, avoidance or carbon removals are assessed with an innovation purported for GHG mitigation”.

Lines 85-86 now reads as follows:

“While land managers have multiple opportunities to mitigate GHG emissions (e.g. through carbon removals, and GHG emissions avoidance and reduction), scientific literature that develops and contrasts economic pathways to carbon neutral farming systems is scarce.”

Lines 154-155. Was the impact on nitrous oxide emissions from lucerne included in this analysis? Continuous lucerne/alfalfa can have significant nitrous oxide emissions (e.g., <https://www.nature.com/articles/s41467-023-37391-2>).

Au: We added the following to the discussion:

Effects of pasture renovation with lucerne modelled here account for impacts of root depth, soil moisture, SOC stocks, sward crude protein and digestibility, feed intake, and nitrogenous fertiliser, among other factors¹⁰² (Table S16). Impacts of stocking rates (urine and faecal loading per area), pasture growth (mineral N use) and nitrogenous fertiliser on N₂O emissions were accounted for using equations prescribed under the Australian National GHG Inventory (SB-GAF version 1.4¹⁰²). While we note that the Australian National GHG Inventory and IPCC exclude N₂O associated with BNF^{95,102}, we acknowledge that nascent empirical experimentation demonstrates that lucerne agroecosystems can propagate significant N₂O⁹⁴ due to BNF, among other factors⁹⁶. While inclusion of such empirical data are not within the purview of the present study, we contend that impacts of BNF on N₂O at scale is worthy of deeper analysis in future, provided robust empirical datasets across a range of agroecosystems and management

options are available. Economic benefits associated with lucerne in the present study were derived from their deep root infrastructure, improved SOC accrual (Tables S3-S5), and improved pasture quality, which were together were conducive to higher stocking rates, livestock production and profit (Figs 1-2). Comparison of lucerne and Talish clover in Figure 2 (a-d) indicates that lucerne has greater economic, environmental and agronomic benefits compared with Talish clover due to superior sward quality (thus livestock carrying capacity) and SOC accrual (thus carbon removals).

We clarified the methods in this vein.

94 Anthony TL, Szutu DJ, Verfaillie JG, Baldocchi DD, Silver WL. Carbon-sink potential of continuous alfalfa agriculture lowered by short-term nitrous oxide emission events. *Nat. Commun.* 14, 1926 (2023). <https://doi.org/10.1038/s41467-023-37391-2>

95 Intergovernmental Panel on Climate Change (IPCC). 2019 Refinement to the 2006 IPCC Guidelines for National Greenhouse Gas Inventories. Calvo Buendia, E., Tanabe, K., Kranjc, A., Baasansuren, J., Fukuda, M., Ngarize, S., Osako, A., Pyrozhenko, Y., Shermanau, P., and Federici, S. (eds). IPCC, Switzerland, Volume 4: Agriculture, Forestry and Other Land Use (2019). Chapter 11: Hergoualc'h, K., et al. N₂O emissions from managed soils, and CO₂ emissions from lime and urea application.

96 Liu Y, Wu L, Baddeley JA, Watson CA. Models of Biological Nitrogen Fixation of Legumes. In: Sustainable Agriculture Volume 2 (eds Lichtfouse E, Hamelin M, Navarrete M, Debaeke P). Springer Netherlands (2011). <https://doi.org/10.1051/agro/2010008>

102 Dunn, J., Wiedemann, S. & Eckard, R. J. A Greenhouse Accounting Framework for Beef and Sheep properties based on the Australian National Greenhouse Gas Inventory methodology. (2020). Available: <http://piccc.org.au/Tools> [Accessed 4 February 2025].

Line 155. The manuscript states that “trees reduced productive pasture area and livestock carrying capacity”. It would be helpful to provide some context. While the beef scenario was largely eucalyptus and the sheep scenario included more diverse species, it was not designed for forage. However, some tree species can provide feed and forage for sheep and cattle as well as shade.

Au: Thanks. We assume trees did not impact on livestock production for conservatism:

This occurred because lucerne enabled pasture growth and improved livestock production, whereas planting trees were assumed to represent a new investment

(beef farm) or to occur within remnant vegetation (sheep farm), with no ensuing effect on livestock production. While we made this assumption for conservatism, we acknowledge that some tree species could provide productivity co-benefits via provision of forage and shelter^{42, 43}.

- 42 Masters DG, Blache D, Lockwood AL, Maloney SK, Norman HC, Refshauge G, Hancock SN. Shelter and shade for grazing sheep: implications for animal welfare and production and for landscape health. *Anim. Prod. Sci.* 63, 623-644 (2023). <https://doi.org/10.1071/AN22225>
- 43 Pent GJ, Fike JH. Enhanced Ecosystem Services Provided by Silvopastures. In: *Agroforestry and Ecosystem Services* (eds Udawatta RP, Jose S). Springer International Publishing (2021). https://doi.org/10.1007/978-3-030-80060-4_7

Line 722 notes “(silvopasture) was not permissible following advice from the RRG”. While the rationale for this decision is not described, this may reflect regional circumstances, and silvopasture may be a preferred practice in other regions. Additional context here and/or at line 722 would minimize the potential for this result to be incorrectly generalized and discourage silvopasture where it is appropriate.

We revised as follows:

“...(silvopasture) was not permissible following advice from the RRG that such farming systems would require additional knowledge and practical expertise to successfully operationalise (elsewhere however, silvopasture has been promulgated as a sustainable management practice, having benefits for livestock productivity via provision of shade and shelter, carbon sequestration, and biodiversity conservation^{120,121}).”

- 120 Zeppetello LRV, et al. Consistent cooling benefits of silvopasture in the tropics. *Nat. Commun.* 13, 708 (2022). <https://doi.org/10.1038/s41467-022-28388-4>
- 121 Amorim HCS, Ashworth AJ, O’Brien PL, Thomas AL, Runkle BRK, Philipp D. Temperate silvopastures provide greater ecosystem services than conventional pasture systems. *Sci. Rep.* 13, 18658 (2023). <https://doi.org/10.1038/s41598-023-45960-0>

Line 167. In the legend for figure 2, which is about sheep farms, CCD (changing calving date) is defined but not shown, and the figure instead shows Alt LD (alternative lambing date?) which is not defined. Please clarify.

Au: This should have referred to lambing rather than calving. We revised the caption to clarify:

Fig. 2. Production, profit, adoptability, mitigation (a and b) and marginal abatement cost curves (c and d) of thematic adaptation/mitigation interventions for sheep farming systems. Interventions were co-designed with a Regional Reference Group (RRG) of expert practitioners for 2030 (a and c) and 2050 (b and d) climates. Purple stars depict baseline scenario. Total emissions for the baseline shown in parenthesis in (a) and (b). Alt. LD: altered lambing date; Alt. LD/SR: altered lambing date and increased stocking rate; Asp: *Asparagopsis taxiformis* as a feed supplement; CH₄ vac: injecting animals with an enteric CH₄ inhibitor vaccine; Deep-Root: increasing pasture root depth with perennial legume renovation; FCE: increasing livestock feed conversion efficiency; SR: increasing stocking rate; T. Clover: introduction of Talish clover (*Trifolium tumens*); TFCE: transformational increase in livestock feed conversion efficiency.

Line 172. Reviewer #1 pointed out “evidence around important measures such as biochar feeding is also patchy and at least needs more rigorous evidence.” In response the authors added citations (Line 172) but should also mention that there are upstream emissions associated with biochar production that were not considered in this study. These upstream emissions could potentially reduce the climate benefits of this intervention.

Au: In response to this comment, and to add depth to our response to the prior reviewer, we amended as follows:

However, the aforementioned studies do not account for upstream (pre-farm) GHG emissions associated with biochar production⁴⁸, which may reduce perceived climate benefits at the farm scale. Even so, biochar has many benefits than those associated with mitigation, including recycling of agricultural or forestry waste in line with the circular economy⁴⁹, and potential to displace energy derived from fossil-fuels through electricity generation and eucalyptus oil production⁵⁰. These observations underline a need for more systematic, holistic assessments that evaluate environmental, agronomic and economic implications of using biochar as a livestock feed supplement.

48 Norgate T., Haque N., Somerville M., Jahanshahi S. The greenhouse gas footprint of charcoal production and of some applications in steelmaking. In: 7th Australian conference on life cycle assessment (2011). Available at: https://www.researchgate.net/profile/Sharif-Jahanshahi/publication/281296295_Nuclear_power_and_greenhouse_gases_-_fuelling_the_debate/links/55e89d5608ae65b638998ae9/Nuclear-power-and-greenhouse-gases-fuelling-the-debate.pdf

49 Hedley M, Camps-Arbestain M, McLaren S, Jones J, Chen Q. A review of evidence for the potential role of biochar to reduce net GHG emissions from New Zealand agriculture. Retrieval from:

<https://www.nzagrc.org.nz/assets/Publications/Potential-Role-of-Biochar-in-NZ-2021.pdf>

50 Chandrasekaran S, Jadhav S, Mari Selvam S, Krishnamoorthy N, Balasubramanian P. Biochar-based materials for sustainable energy applications: A comprehensive review. *J. Environ. Chem. Eng.* 12, 114553 (2024). <https://doi.org/10.1016/j.jece.2024.114553>

Lines 182-184. Given the 15% to 20% increase in protein emissions intensity from the purchase of additional land in a diverse agro-climatic region, it would be helpful if the authors summarized the primary driver(s) of these increased GHG emissions. It would also be helpful to point the reader to the supplemental information about this particularly complex intervention, especially table S17 detailing this strategy for adaptation.

Au: We thank the reviewer for their opinion. The paragraph was revised as follows:

While the new land received a similar annual rainfall quantum as that of the existing farm in north-western Tasmania, the sandy loam soils (Kurosols) of the north-east were less productive than those of the clay-rich Ferrosols in the north-west⁵¹, reducing annual pasture production by around 40% (Tables S3, S4, S17). As such, the sustainable stocking rate of the north-east was lower than that in the north-west, as was SOC accrual (Table S17). Taken together, these factors were conducive to higher net GHG per unit livestock production, resulting in greater net GHG emissions per unit protein for the enterprise (Tables S3-S4). Our analysis of the purchasing of additional land highlights the tight coupling between livestock numbers and GHG emissions, as well as effects of SOC sequestration on protein emissions intensity. While purchasing land in a distinct agroecological zone arguably reduces enterprise risk of exposure to simultaneous extreme weather events, purchasing additional land may not necessarily improve enterprise carbon footprint (Tables S3-S4).

51 Cotching WE, Oliver G, Downie M, Corkrey R, Doyle RB. Land use and management influences on surface soil organic carbon in Tasmania. *Soil Res.* 51, 615-630 (2013). <https://doi.org/10.1071/SR12251>

Lines 234-242, caption for Figure 3 and (see comment immediately below)

Lines 274-281, caption for Figure 4: These figures are central to reporting the results of the holistic analysis for various “stacked” combinations of practices. The left side of the figures includes modeled quantities (production, profit, emissions) with explicit units, while the right side of the figures aggregates the three variables in (a) and (d) into the stacked horizontal bar charts in (b) and (e). Because of the structure of this journal your description of that conversion (the normalization in the section titled “Normalised

multidimensional impact assessments” is not anywhere near the figure. Please indicate this is a normalization in the figure caption and/or in the text here, and also refer the reader to that later section. It would also be helpful to explain how (b) is converted to (c) and (e) is converted to (f), although the axis legends in the pyramids (c) and (f) stating % factor provide a hint, presumably meaning that these are the percentages each variable contributes to the totals in (b) and (e). But that may not be obvious to many readers.

Au: We clarified as follows:

Fig. 3. Livestock production, profit (pre-carbon tax), and net farm emissions for co-designed thematic adaptations to beef farming systems under 2030 (a, b, c) and 2050 (d, e, f) climate horizons. Bar charts on the left depict simulated values with dimensions shown on vertical axes (a and d) for livestock production (top), profit (centre) and net GHG emissions (bottom). These values were normalised by the greatest corresponding value for each metric (see Methods: Normalised Multidimensional Impact Assessments) in the stacked horizontal bar charts (b and e) for multidimensional impact assessment. Normalised values for each metric in (b), (c) and (e) range from zero to one. Ternary plots (c and f) show normalised net emissions, profit and livestock production as well as ease of adoption attributed by the regional reference group. Hist: historical climates; Base: existing farming system under future climates; LHF: low-hanging fruit package; TCN: towards carbon neutral package; ID: income diversification; Asp: Asparagopsis taxiformis as a feed supplement; Asp + PT (Asp + planting 50 ha trees); TFCE, adopting livestock genotypes with transformational feed conversion efficiency; CN1: carbon neutral package 1 (Asp + TFCE + planting 50 ha trees), CN2: carbon neutral package 2 (Asp + TFCE + 55 ha trees 2030 and 110 ha trees 2050), CN3: carbon neutral package 3 (Asp + renovating pastures with lucerne + planting 50 ha trees); CN4: carbon neutral package 4 (Asp + renovating pastures with lucerne + 55 ha trees 2030 and 110 ha trees 2050).

Fig. 4. Livestock production, profit (pre-carbon tax), and net farm emissions for co-designed thematic adaptations to sheep farming systems under 2030 (a, b, c) and 2050 (d, e, f) climate horizons. Bar charts on the left depict simulated values with dimensions shown on vertical axes (a and d) for livestock production (top), profit (centre) and net GHG emissions (bottom). These values were normalised by the greatest corresponding value for each metric (see Methods: Normalised Multidimensional Impact Assessments) in stacked horizontal bar charts (b and e) for multidimensional impact assessment. Normalised values for each metric in (b), (c) and (e) range from zero to one. Ternary plots (c and f) show normalised net emissions, profit and livestock production as well as ease of adoption attributed by the regional reference group. Hist: historical climates; Base: existing farming system under future climates; LHF: low-hanging fruit package; TCN: towards

carbon neutral package; ID: income diversification; Asp: Asparagopsis taxiformis as a feed supplement; TFCE, adopting livestock genotypes with transformational feed conversion efficiency; Asp + PT (Asp + planting 200 ha trees); CN1: carbon neutral package 1 (Asp + TFCE + planting 200 ha trees), CN2: carbon neutral package 2 (Asp + TFCE + 220 ha trees), CN3: carbon neutral package 3 (Asp + renovating pastures with lucerne + planting 200 ha trees); CN4: carbon neutral package 4 (Asp+ renovating pastures with lucerne + 220 ha trees).

Line 247. The explanation of the carbon tax is much later in the methods section, so here, where the results are reported, a quick mention of the specific tax rate would provide important context to these results. For example, “...(business as usual), carbon taxes of \$XX/Mg CO₂e would...”. As noted elsewhere in this review, a range of \$60-\$100/Mg CO₂e is provided in Line 739 but it is not clear whether the authors used an average of that range (\$80/Mg CO₂e), different prices for the 2030 and 2050 time periods, or something else. Please also see other comments about the use of the term carbon credits, apparently as a synonym for carbon taxes, without explanation.

Au: These are helpful suggestions. We clarify our definitions of carbon credits and carbon taxes in the manuscript and in previous responses (please see above).

We also clarified the suggested by adding a preceding sentence:

We assessed implications of a hypothetical regulatory scenario wherein annual gains in net GHG emissions above net-zero were taxed, and additional net GHG below net-zero was credited. For consistency, we used an average carbon tax of \$80/Mg CO₂e and carbon credit spot price of \$28/Mg CO₂e in all analyses (see Methods).

We also clarified the Methods:

Carbon taxes were defined as the cost of reducing net-positive farm GHG emissions to net-zero, with the quantum of mitigation required being the equivalent of the average annual net-positive GHG over the simulation duration. Farm businesses with annual average net-negative farm GHG emissions were assumed to receive one carbon credit per tonne of GHG below net-zero. Farm businesses with net-negative GHG received \$28 per tonne CO₂ e, following Australian Carbon Credit Unit (ACCU) spot prices¹²³. Residual GHG emissions above net-zero were costed at \$80 per tonne CO₂e, reflecting actual environmental and social impacts of such emissions (often referred to as the social carbon cost^{62, 124}). The price differential between carbon credits and taxes ensures that greater emphasis is placed in realisation of net-zero (rather than further mitigation when GHG status is already net negative). As such, the \$80 per tonne CO₂e value reflects actual social cost of carbon, while the \$28 per net-negative tonne CO₂e reflects (ACCU) supply and demand. As in voluntary and

compliance markets, when an entity achieves net-negative emissions (exceeding GHG neutrality), surplus reductions or removals beyond net-zero emissions can be sold.

62 Rennert K, et al. Comprehensive evidence implies a higher social cost of CO₂. Nature 610, 687-692 (2022).

123 CER. Clean Energy Regulator. Quarterly Carbon Market Report June Quarter 2022. Available at: <https://cer.gov.au/markets/reports-and-data/quarterly-carbon-market-reports/quarterly-carbon-market-report-june-quarter-2022/australian-carbon-credit-units-accus>. (2022).

124 Hutley N. A Social Cost of Carbon for the ACT. Prepared for the ACT Government. Available at: https://www.climatechoices.act.gov.au/_data/assets/pdf_file/0006/1864896/a-social-cost-of-carbon-in-the-act.pdf (2021).

Line 251. Should this say “inset” or “counterbalanced” instead of “offset” to differentiate from a system where the farm is purchasing offset credits from outside the farm enterprise?

Au: We replaced “offset” with “counterbalanced”.

Lines 251-252. A previous reviewer asked if this should be “gross” and the authors noted it was amended in their response, but the text still says “net.” I think this should be gross GHGs, not net.

Au: Here we were referring to stacking of additional land planted with trees to net GHG emissions, as the data in Fig. 5 already include SOC sequestration.

Line 257. It appears a word is missing, presumably “reduce”. Should it read “significantly reduce net GHG emissions.”

Au: Revised as suggested.

Line 271. Should this say carbon *credit* instead of carbon

offset? **Au: Revised as suggested.**

Lines 340-341 and Line 356. Although previously (see comments for Lines 246-247) and in later in the methods section (Lines 739-741) the manuscript references the price of a carbon tax, here the terminology seems to have shifted to carbon credits. Is the cost of a market-based carbon credit (that presumably is backed up by some sort of certified carbon removal) assumed to be the same as a government required carbon tax (noting that a carbon tax may or may not result in carbon removal)? Please clarify here and in the methods section, and then be consistent about terminology. At this point the manuscript uses the term carbon tax thirteen times and carbon credit ten times, so a manual search-and replace (to assure correct syntax) would not be too onerous.

Au: We have clarified our definitions of carbon taxes and carbon credits in the manuscript on first mention, in the responses above, and in detail in the methods. Please see previous responses.

Line 342. Does “direct removal” mean insetting with carbon removal? If so, the authors should clarify.

Au: Revised as suggested.

Line 366. Here the authors define “insetting” with a parenthetical “(practices to reduce GHG within the value chain). The term “value chain” can refer to a single enterprise, but often extends across multiple enterprises as a product travels from producer to consumer. As previously indicated, emerging consensus defines “insetting” as within a single enterprise. Please clarify this definition for consistency.

Au: We replaced “value chain” with “enterprise” for consistency with text elsewhere in the manuscript.

Line 378. This sentence starts “Future market demand for carbon credits (e.g., from the transport and energy production sectors)”. It would be helpful to mention that in the future carbon credits will be needed in the future to draw down atmospheric carbon (achieve negative emissions) to bring the climate back to a normal state, in addition to counterbalancing residual emissions that are difficult or impossible to avoid entirely. The authors may also want to change the examples from “transport and energy production sectors” to “other agriculture, industry, and transport” as the scientific literature anticipates the majority of residual emissions will be from agriculture with some from industry and transport sectors (e.g., Buck et al. 2022).

Au: Revised as follows:

As the need to counterbalance residual emissions and draw down atmospheric carbon intensifies^{71,72}, future market demand for carbon credits from the agriculture, industry and transport sectors is expected to rise⁶⁷. Higher carbon prices could evoke appropriation of land currently used for commodity production, as markets often dictate land-use decisions⁴². Such shifts at scale, coupled with higher carbon prices, may have significant implications for food and fibre supply, potentially exacerbating nutrition insecurity or poverty, particularly in regions where population growth increases demand for these commodities. Jurisdictions prioritising carbon storage and environmental conservation may unintentionally cause carbon or biodiversity leakage⁴⁸, where commodity-based production originally occurring there shifts to other regions, driving a cascade of land clearing and biodiversity loss in areas historically serving as global carbon sinks or biodiversity hot-spots (e.g., the Amazon rainforest). Such trade-offs highlight the critical need for harmonised global policies and legislation which ensure that carbon markets do not unintentionally undermine food security, social equity, or environmental sustainability.

71 Ho, D.T. Carbon dioxide removal is not a current climate solution - we need to change the narrative. *Nature*, 616 (2023), p. 9.
<https://doi.org/10.1038/d41586-023-00953-x>

72 Buck HJ, Carton W, Lund JF, Markusson N. Why residual emissions matter right now. *Nat. Clim. Change*. 13, 351-358 (2023).
<https://doi.org/10.1038/s41558-022-01592-2>

Line 388-411. This discussion about permanence and temporal variation of nature-based carbon removal strategies, including planting trees, would be a good place for a brief note about the risks of losing carbon stored in trees through wildfires. As mentioned in the general comments, across the globe forests are at increasing risk of wildfire, including in the region in eastern Australia and Tasmania targeted by this study. It's important to at least mention this risk, and that buffer pools (i.e., setting aside a certain fraction of carbon credits generated by a project as insurance against unanticipated losses), which were not included in the analysis, would likely need to be included for effective implementation of this concept. Here, or elsewhere in the paper, the authors should also mention that carbon credits / offsets have requirements, including e.g., additionality and permanence, that may differ from the effects of the insetting practice assessed here (planting trees). For example, there are several biomass carbon removal and storage (BiCRS) approaches that result in permanent or geologic sequestration. For the global "livestock sector" to reach net zero emissions, permanent removals will likely be required, and while these were not explored in this study they deserve mention in the discussion.

Au: Following this suggestion, we now account for implications of wildfires, need for buffer pools and permanence associated with BiCRS:

While permanence refers to the longevity of the carbon sequestration, ‘additionality’ ensures that GHG emissions reduction or removal would not have occurred without incentive provided by carbon credits^{67, 68}, ie. GHG mitigation is additional to mitigation that may have otherwise occurred. Obligations under carbon offsetting programs may differ from those of insetting. For example, there are several biomass carbon removal and storage (BiCRS) approaches that result in permanent or geologic sequestration⁶⁹, offering greater durability than temporary carbon storage achieved through tree planting, although perhaps less biodiversity benefits. BiCRS combines the natural ability of plants to convert carbon dioxide into biomass with human engineering to store the biomass or derivatives thereof, such as biochar⁷⁰, in a manner that prevents carbon from re-entering the atmosphere. For the global livestock sector to achieve net-zero emissions, permanent removals will likely be necessary; BiCRS would be a prospective intervention worthy of investigation in this regard.

In response to the reviewer comment relating to offset requirements, we refer to our earlier response and addition to the manuscript:

Offsets are subject to rules and environmental integrity criteria to ensure that carbon credits achieve their stated mitigation, including avoidance of double counting and leakage, use of appropriate baselines, additionality, transparency, conservatism, permanence (or measures to address impermanence) and the extent to which they are measurable and independently verified¹²⁹. Offsets can be quantified in ‘carbon credits’, a market instrument in the form of a tradable certificate representing one metric tonne of carbon dioxide-equivalent emission reduction, avoidance, or removal as a result of a project, intervention or activity^{14,130}.

67 Matthews HD, Zickfeld K, Koch A, Luers A. Accounting for the climate benefit of temporary carbon storage in nature. Nat. Commun.14, 5485 (2023). <https://doi.org/10.1038/s41467-023-41242-5>

68 Swinfield T, Shrikanth S, Bull JW, Madhavapeddy A, zu Ermgassen SOSE. Nature-based credit markets at a crossroads. Nat. Sustain. 7, 1217-1220 (2024). <https://doi.org/10.1038/s41893-024-01403-w>

69 Dees JP, Sagues WJ, Woods E, Goldstein HM, Simon AJ, Sanchez DL. Leveraging the bioeconomy for carbon drawdown. Green Chem. 25, 2930-2957 (2023). <https://doi.org/10.1039/D2GC02483G>

70 Lehmann J, et al. Biochar in climate change mitigation. Nat. Geosci. 14, 883-892 (2021). <https://doi.org/10.1038/s41561-021-00852-8>

One consideration relating to permanence of carbon sequestration via planting trees is risk of wildfire. Warmer, drier conditions borne by climate change have increased fire propensity and seasonal duration, significantly increasing areas

burnt over the last decade⁷⁴. To ensure that carbon sequestered in vegetation used for insetting is prevented from re-entering the atmosphere from fire, a buffer pool may be necessary. This may include planting trees across multiple locations and setting aside some carbon credits as insurance against potential losses due to wildfires. Buffer pools act as safeguards, ensuring that each carbon credit delivers the intended CO₂ removal or avoidance, in the event that some carbon stocks are lost⁷⁵. However, this strategy may increase costs associated with insetting, potentially undermining their economic viability, particularly if farm area is small or if access to multiple locations is constrained. For instance, in California's forest carbon offset program, wildfires have solicited retirement of millions of credits from buffer pools, highlighting financial risk associated with planting trees for carbon removals⁷⁶. Risk of fire associated with any given region could be used to compute contextualised buffer sizes, should future studies wish to add further sophistication to analytical approaches we employed here.

74 Canadell JG, et al. Multi-decadal increase of forest burned area in Australia is linked to climate change. *Nat. Commun.* 12, 6921 (2021). <https://doi.org/10.1038/s41467-021-27225-4>

75 Badgley G. Increasingly Active Wildfire Seasons Threaten the Sustainability of Forest-Backed Carbon Offset Programs. *Glob Chang Biol.* 30, e17599 (2024). <https://doi.org/10.1111/gcb.17599>

76 Badgley G, Chay F, Chegwiddden OS, Hamman JJ, Freeman J, Cullenward D. California's forest carbon offsets buffer pool is severely undercapitalized. *Front. For. Glob. Change.* 5, 1-15 (2022). <https://doi.org/10.3389/ffgc.2022.930426>

Line 394. There appears to be a typo, where “avoiding” should be “avoid”.

Au: Revised as suggested.

Line 412, Figure 5. In the figure, suggest adding sheep and beef labels to the a-d and eh plot for increased clarity. In the caption, please indicate the price of the purchased carbon credits (\$80/Mg CO₂e?) as well as the sold carbon credits (\$28/Mg CO₂e). As mentioned elsewhere, please be consistent about terminology. For example, here in this one figure the legend includes both carbon credits and carbon tax on lines 413 and 414, and uses “post carbon tax”, “bought carbon credits”, and “sold carbon credits”. If the authors are going to use the “carbon credit” terminology instead of “carbon tax”, then please be consistent.

Au: We revised terminology throughout the paper, including the caption of the said figure. We note that *both* terms are necessary, for they represent different financial instruments and costs per unit CO₂e. See previous responses.

Fig. 5. Pathways to net-zero emissions (red and light green bars) with associated carbon taxes (blue), post-carbon tax profit (dark green), and income from selling carbon credits (yellow) across climate horizons and thematic adaptations for the beef farm (a-d) and sheep farm (e-h). Stacked bars (c, d, g, h) represent operating profit before deductions from carbon taxes (\$80/Mg CO₂e) or income from carbon credits (\$28/Mg CO₂e; see Methods for further details). Pathways 1 and 2 reflect net-zero farming systems attained by improving animal genetics (CN1 and CN2) or renovating pasture swards with lucerne (CN3 and CN4). Base: performance in 2030 or 2050; ASP: *A. taxiformis* as livestock feed supplement; Asp+PT: *A. taxiformis* + planting trees (50 ha); CN1: carbon neutral package 1 [*A. taxiformis* + planting trees, 50 ha (beef farm) or 200 ha (sheep farm) + transformational feed conversion efficiency]; CN2: carbon neutral package 2 [*A. taxiformis* + planting trees, 55 ha in 2030 and 110 ha in 2050 (beef farm) or 220 ha (sheep farm) + transformational feed conversion efficiency]; CN3: carbon neutral package 3 [*A. taxiformis* + planting trees, 50 ha (beef farm) or 200 ha (sheep farm) + Lucerne]; CN4: carbon neutral package 4 [*A. taxiformis* + planting trees, 55 ha in 2030 and 110 ha in 2050 (beef farm) or 220 ha (sheep farm) + Lucerne]; OP, operating profit; CT, carbon tax.

Line 437. This is another good place to include the amount of the carbon tax/carbon credit price (whichever is selected), as in “after an \$XX Mg⁻¹ C-1 carbon tax of AUD 321-340K...”

Au: Thank you. In the sensitivity analysis, use of *Asparagopsis* feed supplement alone did not achieve carbon neutrality; residual GHG emissions were taxed accordingly. We clarified the text in the paper as follows:

For the beef farm, net emissions ranged from 1,484-1,709 Mg CO₂ e with a 99% CH₄ reduction, to 3,533-3,761 Mg CO₂ e for 10% CH₄ reduction. After inclusion of carbon taxes (AUD 80/Mg CO₂ e), such CH₄ reduction yielded profits of AUD 321–340K and AUD 157–176K, respectively (Figs. S3 and S5). For the sheep farm, net GHG varied from 596-871 Mg CO₂ e (99% CH₄ reduction) to 4,726-4,997 Mg CO₂ e (10% CH₄ reduction), with post-carbon tax profit ranging between AUD 1,088-1,113K and AUD 758-783K (Figs. S4 and S6).

Lines 452-453 As per the consensus definitions previously discussed, “mitigation” is the overarching term and not a subset. Suggest either eliminating the parenthesis entirely and simplify to “carbon reduction and removals given that sequestration...” or replace with something like “GHG mitigation (reduction, avoidance, inset and offset carbon sequestration) given that sequestration...”.

Au: We deleted the parenthesis.

Lines 485-486. As mentioned in the general comments, there should be a mention here of nitrous oxide implications. There is a common assumption that legumes emit little nitrous oxide, but studies have shown otherwise. For example, Anthony et al. 2023 (<https://doi.org/10.1038/s41467-023-37391-2>) report that nitrous oxide emissions reduce the potential carbon sink of alfalfa/lucerne by 14%).

Au: Revised as follows:

Effects of pasture renovation with lucerne modelled here account for impacts of root depth, soil moisture, SOC stocks, sward crude protein and digestibility, feed intake, and nitrogenous fertiliser, among other factors¹⁰² (Table S16). Impacts of stocking rates (urine and faecal loading per area), pasture growth (mineral N use) and nitrogenous fertiliser on N₂O emissions were accounted for using equations prescribed under the Australian National GHG Inventory (SB-GAF version 1.4¹⁰²). While we note that the Australian National GHG Inventory and IPCC exclude N₂O associated with BNF^{95,102}, we acknowledge that nascent empirical experimentation demonstrates that lucerne agroecosystems can propagate significant N₂O⁹⁴ due to BNF, among other factors⁹⁶. While inclusion of such empirical data are not within the purview of the present study, we contend that impacts of BNF on N₂O at scale are worthy of deeper analysis in future, provided robust empirical datasets across a range of agroecosystems and management options are available. Economic benefits associated with lucerne in the present study were derived from their deep root infrastructure, improved SOC accrual (Tables S3-S5), and improved pasture quality, which were together were conducive to higher stocking rates, livestock production and profit (Figs 1-2). Comparison of lucerne and Talish clover in Figure 2 (a-d) indicates that lucerne has greater economic, environmental and agronomic benefits compared with Talish clover due to superior sward quality (thus livestock carrying capacity) and SOC accrual (thus carbon removals).

We have also clarified the methods (see previous responses).

94 Anthony TL, Szutu DJ, Verfaillie JG, Baldocchi DD, Silver WL. Carbon-sink potential of continuous alfalfa agriculture lowered by short-term nitrous oxide emission events. *Nat. Commun.* 14, 1926 (2023). <https://doi.org/10.1038/s41467-023-37391-2>

95 Intergovernmental Panel on Climate Change (IPCC). 2019 Refinement to the 2006 IPCC Guidelines for National Greenhouse Gas Inventories. Calvo Buendia, E., Tanabe, K., Kranjc, A., Baasansuren, J., Fukuda, M., Ngarize, S., Osako, A., Pyrozhenko, Y., Shermanau, P., and Federici, S. (eds). IPCC, Switzerland, Volume 4: Agriculture, Forestry and Other Land Use (2019). Chapter 11: Hergoualc'h, K., et al. N₂ O emissions from managed soils, and CO₂ emissions from lime and urea application.

96 Liu Y, Wu L, Baddeley JA, Watson CA. Models of Biological Nitrogen Fixation of Legumes. In: Sustainable Agriculture Volume 2 (eds Lichtfouse E, Hamelin M, Navarrete M, Debaeke P). Springer Netherlands (2011). <https://doi.org/10.1051/agro/2010008>

Lines 492-495. This paragraph is providing an overview of “discoveries related to prospective pathways to net zero farm enterprises.” These specific lines summarize several interventions including purchasing land in diversified climate zones that increased productivity and profitability. While presumably this intervention was thought to be a potential pathway to net zero, it turned out to dramatically increase GHG intensity, not just for the farm but per kg of livestock live weight and protein produced. While not as dramatic, increased GHG intensity was also consistently observed for the increased stocking rate and alternate lambing/calving date interventions (see figures 1 and 2 as well as tables S3 to S6). While the RRG proposed some interventions for income and/or climate diversification (lines 523 to 532, table S16), given the stated goal of this study (pathways to net-zero) it is important to report that some interventions actually increased emissions intensity. In the case of purchasing land in a diversified climate zone this increase was quite large. This is an important discovery, and therefore should be discussed in this section.

Au: This comment has been addressed above.

Lines 669-672. These lines detail the nitrogen-related GHG emissions and are reported as including: “N₂O from nitrogenous (N) fertiliser, waste management, urinary deposition and indirect N emissions via nitrate leaching and ammonia volatilisation”. This list does not include direct N₂O emissions from nitrification/denitrification in cropland. It is possible that SB-GAF version 1.4 calculates cropland N₂O emissions as a function of N fertilizer inputs (a reasonable simplification for most cropland where synthetic fertilizers are used) and therefore does not consider N₂O derived from N fixed by legumes. However, as previously noted (see comments for lines 154-155 and lines 485-486), there are significant N₂O emissions associated with N-fixing legumes, including lucerne and presumably also talish clover. These emissions could reduce the carbon benefit of these crops, so if they are not reflected in this model that limitation should be discussed. This is especially important since legume interventions were included in the “low hanging fruit”, “toward carbon neutral” and carbon several neutral packages.

Au: We addressed this comment above and made corresponding revisions to the Methods and Discussion.

Line 739-740. Here the carbon tax price is provided as a range (\$60-\$100/Mg CO₂e) and not a specific number, while the next sentence appears to be missing some words, stating: “Any carbon sequestration beyond net farm GHG emission were ‘credited’ at \$28/Mg CO₂e [missing word or words] that of a carbon tax.” Because this is such an important component of the profitability analysis, it was previously suggested that carbon price be incorporated in a few key places in the results and discussion. In

addition, this methods section needs to be clear about the price(s) that are used and the current language and range raises several questions. Is the stated carbon tax also used as the price for purchasing carbon credits? If there is just one value used for the carbon tax, what is it? If two or more values in this range are used, what are they and when is each used? And if the carbon sequestration “credits” are sold off-farm at only \$28/Mg CO₂e, why is that price so much lower than the carbon tax and/or carbon credit purchase price?

Au: We added the following to the Methods:

We applied a hybrid mechanism to GHG emissions post-intervention, where carbon taxes were imposed on residual net-positive GHG emissions and carbon credits were issued where net farm GHG emissions were negative. Carbon taxes were defined as the cost of reducing net-positive farm GHG emissions to net-zero, with the quantum of mitigation required being the equivalent of the annual net-positive GHG averaged over the simulation duration. Farm businesses with net-negative average annual GHG emissions received one carbon credit per tonne of GHG valued at AUD 28 per tonne CO₂ e following Australian Carbon Credit Unit (ACCU) spot prices¹²³. Residual GHG emissions above net-zero were taxed at AUD 80 per tonne CO₂e, reflecting actual environmental and social impacts of such emissions (often referred to as the social carbon cost^{62, 124}). The price differential between carbon credits and taxes ensures that greater emphasis is placed in realisation of net-zero, rather than further mitigation when GHG status is already net negative. The AUD 80 per tonne CO₂e value attributed to carbon taxes reflects actual social cost of carbon⁶², while the AUD 28 per tonne CO₂e attributed to carbon credits reflects (ACCU) market supply and demand¹²⁴. As in voluntary and compliance markets, when an entity achieves net-negative emissions (exceeding GHG neutrality), surplus reductions or removals beyond net-zero emissions can be sold.

62 Rennert K, et al. Comprehensive evidence implies a higher social cost of CO₂. Nature 610, 687-692 (2022).

123 CER. Clean Energy Regulator. Quarterly Carbon Market Report June Quarter 2022. Available at: <https://cer.gov.au/markets/reports-and-data/quarterly-carbon-market-reports/quarterly-carbon-market-report-june-quarter-2022/australian-carbon-credit-units-accus>. (2022).

124 Hutley N. A Social Cost of Carbon for the ACT. Prepared for the ACT Government. Available at: https://www.climatechoices.act.gov.au/_data/assets/pdf_file/0006/1864896/a-social-cost-of-carbon-in-the-act.pdf. (2021).

Reviewer #4 (Remarks to the Author):

Au: We thank the reviewers for their constructive comments.

REVIEWERS' COMMENTS

Reviewer #3 (Remarks to the Author):

Review of Manuscript #NCOMMS-23-21332B Costs of transitioning the livestock sector to net-zero emissions under future climates

Submitted to Nature Communications

General Comments:

The paper has strong merit as an example of needed transdisciplinary research, with both biophysical scientists and social scientists engaging with stakeholders to co-develop research scenarios, approaches, and criteria. The authors did an excellent job of addressing the previous suggestions. We recommend the following minor changes and subsequent acceptance of this paper.

Response: Thank you for your positive feedback on our paper.

Specific Comments:

Line 24 "Serendipitously" should be spelled serendipitously.

Response: Amended.

Line 44: There is a missing word in this line: ...aimed at limiting GHG emissions...

Response: Amended.

Line 75 Suggest changing "global agricultural GHG emissions" to "global agricultural (farm level) GHG emissions" to be consistent with the language in lines 67-70. According to recent FAO data, enteric emissions represent 35% of global agriculture (farm gate) emissions or 18% of global agrifood systems (farm gate, pre- and post- production, and land use change).

Response: Amended.

Lines 289 and 323 (figure captions): This new phrasing with an "I" is unclear and might benefit from including mention of all subparts of this figure with normalized values, as in:... Normalised values for each metric in (b), (c), "(e), and (f)," range from zero to one.

Response: The "I" has been removed as part of the revisions to all figure and table captions in the manuscript and supplementary information. Additional comments regarding normalisation have also been added to the said captions.

Line 390 "Serendipitously" should be spelled serendipitously.

Response: Amended.

Line 400. Unfortunately this sentence is now past tense: For example, the US government has "used" the SCC to ... (see <https://www.propublica.org/article/trump-climate-change-social-cost-of-carbon-executive-order>)

Response: This sentence has been removed in response to the Editor's request to reduce the length of the discussion section.

Line 593. This appears to be the first use of the acronym BNF for Biological Nitrogen Fixation, so should be defined.

Response: Amended.

Reviewer #4 (Remarks to the Author):

Response: Thank you.